# ENMA: Tokenwise Autoregression for Generative Neural PDE Operators

**Armand Kassaï Koupaï**[1][*]   **Lise Le Boudec**[1][*]   **Louis Serrano**[1]   **Patrick Gallinari**[1,2]

[1] Sorbonne Université, CNRS, ISIR, 75005 Paris, France
[2] Criteo AI Lab, Paris, France

## Abstract

Solving time-dependent parametric partial differential equations (PDEs) remains a fundamental challenge for neural solvers, particularly when generalizing across a wide range of physical parameters and dynamics. When data is uncertain or incomplete—as is often the case—a natural approach is to turn to generative models. We introduce **ENMA**, a generative neural operator designed to model spatio-temporal dynamics arising from physical phenomena. ENMA predicts future dynamics in a compressed latent space using a generative masked autoregressive transformer trained with flow matching loss, enabling *tokenwise generation*. Irregularly sampled spatial observations are encoded into uniform latent representations via attention mechanisms and further compressed through a spatio-temporal convolutional encoder. This allows ENMA to perform in-context learning at inference time by conditioning on either past states of the target trajectory or auxiliary context trajectories with similar dynamics. The result is a robust and adaptable framework that generalizes to new PDE regimes and supports one-shot surrogate modeling of time-dependent parametric PDEs. *Project page*: https://enma-pde.github.io/

## 1   Introduction

Neural surrogates for spatio-temporal dynamics and PDE solving have emerged as efficient alternatives to traditional numerical solvers, driving rapid advances in the field. Early work (de Bezenac et al., 2018; Long et al., 2018; Raissi et al., 2019; Wiewel et al., 2019) laid the groundwork, followed by a wave of models leveraging diverse architectural designs (Yin et al., 2022b; Brandstetter et al., 2022; Wu et al., 2024a; Ayed et al., 2022). Neural operators (NOs) (Chen & Chen, 1995; Li et al., 2020a; Lu et al., 2021) expanded the paradigm by learning mappings between infinite-dimensional function spaces. Recent models have widely adopted this framework, improving scalability and flexibility (Gupta et al., 2021; Hao et al., 2023; Alkin et al., 2024a; Serrano et al., 2024b; Wu et al., 2024a). While earlier approaches focused on fixed PDE instances, current research tackles the more general task of learning *parametric PDEs* (Kirchmeyer et al., 2022; Nzoyem et al., 2025; Koupaï et al., 2024), and is now moving toward foundation models capable of handling multi-physics regimes (Subramanian et al., 2023; McCabe et al., 2023; Liu et al., 2025; Cao et al., 2024; Morel & Oyallon, 2025).

Most neural PDE solvers to date have focused on learning deterministic mappings, limiting their ability to capture complex or uncertain physical behaviors. This has spurred interest in stochastic modeling through generative probabilistic methods. A primary motivation arises from chaotic systems like weather forecasting (Price et al., 2025; Couairon et al., 2024) and turbulent flows (Kohl et al., 2024). Another challenge is error accumulation in autoregressive models which hampers long-term predictions and could be mitigated through probabilistic forecasters (Lippe et al., 2023;

---

[*]Equal contribution. Correspondence: armand.kassai[at]isir.upmc.fr, lise.leboudec[at]isir.upmc.fr

39th Conference on Neural Information Processing Systems (NeurIPS 2025).

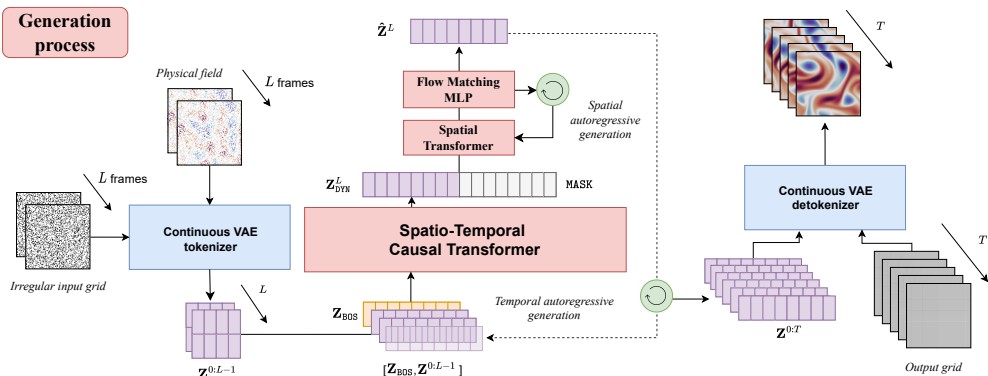

Figure 1: **ENMA: Continuous Spatio-Temporal Autoregressive Generation.** Given an initial sequence of latent states $\boldsymbol{Z}^{0:L-1}$, a causal transformer predicts the next latent frame $\boldsymbol{Z}_{\text{DYN}}^{L}$ using block-wise attention for temporal autoregression (Sec.3.1.1). This state is concatenated with masked tokens and passed to a spatial transformer, which performs masked spatial autoregression by progressively decoding tokens across multiple steps. A lightweight MLP trained with flow matching produces per-token predictions conditioned on spatial transformer predictions (Sec.3.1.2). The completed frame $\hat{\boldsymbol{Z}}^{L}$ is appended to the history for rollout. The full latent sequence is then decoded into the physical domain using a continuous VAE decoder (Sec. 3.2).

Kohl et al., 2024). More generally, data-driven surrogates must handle both aleatoric and epistemic uncertainty (Bülte et al., 2025; Wu et al., 2024b), often under conditions of partial observability, sensor noise, or coarse space-time resolution. Additionally, they face distribution shifts at test time, where PDE parameters may deviate from training data. Epistemic uncertainty may also stem from model mismatch or limited training data, reducing generalization to out-of-distribution regimes (Mouli et al., 2024).

Generative neural surrogates for time-dependent PDEs increasingly draw on advances from computer vision and, more recently, language modeling. These approaches typically fall into two categories. The first includes *diffusion models*, which operate in continuous spaces—pixel-level or latent—and model the joint distribution of future states by reversing a noising process. Based on U-Nets or transformer encoders (Ho et al., 2020; Peebles & Xie, 2022), they have been extended to PDE forecasting via iterative denoising (Zhou et al., 2025; Lippe et al., 2023; Kohl et al., 2024; Li et al., 2025). The second family follows an *autoregressive* (AR) paradigm, predicting per-token conditional distributions (Touvron et al., 2023). These models discretize physical fields using vector quantization (Oord et al., 2017; Mentzer et al., 2024), and generate sequences either sequentially (Yu et al., 2024) or via masked decoding (He et al., 2022; Yu et al., 2023), with recent adaptations for PDE surrogates and in-context learning (Serrano et al., 2025). Each approach comes with trade-offs. Diffusion models are robust to distribution drift and support uncertainty quantification via noise injection, but require expensive training and inference (Leng et al., 2025). AR models offer efficient, causality-aligned generation via KV caching and support in-context learning (Brown et al., 2020), but suffer from reduced expressiveness due to discretization and often yield uncalibrated uncertainty estimates (Jiang et al., 2020). They are also more sensitive to error accumulation from teacher forcing. Recently, continuous-token AR models have emerged as a promising alternative to discrete decoding in vision (Li et al., 2025), motivating their application to scientific modeling. We address fundamental limitations in existing generative surrogate models for PDEs—particularly their reliance on discrete tokenization or full-frame diffusion, which hampers scalability, uncertainty modeling, and physical consistency. We introduce **ENMA** (presented in Figure 1), a continuous autoregressive neural operator for modeling *time-dependent parametric PDEs*, where parameters such as initial conditions, coefficients, and forcing terms may vary across instances. ENMA operates entirely in a continuous latent space and advances both the encoder-decoder pipeline and the generative modeling component crucial to neural PDE solvers. The encoder employs attention mechanisms to process irregular spatio-temporal inputs (i.e., unordered point sets) and maps them onto a structured, grid-aligned latent space. A causal spatio-temporal convolutional encoder (Yu et al., 2024) then compresses these observations into compact latent tokens spanning multiple states.

Generation proceeds in two stages. A causal transformer first predicts future latent states autoregressively. Then, a masked spatial transformer decodes each state at the token level using Flow

Matching ([Lipman et al., 2023](#), [2024](#)) to model per-token conditional distributions in continuous space—providing a more efficient alternative to full-frame diffusion ([Lippe et al., 2023](#); [Kohl et al., 2024](#); [Li et al., 2025](#)). Finally, the decoder reconstructs the full physical trajectory from the generated latents. ENMA improves state-of-the-art generative surrogates by combining flexible per-token generation with expressive and compact latent modeling. It extends language-inspired AR models ([He et al., 2022](#); [Serrano et al., 2025](#)) to continuous tokens, avoiding the limitations of quantization. Compared to diffusion models, it offers lower computational cost by sampling tokens via a lightweight MLP instead of full-frame denoising. At inference, ENMA supports uncertainty quantification and in-context adaptation—conditioning on either past target states or auxiliary trajectories governed by similar dynamics. Our contributions are as follows:

- We introduce **ENMA**, the first neural operator to perform autoregressive generation over continuous latent tokens for physical systems, enabling accurate and scalable modeling of parametric PDEs while avoiding the limitations of discrete quantization.
- Using a masked spatial transformer trained with a Flow Matching objective to model per-token conditional distributions, ENMA offers a principled and efficient alternative to full-frame diffusion models for generation.
- ENMA supports probabilistic forecasting via tokenwise sampling and adapts to novel PDE regimes at inference time through temporal or trajectory-based conditioning—without retraining.
- To handle irregularly sampled inputs and support multi-state tokenization, ENMA leverages attention-based encoding combined with causal temporal convolutions.

## 2 Problem Setting

We consider time-dependent parametric partial differential equations defined over a spatial domain $\Omega \subset \mathbb{R}^d$ and a time interval $[0, T]$. Each instance is characterized by an initial condition $\boldsymbol{u}^0 \in L^2(\Omega, \mathbb{R}^{d_u})$ and a set of parameters $\gamma = (\boldsymbol{b}, \boldsymbol{f}, \boldsymbol{c})$, which include boundary conditions $\boldsymbol{b} \in L^2(\partial\Omega \times [0, T], \mathbb{R}^{d_b})$, a forcing term $\boldsymbol{f} \in L^2(\Omega \times [0, T], \mathbb{R}^{d_f})$, and PDE coefficients $\boldsymbol{c}$. The system of equations is:

$$\mathcal{N}[\boldsymbol{u}; \boldsymbol{c}, \boldsymbol{f}](x, t) = 0, \qquad \text{for } (x, t) \in \Omega \times (0, T], \qquad (1)$$

$$\mathcal{B}[\boldsymbol{u}; \boldsymbol{b}](x, t) = 0, \qquad \text{for } (x, t) \in \partial\Omega \times [0, T], \qquad (2)$$

$$\boldsymbol{u}(x, 0) = \boldsymbol{u}^0(x), \qquad \text{for } x \in \Omega, \qquad (3)$$

where $\mathcal{N}$ denotes a (potentially nonlinear) differential operator, and $\mathcal{B}$ encodes the boundary conditions. From an operator learning point of view, the objective is to approximate with a neural network $\widehat{\mathcal{G}}$ the temporal evolution operator $\mathcal{G}_\gamma$: $\boldsymbol{u}_i^{t+\Delta t} = \mathcal{G}_{@\gamma_i}(\boldsymbol{u}_i^t)$.

For training, we assume access to a dataset of $N$ solution trajectories, $\{\boldsymbol{u}_i\}_{i=1}^N$, where each trajectory is characterized by an initial condition $\boldsymbol{u}_i^0 \sim \nu_{u^0}$ and environment-specific parameters $\gamma_i \sim \nu_\gamma$, and is observed over a spatial grid $\mathcal{X}_i$ and on a temporal horizon $[0, T]$. Since the parameters $\gamma_i$ are unobserved, we supply the neural network with additional information beyond the current state $\boldsymbol{u}_i^t$ to help resolve this ambiguity. Specifically, we consider two predictive settings:

(i) **Temporal conditioning.** The model $\widehat{\mathcal{G}}$ observes an initial trajectory segment of $L$ states $\boldsymbol{u}^{0:L-1}$ and must autoregressively forecast future states up until $T$:

$$\boldsymbol{u}^L = \widehat{\mathcal{G}}(\boldsymbol{u}^{0:L-1}), \quad \boldsymbol{u}^{L+1} = \widehat{\mathcal{G}}(\boldsymbol{u}^{0:L}), \quad \dots$$

(ii) **Generalization from a context trajectory.** In this setting closer to the one of classical numerical solvers, the model observes only the initial state $\boldsymbol{u}^0$ of the target trajectory, along with a separate *context trajectory* $\boldsymbol{u}_{\text{context}}^{0:L}$ governed by the same $\gamma$, but from a different initial condition. The model uses this context to forecast the evolution from $\boldsymbol{u}^0$:

$$\boldsymbol{u}^1 = \widehat{\mathcal{G}}(\boldsymbol{u}^0; \boldsymbol{u}_{\text{context}}^{0:T}), \quad \boldsymbol{u}^2 = \widehat{\mathcal{G}}(\boldsymbol{u}^{0:1}; \boldsymbol{u}_{\text{context}}^{0:T}), \quad \dots$$

In these two settings, the surrogate model must emulate the underlying dynamics from data to unroll the target trajectory. These settings highlight the challenges of forecasting time-dependent systems under partial observability and unobserved parameters.

# 3 ENMA

We introduce **ENMA**, a neural operator tailored for *continuous tokenwise* autoregressive generation of spatio-temporal dynamics. ENMA follows an *encode–generate–decode* pipeline to approximate the solution operator $\mathcal{G}$, and can be decomposed in the three corresponding steps $\mathcal{G} \approx \widehat{\mathcal{G}} = \mathcal{D}_\psi \circ \mathcal{P}_\theta \circ \mathcal{E}_\omega$,. The encoder $\mathcal{E}_\omega$ maps irregularly sampled spatio-temporal inputs $\boldsymbol{u}^{0:L-1} \in \mathbb{R}^{|\mathcal{X}| \times L \times c}$—observed at $|\mathcal{X}|$ spatial locations over $L$ time steps with $c$ physical channels—into a structured latent representation $\boldsymbol{Z}^{0:L-1} \in \mathbb{R}^{M \times L \times d}$. The generative model $\mathcal{P}_\theta$ then autoregressively predicts future latent states in a tokenwise fashion, and the decoder $\mathcal{D}_\psi$ maps the generated latents back into the physical domain. The full process for a one-step prediction can be summarized as:

$$\boldsymbol{u}^{0:L-1} \in \mathbb{R}^{|\mathcal{X}| \times L \times c} \xrightarrow{\text{encode}} \boldsymbol{Z}^{0:L-1} \in \mathbb{R}^{M \times L \times d} \xrightarrow{\text{generate}} \hat{\boldsymbol{Z}}^L \in \mathbb{R}^{M \times d} \xrightarrow{\text{decode}} \hat{\boldsymbol{u}}^L \in \mathbb{R}^{|\mathcal{X}| \times c} \quad (4)$$

Here, $M$ is the number of spatial latent tokens per state $Z$ and $d$ the latent embedding dimension. The generative model can be repeatedly applied to roll out predictions autoregressively over a horizon of $T$ steps. We describe the generative model in Section 3.1, and the encoder–decoder in Section 3.2. For simplicity, we describe the model for the *temporal conditioning* setting, the adaptation for the *generalisation from context* setting is immediate.

## 3.1 Tokenwise Autoregressive Generation

The core contribution of ENMA lies in its continuous autoregressive architecture, designed for modeling spatiotemporal dynamics. The generative model proceeds in two stages (5): it first extracts a spatio-temporal representation $\boldsymbol{Z}_{\text{DYN}}^L$ from the encoded trajectory $\boldsymbol{Z}^{0:L-1}$ using a causal transformer, and then predicts the spatial distribution of the next time step via a tokenwise decoding mechanism. This distribution is estimated with a flow-matching component, with the help of a spatial transformer and a lightweight MLP, inspired by recent autoregressive models such as MAR (Li et al., 2024).

$$\boldsymbol{Z}^{0:L-1} \in \mathbb{R}^{M \times L \times d} \xrightarrow{\text{Causal Transformer}} \boldsymbol{Z}_{\text{DYN}}^L \in \mathbb{R}^{M \times d} \xrightarrow{\text{AR Spatial Generation}} \hat{\boldsymbol{Z}}^L \in \mathbb{R}^{M \times d}. \quad (5)$$

This two-stage design enables ENMA to capture both long-range temporal dependencies through the causal transformer and fine-grained spatial structure with the generative spatial decoder, in a scalable, autoregressive manner. This choice is further motivated by the distinction between temporal and spatial dimensions: temporal prediction naturally benefits from the sequential ordering of the trajectory, whereas there is no such ordering for spatial generation. We provide architectural and training details for each stage, respectively, in Sections 3.1.1 and 3.1.2, following the temporal conditioning setup introduced in Section 2.

### 3.1.1 Causal Transformer

The causal transformer is designed to extract spatio-temporal representations of a trajectory while enabling scalable training. At each time step $i$, all tokens within the current state $\boldsymbol{Z}^i$ can attend to one another, as well as to all tokens from preceding states $\boldsymbol{Z}^j$ for $j < i$. Formally, given a sequence of latent states $\boldsymbol{Z}^{0:L-1} = (\boldsymbol{Z}^0, \ldots, \boldsymbol{Z}^{L-1})$, the transformer produces the dynamic context representation for time step $t$: $\boldsymbol{Z}_{\text{DYN}}^L = \text{CausalTransformer}(\boldsymbol{Z}_{\text{BOS}}, \boldsymbol{Z}^{0:L-1})$, where $\boldsymbol{Z}_{\text{BOS}}$ is a learned begin-of-sequence (BOS) token. $\boldsymbol{Z}_{\text{DYN}}^L$ can be seen as a latent context capturing *the dynamics*, from the observed time-steps $[0, L-1]$, to predict the next step $L$.

To support parallel training and efficient inference, this factorization is implemented using a block-wise causal attention mask (see Appendix D). The model's causal structure also enables key-value caching at inference, allowing reuse of past computations and facilitating fast autoregressive rollout.

During training, we apply teacher forcing—conditioning the model on the full sequence of ground-truth latent states. At inference, the causal Transformer can accept arbitrary sized sequence inputs $\boldsymbol{Z}^{0:L-1}$ for $L \in [0, T]$. $\boldsymbol{Z}_{\text{DYN}}^L$ is then used as a context for the spatial transformer described in 3.1.2.

### 3.1.2 Masked Autoregressive Generation

To perform tokenwise continuous autoregression, ENMA employs a masked decoding scheme conditioned on the context $\boldsymbol{Z}_{\text{DYN}}^L$. This is implemented with a spatial transformer that produces

conditioning intermediate representations $\tilde{\boldsymbol{Z}}^L = (\tilde{z}_1^L, \ldots, \tilde{z}_M^L)$—which capture spatial correlations and temporal context for each token—combined with a lightweight MLP that models the per-token output distribution $p(\hat{z}_i^L \mid \tilde{z}_i^L)), i = 1, \ldots, M$. We adopt the *masked autoregressive (MAR)* strategy (Li et al., 2024), which enables permutation-invariant autoregressive generation over the spatial domain. Training and inference pipelines for spatial generation are illustrated in Figure 2.

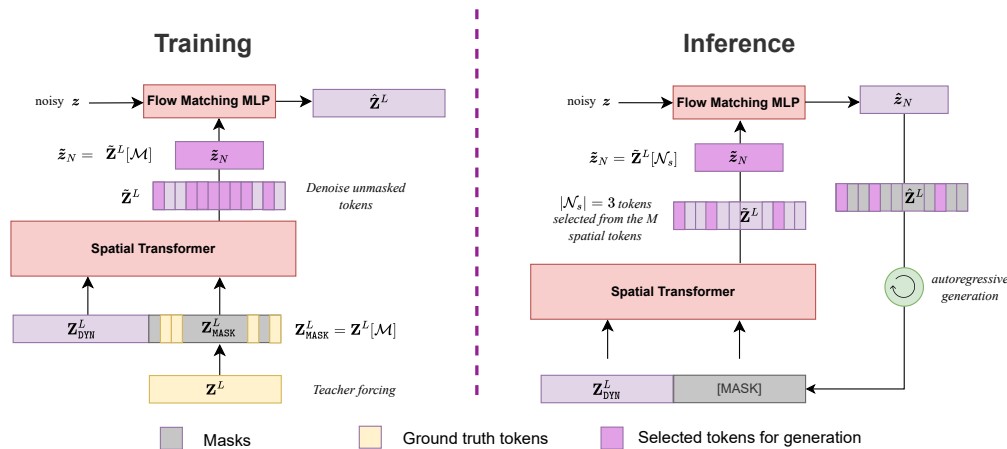

Figure 2: **Training and inference for the spatial transformer**. For training, a random subset of tokens is masked and decoded in a single step. At inference, generation proceeds iteratively: starting from a fully masked state $\boldsymbol{Z}^{L,0}$, a subset $\mathcal{N}_s$ of tokens is selected at each step to be generated by the MLP, conditioned on spatial transformer outputs. This process is repeated $S$ steps to produce $\hat{\boldsymbol{Z}}^L$.

**Spatial Transformer.** During *training*, a random subset of token indices $\mathcal{M} \subset \{1, \ldots, M\}$ is selected to be masked in the ground-truth latent frame $\boldsymbol{Z}^L$, with the masking ratio sampled uniformly between 75% and 100%. These tokens are replaced with a learned [MASK] embedding to form the partially masked input $\boldsymbol{Z}_{\text{MASK}}^L$. The spatial transformer processes the concatenated pair $(\boldsymbol{Z}_{\text{DYN}}^L, \boldsymbol{Z}_{\text{MASK}}^L)$ and outputs contextual representations $\tilde{\boldsymbol{Z}}^L$. The tokens $\tilde{z}_{\mathcal{M}}^L = \tilde{\boldsymbol{Z}}^L[\mathcal{M}]$ are then used independently to condition a lightweight MLP trained with a Flow Matching objective. For each $i \in \mathcal{M}$, the MLP maps a noisy sample—conditioned on $\tilde{z}_i^L$—into a denoised latent prediction $\hat{z}_i^L$. The loss is computed only over the masked positions $\mathcal{M}$.

At *inference*, tokenwise autoregression proceeds over $S$ autoregressive steps to predict $\hat{\boldsymbol{Z}}^L$. At step $s = 0$, $\hat{\boldsymbol{Z}}^{L,0}$ is initialized with the mask embedding, i.e. $\hat{z}_i^{L,0} = $ [MASK] for $i \in \{1, \ldots, M\}$. At each step, a subset $\mathcal{N}_s \subset \{i \in \{1, \ldots, M\} \mid \hat{z}_i^{L,s} = $ [MASK]$\}$ of the masked positions (not previously selected if $s > 0$) are selected for generation. The spatial transformer processes $(\boldsymbol{Z}_{\text{DYN}}^L, \hat{\boldsymbol{Z}}^{L,s})$ and predicts $\tilde{\boldsymbol{Z}}^{L,s}$; the selected positions to be generated $\tilde{z}_i^{L,s}$ are used to condition the flow matching MLP, generating the selected tokens $\hat{z}_i^{L,s}$. These tokens are inserted back into $\hat{\boldsymbol{Z}}^{L,s}$ to form $\hat{\boldsymbol{Z}}^{L,s+1}$, and the process continues until all positions are generated at step $S$, yielding $\hat{\boldsymbol{Z}}^L = \hat{\boldsymbol{Z}}^{L,S}$. Multiple tokens can be generated at each step. The number of decoded tokens per step follows a cosine schedule: $|\mathcal{N}_s| = \lfloor M \cdot \cos^2\left(\frac{\pi s}{2S}\right)\rfloor$. Compared to fully parallel decoding, this approach provides fine-grained control over uncertainty and sample diversity, while enabling efficient generation through vectorized MLP sampling.

**Flow Matching for Per-Token Prediction**  To model per-token conditional distributions in continuous latent space, ENMA employs *Flow Matching (FM)*.

Let $z_i^L \in \mathbb{R}^d$ denote the ground-truth latent token at position $i$ in the frame $\boldsymbol{Z}^L = (z_1^L, \ldots, z_M^L)$, and let $\tilde{z}_i^L$ be its contextual embedding produced by the spatial transformer. We drop the superscripts for clarity here. We seek to learn the conditional distribution $p(z \mid \tilde{z})$ via a flow-based transport from a base distribution $p_0 = \mathcal{N}(0, I)$ to the target $p_1 = p(z \mid \tilde{z})$. This is parameterized as an ordinary differential equation (ODE):

$$\frac{d\mathbf{z}^r}{dr} = v(\mathbf{z}^r, r), \tag{6}$$

where $r \in [0, 1]$ is a denoising index for the flow matching, and $v$ is a velocity field implemented by a neural network. During training, we sample intermediate points along the probability path using:

$$\mathbf{z}^r = r\mathbf{z} + [1 - r]\,\boldsymbol{\epsilon}, \quad \boldsymbol{\epsilon} \sim \mathcal{N}(0, I), \tag{7}$$

We train a lightweight MLP, conditioned on the context $\tilde{\mathbf{z}}$, to approximate the velocity field. It takes as input $(\mathbf{z}^r, \tilde{\mathbf{z}}, r)$ and predicts the transport direction. The flow matching training objective is:

$$\mathcal{L}_{\text{FM}} = \mathbb{E}_{\boldsymbol{\epsilon}, r}\left[\left\|\text{MLP}(\mathbf{z}^r, \tilde{\mathbf{z}}, r) - \mathbf{z} + \boldsymbol{\epsilon}\right\|_2^2\right], \tag{8}$$

which encourages the model to match the velocity that would move $\mathbf{z}^r$ along the interpolating path toward $\mathbf{z}$. At inference, token generation is performed by solving the ODE:

$$\hat{\mathbf{z}} = \text{ODESOLVE}\left(\text{MLP}(\mathbf{z}^r, \tilde{\mathbf{z}}, r)\right), \quad \text{with } \mathbf{z}^0 \sim \mathcal{N}(0, I),$$

from $r = 0$ to $r = 1$ using the midpoint method. This enables efficient of continuous latent tokens conditioned on their spatial context, without requiring discrete quantization or full-frame denoising. We present the full inference method of ENMA for latent generation in the pseudo-code 1:

---

**Algorithm 1:** ENMA Inference: Autoregressive Latent Generation with Cosine Masked Decoding

---

**Input:** Encoded latents $\boldsymbol{Z}^{0:L-1}$ from observed inputs $\boldsymbol{u}_k^{0:L-1}$
**Output:** Predicted latents $\{\hat{\boldsymbol{Z}}^L, \ldots, \hat{\boldsymbol{Z}}^T\}$
**for** $t = L$ **to** $T$ **do**
    $\boldsymbol{Z}_{\text{DYN}}^t \leftarrow \text{CausalTransformer}(\boldsymbol{Z}^{0:t-1})$;
    $\hat{\boldsymbol{Z}}^t \leftarrow [\texttt{MASK}]$;
    **for** $s = 1$ **to** $S$ **do**
        $\tilde{\boldsymbol{Z}}^t \leftarrow \text{SpatialTransformer}([\boldsymbol{Z}_{\text{DYN}}^t; \hat{\boldsymbol{Z}}^t])$;
        $n_s \leftarrow \text{CosineSchedule}(s, S)$;
        Sample $n_s$ masked indices: $\mathcal{N}_s \subset \{i \mid \hat{z}_i^t = [\texttt{MASK}]\}$;
        $\boldsymbol{\epsilon} \sim \mathcal{N}(0, \mathbf{I})^{n_s \times d}$;
        $\hat{\boldsymbol{Z}}^t[\mathcal{N}_s] \leftarrow \text{ODESOLVE}(\text{MLP}, \tilde{\boldsymbol{Z}}^t[\mathcal{N}_s], \boldsymbol{\epsilon})$;
    $\boldsymbol{Z}^t \leftarrow \hat{\boldsymbol{Z}}^t$;

---

## 3.2 Auto-Encoding

Encoding irregular spatio-temporal data into a compact latent representation is a central component of ENMA. Inspired by recent advances in vision models (Yu et al., 2023), ENMA combines two key elements. $(i)$ a cross-attention module performs spatial interpolation, mapping the irregular inputs $\boldsymbol{u}^{0:L-1}(\mathcal{X})$, defined over an arbitrary spatial grid $\mathcal{X}$ at $L$ time steps, onto a regular grid $\boldsymbol{\Xi}$, yielding intermediate representations $\boldsymbol{u}^{0:L-1}(\boldsymbol{\Xi})$. $(ii)$ a temporally causal CNN processes the intermediate sequence $\boldsymbol{u}^{0:L-1}(\boldsymbol{\Xi})$, across space and time into tokens $\boldsymbol{Z}^{0:L-1}$. Mapping irregular inputs onto a regular grid-aligned space, allows ENMA to leverage convolutional inductive biases of the CNN for efficient and coherent compression. Decoding mirrors the encoder structure. The encoder and decoder are optimized jointly with a VAE loss: $\mathcal{L} = \mathcal{L}_{\text{recon}} + \beta \cdot \mathcal{L}_{\text{KL}}$. To improve robustness to varying input sparsity, we randomly subsample the spatial grid at training time: the number of input points varies between 20% and 100% of the full grid $\mathcal{X}$. These two elements are described in more details below.

$(i)$ **Interpolation via Cross-Attention** For interpolation, we modify the cross-attention module from Serrano et al. (2024a) to favor spatial locality, by introducing a *geometry-aware attention bias*. This bias, exploits the geometry of the inputs and induces locality in the cross-attention operation, enabling the regular gridded interpolation $\boldsymbol{u}^{0:L-1}(\boldsymbol{\Xi})$ to capture the spatial structure of the physical field.

Formally, we define the cross-attention from queries located at $\mathcal{X}$ to keys and values located at $\boldsymbol{\Xi}$ as:

$$\text{Attention}(Q, K, V) = \text{Softmax}\left(\frac{QK^\top + B}{\sqrt{d_k}}\right)V, \quad B_{i,j} = -m \cdot \text{dist}(x_i - \xi_j), \text{ for } x_i \in \mathcal{X}, \xi_j \in \boldsymbol{\Xi}$$

where $Q = q(\mathcal{X})$, $K = k(\Xi)$, $V = v(\Xi)$ are query, key and value tokens defined on the respective spaces $\mathcal{X}$ and $\Xi$, $d_k$ denotes the key and query dimension, and $m$ is a scaling factor (either fixed or learned, as in ALiBi (Press et al., 2022)). Intuitively, larger distances between query and key positions induce stronger negative biases, reducing attention weights and promoting locality in the interpolation.

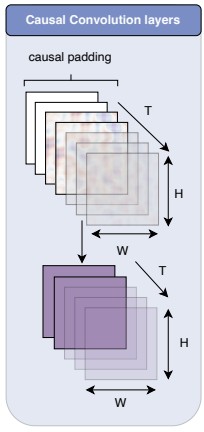

Figure 3: Causal in time convolutional layer.

($ii$) **Processing via Causal Convolution**  Following the interpolation modules, ENMA compresses latent trajectories using a causal 3D convolutional encoder, enabling efficient representation across space and time. Unlike prior work (e.g., Wang et al. (2024)), our use of causal convolutions supports variable-length inputs, allowing the model to process both temporally conditioned settings and initial value problems. For 2D spatial domains sequences, we use 3D convolutions with kernel size $(k_t, k_h, k_w)$ and a causal padding in the temporal dimension (Yu et al., 2023). This techniques only applies a $k_t - 1$ padding in the past of the temporal dimension. This design ensures that outputs at time $t$ depend only on inputs up to $t$, enabling inference from a single initial frame (see Figure 3). Spatial and temporal compression are applied independently, with the overall compression rate controlled by the number of stacked blocks—each reducing the corresponding dimension by a factor of 2. The overall architecture design is detailed in appendix D.

**Decoding**  Decoding mirrors the encoder structure. Transposed convolutions reverse the spatio-temporal compression, and a final cross-attention layer interpolates latent outputs to arbitrary physical grids. Here, the desired output coordinates serve as queries, while the latent tokens provide keys and values, enabling high-resolution reconstruction independent of the training grid.

## 4 Experiments

We conduct extensive experiments to evaluate ENMA. Section 4.2 assesses the encoder–decoder in terms of reconstruction error, time-stepping accuracy, and compression rate, comparing against standard neural operator baselines. Section 4.3 evaluates ENMA's generative forecasting ability for both temporal conditioning and Initial Value Problem with context trajectory. Dataset, training, and implementation details are provided respectively in Appendices C, E, and E. We also provide additional experiments and ablations in appendix F.

### 4.1 Datasets

We evaluate ENMA on five dynamical systems: two 1D and three 2D PDEs. Data is generated in batches where all trajectories share PDE parameters $\gamma$, but differ in initial conditions. The PDE description is provided in Appendix C. For each system, we generate 12,000 training and 1,200 test trajectories, using a batch size of 10. In 1D, we use the **Combined** equation (Brandstetter et al., 2022), where coefficients $(\alpha, \beta, \gamma)$ vary for the terms $-\frac{\partial u^2}{\partial x}$, $+\frac{\partial^2 u}{\partial x^2}$, and $-\frac{\partial^3 u}{\partial x^3}$; and the **Advection** equation, where both advection speed and initial condition vary. In 2D, we consider: the **Vorticity** equation with varying viscosity $\nu$; the **Wave** equation with varying wave speed $c$ and damping coefficient $k$; and the **Gray-Scott** system (Pearson, 1993), where the reaction parameters $F$ and $k$ vary across batches.

### 4.2 Encoder/ decoder quality evaluation

**Setting**  We evaluate the encoder-decoder quality across baselines that use latent-space representations — a key component of neural surrogates. Since prediction accuracy depends on the encoder–decoder pair, we first assess its limitations and show ENMA's improvements. We conduct two tests: (i) **reconstruction**, evaluating auto-encoding fidelity, and (ii) **time-stepping**, where a small fixed FNO, trained per encoder, performs rollouts in latent space. All models are trained with the same objective as ENMA, using input subsets varying between 20% and 100% of the grid $\mathcal{X}$. At test time, inputs use irregular grids with $\pi = 20\%, 50\%$, or full (100%) sampling; reconstruction is always evaluated on the full grid. See appendix E for more implementation details and experimental protocol.

**Baselines** We compare ENMA's encoder-decoder against representative neural operator architectures: two transformer-based models—**OFormer** (Li et al., 2023a) and **AROMA** (Serrano et al., 2024a); the INR-based **CORAL** (Serrano et al., 2023); the neural operator **GINO** (Li et al., 2023b). These baselines differ in compression: **OFormer** performs no compression (point wise), **AROMA**, **CORAL**, and **GINO** compress spatially, while **ENMA** compress both spatially and temporally. All baseline are compared with similar token dimension (e.g. $d = 4$ for 1d datasets and $d = 8$ for 2d datasets), but the compression rate can vary as some baselines do not compress in time and/or in space.

Table 1: **Reconstruction error** – Test results and compression rates. Metrics in Relative MSE. The compression rate reflects how much the latent representation is reduced compared to the input data. A compression rate of $\times 2$ indicates that the latent space contains half as many elements as the input.

| $\downarrow \mathcal{X}_{te}$ | Dataset → | Advection | | | Vorticity | | |
|---|---|---|---|---|---|---|---|
| | Model ↓ | *Reconstruction* | *Time-stepping* | *Compression rate* | *Reconstruction* | *Time-stepping* | *Compression rate* |
| $\pi = 100\%$ | OFormer | 1.70e-1 | 1.11e+0 | ×0.25 | 9.99e-1 | 1.00e+0 | ×0.125 |
| | GINO | 3.15e-1 | 8.55e-1 | ×2 | 5.63e-1 | 9.83e-1 | ×8 |
| | AROMA | 5.41e-3 | 2.23e-1 | ×2 | 1.45e-1 | 1.13e+0 | ×8 |
| | CORAL | 1.34e-2 | 9.64e-1 | ×2 | 4.63e-1 | 9.18e-1 | ×8 |
| | ENMA | **1.83e-3** | **1.64e-1** | **×4** | **9.20e-2** | **2.62e-1** | **×15** |
| $\pi = 50\%$ | OFormer | 1.79e-1 | 1.11e+0 | - | 9.99e-1 | 1.00e+0 | - |
| | GINO | 3.21e-1 | 8.64-1 | - | 5.69e-1 | 9.91e-1 | - |
| | AROMA | 2.34e-2 | 2.29e-1 | - | 1.64e-1 | 1.14e+0 | - |
| | CORAL | 7.57e-2 | 9.74e-1 | - | 4.95e-1 | 9.22e-1 | - |
| | ENMA | **4.60e-3** | **1.72e-1** | - | **9.90e-2** | **2.68e-1** | - |
| $\pi = 20\%$ | OFormer | 2.50e-1 | 1.13e+0 | - | 9.99e-1 | 1.00e+0 | - |
| | GINO | 3.54e-1 | 9.11e-1 | - | 5.90e-1 | 1.04e+0 | - |
| | AROMA | 1.67e-1 | 3.21e-1 | - | 2.29e-1 | 1.14e+0 | - |
| | CORAL | 4.77e-1 | 1.06e+0 | - | 6.89e-1 | 9.37e-1 | - |
| | ENMA | **3.05e-2** | **3.13e-1** | - | **1.37e-1** | **3.11e-1** | - |

**Results** Table 1 presents the reconstruction and time-stepping errors for ENMA and baseline methods. These results highlight the robustness of ENMA across different grid sizes and datasets, in both 1D and 2D settings. While most baselines struggle as the grid becomes sparser, ENMA's encoding–decoding strategy consistently provides informative representations of the trajectories, even under limited observations. When the full grid is available (i.e., $\pi = 100\%$), ENMA outperforms the baselines, reducing the reconstruction error by up to a factor of $3\times$. As the observation rate drops to 50% and 20%, the performance gap widens even further. Notably, at $\pi = 20\%$, ENMA maintains strong performance, while other methods degrade significantly. This demonstrates ENMA's ability to generalize and infer latent dynamics under sparse supervision. For the time-stepping task, the higher-quality tokens produced by ENMA significantly improve forecasting accuracy. We also emphasize that ENMA compresses the temporal dimension more effectively than baselines: even with fewer tokens ($\times 4$ compression in 1d and $\times 15$ in 2d), it remains competitive. This improvement is attributed to ENMA's ability to leverage causal structure during encoding, which leads to more informative and temporally coherent representations. Additional analysis about the ENMA's encoder/decoder architecture are provided in appendix F.

### 4.3 Dynamics forecasting

**Setting** We evaluate our model on the standard *dynamics forecasting* task, which predicts the future evolution of spatio-temporal systems. We consider two settings: (i) **temporal conditioning**, where the model observes the first $L$ time steps and predicts up to horizon $T$; and (ii) **initial value problem with context**, where only the target's initial state $u^0$ is given, along with an auxiliary trajectory governed by the same parameters $\gamma$ but from a different initialization. The model must infer dynamics from this context and forecast from $u^0$. We evaluate both settings under **in-distribution (In-D)** and **out-of-distribution (Out-D)** regimes, where the latter involves PDE parameters not seen during training. Each Out-D evaluation uses 120 held-out trajectories. See Appendix C for details.

**Baselines** We evaluate ENMA against a diverse set of baselines, both deterministic and generative. For **temporal conditioning**, the deterministic models include the transformer-based solvers:

BCAT (Liu et al., 2025) and AViT (McCabe et al., 2023) both designed for multi-physics forecasting, and the Fourier Neural Operator (FNO), a classical reference. For the generative models we consider a continuous autoregressive diffusion transformer AR-DiT (Kohl et al., 2024), and Zebra (Serrano et al., 2025), a language-model-style decoder operating in a quantized latent space. For the **initial value problem with context**, we compare with the deterministic In-Context ViT that concatenates the context trajectory to the target initial state, and with a `[CLS]` ViT variant that uses a learned token to summarize the PDE parameters $\gamma$. For the generative baselines, we include Zebra (Serrano et al., 2025).

Table 2: Comparison of model performance for temporal conditioning and initial value problem tasks across 5 dynamical systems. Metrics in Relative MSE. Lower is better.

| Setting ↓ | Dataset → | Advection | | Combined | | Gray-Scott | | Wave | | Vorticity | |
|---|---|---|---|---|---|---|---|---|---|---|---|
| | Model ↓ | *In-D* | *Out-D* | *In-D* | *Out-D* | *In-D* | *Out-D* | *In-D* | *Out-D* | *In-D* | *Out-D* |
| Temporal Conditioning | FNO | 2.47e-1 | 7.95e-1 | 1.33e-1 | 2.66e+1 | 5.04e-2 | 1.92e-1 | 6.91e-1 | 2.64e+0 | 6.07e-2 | **2.15e-1** |
| | BCAT | 5.55e-1 | 9.23e-1 | 2.68e-1 | 9.28e-1 | 3.74e-2 | 1.57e-1 | 2.19e-1 | 5.38e-1 | 5.39e-2 | 3.00e-1 |
| | AVIT | 1.64e-1 | **5.02e-1** | 5.67e-2 | 3.05e-1 | 4.26e-2 | 1.68e-1 | 1.57e-1 | 5.88e-1 | 1.76e-1 | 3.77e-1 |
| | AR-DiT | 2.36e-1 | 8.56e-1 | 2.95e-1 | 1.80e+0 | 3.69e-1 | 4.99e-1 | 1.12e+0 | 7.52e+0 | 1.98e-1 | 4.80e-1 |
| | Zebra | 2.04e-1 | 1.39e+0 | 1.82e-2 | 2.20e+0 | 4.21e-2 | 1.82e-1 | **1.40e-1** | **3.15e-1** | **4.43e-2** | 2.23e-1 |
| | ENMA | **3.95e-2** | 5.30e-1 | **7.86e-3** | **1.02e-1** | **3.40e-2** | **1.44e-1** | 1.45e-1 | 4.89e-1 | 7.58e-2 | 3.45e-1 |
| Initial Value Problem | In-Context ViT | 1.15e+0 | 1.20e+0 | 5.79e-1 | 1.36e+0 | 6.90e-2 | 1.94e-1 | 1.72e-1 | 6.24e-1 | 1.53e-1 | 3.92e-1 |
| | `[CLS]` ViT | 1.15e+0 | 1.36e+0 | 9.60e-2 | 1.16e+0 | 4.80e-2 | 2.19e-1 | 5.56e-1 | 1.02e+0 | **4.30e-2** | 2.59e-1 |
| | Zebra | 3.16e-1 | 1.47e+0 | 4.78e-2 | 9.63e-1 | **4.40e-2** | **1.22e-1** | 1.69e-1 | **3.52e-1** | 5.90e-2 | **2.29e-1** |
| | ENMA | **2.02e-1** | **8.07e-1** | **1.56e-2** | **3.30e-1** | 4.80e-2 | 1.34e-1 | **1.54e-1** | 5.02e-1 | 8.58e-2 | 3.20e-1 |

**Results** Table 2 summarizes performance across five PDE benchmarks under both temporal conditioning and IVP settings. ENMA achieves state-of-the-art results on most tasks, outperforming both deterministic solvers (FNO, BCAT, AViT) and generative baselines (AR-DiT and Zebra). Unlike most methods that operate in physical space, ENMA performs autoregressive generation entirely in a continuous latent space (time and space compression), which reduces the computational complexity, but represents a much more challenging setting. Only Zebra shares this property (compression in space only) but relies on quantized tokens, which favor low-frequency reconstruction but can limit expressiveness. ENMA shows strong performance on Advection, Combined, and Gray-Scott, and remains competitive on Wave, despite the added challenge of temporal compression. Performance on Vorticity is slightly lower than the best competitor, likely due to the combined effect of temporal and spatial compression, which makes accurate reconstruction more challenging. We analyze this aspect in appendix F.1.2, where we obsreve that reducing the compression ratio naturally lead to better prediction performance. ENMA's continuous modeling allows finer control over generation and uncertainty estimation. This aspect is explored in appendix F.1.1.

## 4.4 Evaluation on High-Dimensional Physics Systems

We evaluate **ENMA** on standard public benchmarks (Rayleigh–Bénard and Active Matter, (Ohana et al., 2024)). These datasets feature highly nonlinear spatio-temporal dynamics, multiple interacting physical fields, and dense spatial grids, making them representative of complex, high-dimensional physical systems. We compare against the same competitive deterministic baselines used in section 4.3.

Table 3: Temporal Conditioning setting on complex physical systems (Relative MSE ↓). Compression ratios are reported per dataset.

| Model | Rayleigh–Bénard | | Active Matter | |
|---|---|---|---|---|
| | *Temporal Conditioning* ↓ | *Comp.* | *Time-stepping* ↓ | *Comp.* |
| BCAT | 1.06e-1 | ×1 | 4.56e-1 | ×1 |
| AVIT | 1.01e-1 | ×1 | 4.62e-1 | ×1 |
| ENMA (ours) | **9.87e-2** | ×64 | **3.33e-1** | ×176 |

**Results** ENMA attains the lowest error on both systems while operating at high compression (e.g., ×64 on Rayleigh–Bénard and ×176 on Active Matter), demonstrating that accurate time-stepping

is achievable even under aggressive latent compaction. Deterministic baselines (BCAT, AVIT) are faster but lack compression and yield higher errors.

## 4.5 Generative Capabilities of ENMA

**ENMA** is a generative neural operator capable of producing stochastic and physically consistent trajectories. We highlight two core experiments that demonstrate its generative ability; extended analyses and qualitative visualizations are provided in appendix F.1.1.

**Uncertainty quantification.** ENMA performs uncertainty estimation by sampling multiple trajectories from its continuous latent space through flow matching. Unlike discrete autoregressive baselines such as Zebra, which rely on categorical sampling, ENMA's continuous formulation yields sharper and better-calibrated probabilistic forecasts. As shown in table 4, ENMA achieves the lowest calibration (RMSCE) and probabilistic (CRPS) errors, confirming its ability to produce both reliable and diverse uncertainty estimates.

Table 4: Uncertainty metrics ($\downarrow$ is better) on the Combined dataset.

| Model | RMSCE $\downarrow$ | CRPS $\downarrow$ |
|---|---|---|
| AR-DiT | $2.68\times10^{-1}$ | $1.27\times10^{-2}$ |
| Zebra | $2.19\times10^{-1}$ | $9.00\times10^{-3}$ |
| ENMA (ours) | $8.68\times10^{-2}$ | $1.70\times10^{-3}$ |

**Data generation.** We further assess ENMA's ability to generate full trajectories *without conditioning on the initial state or PDE parameters*. Given only a context trajectory, ENMA infers the latent physics and synthesizes coherent spatio-temporal fields. In table 5, ENMA achieves the lowest Physics Fréchet Distance (FPD) and highest Precision, indicating superior fidelity and physical consistency. Zebra attains slightly higher Recall, reflecting greater diversity but lower sample quality.

Table 5: Generative metrics on the Combined dataset. Lower FPD and higher Precision/Recall indicate better quality and diversity.

| Model | FPD $\downarrow$ | Precision $\uparrow$ | Recall $\uparrow$ |
|---|---|---|---|
| Zebra | $1.03\times10^{-1}$ | 0.77 | **0.86** |
| ENMA (ours) | $9.50\times10^{-3}$ | **0.79** | 0.78 |

## 5 Conclusion

**Discussion** We presented **ENMA**, a generative model leveraging a continuous latent-space autoregressive neural operator for modeling time-dependent parametric PDEs. ENMA performs generation directly in a compact latent space, at the token level, using masked autoregression and a flow matching objective, enabling efficient forecasting and improves over alternative generative models based either on full-frame diffusion or on discrete quantization. Across 5 dynamical systems, experiments demonstrate that ENMA is a strong neural surrogate that competes with neural PDE solvers operating in the physical space, in both temporal conditioning and initial value problem settings. Reconstruction and time-stepping evaluations confirm the effectiveness of its encoder–decoder design, showing improved latent modeling and robustness to unordered point sets compared to existing neural operator baselines.

**Limitations** ENMA's performance is slightly reduced on vorticity, where latent compression can hinder the accurate recovery of fine-scale features. This highlights a central trade-off in latent-space surrogates: while compression enables scalability, it may limit expressiveness in certain regimes. Regarding computation efficiency, ENMA is computationally efficient compared to full-frame diffusion or next-token models. However, ENMA's model's cost increases with respect to the number of latent tokens per state. This motivates future work on adaptive generation strategies—for instance, using a coarse-to-fine decoding scheme where a token directly synthesizes the frame, and remaining tokens act as refinements (Bachmann et al., 2025).

**Broader impact** PDE solvers are key to applications in weather, climate, medicine, aerodynamics, and defense. While ENMA is not deployed in such settings, it offers fast, uncertainty-aware surrogates adaptable to new regimes.

**Acknowledgments**

We acknowledge the financial support provided by DL4CLIM (ANR-19-CHIA-0018-01), DEEPNUM (ANR-21-CE23-0017-02), PHLUSIM (ANR-23-CE23-0025-02), and PEPR Sharp (ANR-23-PEIA-0008", "ANR", "FRANCE 2030"). This project was provided with computer and storage resources by GENCI at IDRIS thanks to the grants 2025-AD011016890R1, 2025-AD011014938R1 and 2025-AD011015511R1 on the supercomputer Jean Zay's A100/H100 partitions.

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

# Appendix Table of Contents

# A   Notation

We summarize the main notations used throughout the paper:

| Symbol | Description |
| --- | --- |
| $x$ | Spatial coordinate |
| $t$ | Temporal coordinate |
| $u(x,t)$ | Solution of the PDE at $(x,t)$ |
| $T$ | Final time |
| $\pi$ | Observation ratio (proportion of available inputs) |
| $\mathcal{X}$ | Input spatial grid |
| $\mathcal{X}_{\text{te}}$ | Test-time input spatial grid |
| $\gamma$ | PDE parameters |
| $\mathcal{N}$ | Differential operator |
| $\mathcal{B}$ | Boundary condition operator |
| $\mathcal{G}$ | True temporal evolution operator |
| $\widehat{\mathcal{G}}$ | Neural approximation of the evolution operator |
| $\boldsymbol{Z}$ | True latent state from VAE tokenizer |
| $M$ | Number of spatial tokens in $\boldsymbol{Z}$ |
| $\boldsymbol{z}$ | Spatial token, with $\boldsymbol{Z} = (\boldsymbol{z}_1, \ldots, \boldsymbol{z}_M)$ |
| $L$ | Number of observed time steps (history) |
| $\boldsymbol{Z}^{0:L-1}$ | Latent sequence of past states |
| $\boldsymbol{Z}_{\text{DYN}}$ | Predicted latent state from the Causal Transformer |
| $\tilde{\boldsymbol{Z}}$ | Contextualized latent tokens from Spatial Transformer |
| $\tilde{\boldsymbol{z}}$ | Spatial token in $\tilde{\boldsymbol{Z}} = (\tilde{\boldsymbol{z}}_1, \ldots, \tilde{\boldsymbol{z}}_M)$ |
| $\hat{\boldsymbol{Z}}$ | Predicted latent state |
| $\hat{\boldsymbol{z}}$ | Spatial token in $\hat{\boldsymbol{Z}} = (\hat{\boldsymbol{z}}_1, \ldots, \hat{\boldsymbol{z}}_M)$ |
| $\mathcal{M}$ | Masked token indices during training |
| $S$ | Number of autoregressive steps |
| $s$ | Autoregressive step index ($s \in [0, S-1]$) |
| $\mathcal{N}_s$ | Masked token indices to predict at step $s$ |
| $r$ | Flow matching step index |
| $v$ | Velocity field for flow matching |
| $\epsilon$ | Gaussian noise |
| $\Xi$ | Regular interpolation grid |
| $\xi$ | Interpolation grid point ($\xi \in \Xi$) |
| $\mathcal{L}$ | Loss function |
| $Q$ | Query tokens for attention |
| $K$ | Key tokens for attention |
| $V$ | Value tokens for attention |
| $m$ | Scaling factor in cross-attention bias |

# B Related Works

## B.1 Operator Learning

Operator learning has emerged as a powerful framework for modeling mappings between infinite-dimensional function spaces. Foundational works such as DeepONet and the Fourier Neural Operator (FNO) established neural operators (NOs) as effective tools for learning these mappings (Li et al., 2020a; Lu et al., 2021). Subsequent research has sought to improve both expressiveness and efficiency, exploring factorized representations for reduced complexity, wavelet-based techniques for multi-scale modeling (Gupta et al., 2021), and latent-space formulations to support general geometries (Tran et al., 2023; Li et al., 2023b). Implicit neural representations, as in CORAL, have also been proposed to accommodate variable spatial discretizations at inference time (Serrano et al., 2023).

A recent direction in operator learning leverages transformer-based architectures, inspired by their success in vision and language tasks. OFormer introduced a transformer model for embedding input-output function pairs, demonstrating the potential of transformers for operator learning (Li et al., 2023a). This has since led to more advanced architectures such as GNOT and Transolver, which improve input function encoding and generalization to irregular domains (Hao et al., 2023; Wu et al., 2024a). Aroma and UPT further adopt perceiver-style designs to learn compact and adaptable latent neural operators (Serrano et al., 2024a; Alkin et al., 2024b).

One close work is CViT, which learns compressed spatio-temporal representations using transformer encoders and decoders (Wang et al., 2024). In contrast, our approach first applies an attention-based encoder to map irregularly sampled fields to a uniform latent representation, followed by a causal spatio-temporal convolutional encoder that accommodates a flexible number of input states, enabling both regular and irregular conditioning.

## B.2 Generative Models

While most operator learning methods are deterministic, generative modeling introduces critical capabilities for modeling physical systems—most notably the ability to represent uncertainty and capture one-to-many mappings, which are especially relevant in chaotic or partially observed regimes. Two primary generative paradigms have emerged in this context: diffusion models and autoregressive transformers.

**Diffusion Transformers**  Diffusion models synthesize data by learning to reverse a progressive noising process, typically through a sequence of denoising steps (Ho et al., 2020). In computer vision, Latent Diffusion Transformers (DiTs) have demonstrated strong performance by applying this principle in the latent space of a VAE (Peebles & Xie, 2022). More recently, diffusion-based techniques have been adapted to scientific modeling. For example, Kohl et al. (2024) proposed an autoregressive diffusion framework tailored for PDEs, particularly in turbulent settings where capturing stochasticity is essential. Similarly, Lippe et al. (2023) enhanced the modeling of high-frequency chaotic dynamics via noise variance modulation during denoising. Zhou et al. (2025) extended DiTs to the physical domain, generating PDE data from textual prompts.

Despite their expressiveness, diffusion models often require many sampling steps and lack efficient in-context conditioning. As an alternative, *Flow Matching* has recently gained attention. Instead of discrete denoising steps, it learns a continuous-time velocity field that transforms one distribution into another via an ODE. This leads to significantly faster sampling with far fewer steps, making it well-suited for scientific applications (Lipman et al., 2023, 2024).

**Autoregressive Transformers**  Autoregressive models, originally designed for language modeling, have been successfully extended to image and video domains by treating spatial and temporal data as sequences. These models often couple a VQ-VAE with a causal or bidirectional transformer to model discrete token sequences (Oord et al., 2017; Esser et al., 2021; Chang et al., 2022). In video generation, frameworks such as *Magvit* and *Magvit2*(Yu et al., 2023, 2024) encode spatiotemporal information via 3D CNNs and generate frames autoregressively over quantized latent tokens. In the context of PDE modeling, *Zebra* adapts this paradigm by combining a spatial VQ-VAE with a causal transformer for in-context prediction(Serrano et al., 2025). However, the use of discrete codebooks limits expressiveness and may hinder the ability to represent fine-grained physical phenomena, which are inherently continuous.

To address this, recent advances in vision have introduced masked autoregressive transformers that operate directly on continuous latent tokens (Li et al., 2024). These Masked Autoregressive (MAR) models predict subsets of tokens using a diffusion-style loss, allowing multi-token generation without relying on vector quantization. Notably, block-wise causal masking enables efficient inference while preserving temporal consistency (Deng et al., 2024). ENMA builds on this approach by adapting MAR for neural operator learning: our model learns to autoregress over continuous latent fields, supporting in-context generalization while maintaining high fidelity and computational efficiency.

## B.3   Parametric PDEs

Generalizing to unseen PDE parameters is a central challenge in neural operator learning. Approaches span classical data-driven training, gradient-based adaptation, and emerging in-context learning strategies.

**Classical ML Paradigm**   A standard approach trains models on trajectories sampled from a distribution over PDE parameters, aiming to generalize to unseen configurations. This often involves stacking past states as input channels (Li et al., 2021) or along a temporal axis, analogous to video modeling (Ho et al., 2022; McCabe et al., 2023). However, such models typically degrade under distribution shifts, where small parameter changes lead to significantly different dynamics. Fine-tuning has been proposed to address this (Subramanian et al., 2023), but often requires substantial data per new PDE instance (Herde et al., 2024; Hao et al., 2024).

**Gradient-Based Adaptation**   To improve generalization, several methods train across multiple PDE environments. *LEADS*(Yin et al., 2022a) employs a shared model with environment-specific modules, updated during inference. Similarly, Kirchmeyer et al. (2022) introduce a hypernetwork conditioned on learnable context vectors $c^e$, enabling adaptation to new environments. Building on this, *NCF*(Nzoyem et al., 2025) uses a Taylor expansion to dynamically adjust context vectors, offering both uncertainty estimation and inter-environment adaptation.

**In-Context Learning for PDEs**   Inspired by LLMs, recent work explores in-context learning (ICL) for PDEs. Yang et al. (2023) propose a transformer that processes context-query pairs, but scalability is limited to 1D or sparse 2D settings. Cao et al. (2024) extend this to PDEs via patches for a vision transformer. They introduce a special conditionning mechanism, where input-output function are given as inputs to the transformer, allowing to handle flexible time discretizations at inference. Chen et al. (2024) propose unsupervised pretraining of neural operators, followed by ICL-style inference via trajectory retrieval and averaging—without generative modeling. Serrano et al. (2025) address this by introducing a causal transformer over discrete latent tokens, paired with an effective in-context pretraining strategy, allowing adaptation to new PDE regimes via in-context learning and probabilistic predictions.

## C  Datasets details

To evaluate the performance of **ENMA**, we generate several synthetic datasets based on diverse **parametric PDEs** in 1D and 2D. Each dataset consists of 12,000 trajectories for training. For evaluation, we generate two test sets: 1,200 trajectories for in-distribution (*In-D*) and 120 for out-of-distribution (*Out-D*) evaluation. In the *In-D* case, test parameters $\gamma$ differ from the training set but are sampled from the same distribution; in *Out-D*, they lie outside this range. Parameter ranges for the *in-D* and Out-D are presented in Table 6 and further detailed in each dataset section.

Data generation proceeds as follows: for each sampled set of PDE parameters $\gamma$ (as described in section 2), we simulate a batch of 10 trajectories using a numerical solver from 10 different initial conditions. Trajectories are computed over a time horizon $t \in [0, T]$ on a spatial grid $\mathcal{X}$. This pipeline yields datasets with diverse and complex dynamics.

We detail below the PDEs and parameter ranges used. In 1D, we consider the Combined equation (appendix C.1) and the Advection equation (appendix C.2); in 2D, we study the Wave equation (appendix C.3), the Gray-Scott system, and the Vorticity equation (appendix C.5).

Table 6: In-distribution (In-D) and out-of-distribution (Out-D) parameter ranges for each dataset.

| Dataset | Parameter | In-D | Out-D |
|---|---|---|---|
| **Combined** | $\alpha$ | $\mathcal{U}([0.3, 0.5])$ | $\mathcal{U}([0.3, 0.5])$ |
| | $\beta$ | $\mathcal{U}([0.0005, 0.5])$ | $\mathcal{U}([0.0005, 0.5])$ |
| | $\gamma$ | $\mathcal{U}([0.01, 1])$ | $\mathcal{U}([0.01, 1])$ |
| | $\delta$ | — | $\mathcal{U}([0.5, 1])$ |
| **Advection** | $\alpha$ | $\mathcal{U}([-5, 5])$ | $\mathcal{U}([-7, -5] \cup [5, 7])$ |
| **Wave** | $c$ | $\mathcal{U}([100, 500])$ | $\mathcal{U}([500, 550])$ |
| | $k$ | $\mathcal{U}([0, 50])$ | $\mathcal{U}([50, 60])$ |
| **Gray-Scott** | $F$ | $\mathcal{U}([0.023, 0.045])$ | $\mathcal{U}([0.045, 0.0467])$ |
| | $k$ | $\mathcal{U}([0.0590, 0.0640])$ | $\mathcal{U}([0.0570, 0.0590])$ |
| **Vorticity** | $\nu$ | $\mathcal{U}([10^{-3}, 10^{-2}])$ | $\mathcal{U}([10^{-5}, 10^{-4}])$ |

### C.1  Combined Equation

The **Combined equation** (Brandstetter et al., 2022) unifies several canonical PDEs—such as the heat and Korteweg–de Vries equations—by varying a set of coefficients $(\alpha, \beta, \gamma)$, enabling the modeling of diverse dynamical behaviors. It is defined as:

$$\frac{\partial u(x,t)}{\partial t} - \frac{\partial}{\partial x}\left(\alpha u(x,t)^2 - \beta \frac{\partial u(x,t)}{\partial x} + \gamma \frac{\partial^2 u(x,t)}{\partial x^2}\right) = 0, \quad (x,t) \in \Omega \times (0,T],$$

$$u(x,0) = u^0(x), \quad x \in \Omega,$$

with parameters sampled uniformly as $\alpha \sim \mathcal{U}([0.3, 0.5])$, $\beta \sim \mathcal{U}([0.0005, 0.5])$, and $\gamma \sim \mathcal{U}([0.01, 1])$ for both training and in-domain test sets.

Simulations are run over the domain $\Omega \times [0, T] = [0, 2\pi] \times [0, 1]$. The initial condition is defined as follows:

$$u^0(x) = \sum_{i=1}^{N} A_i \sin\left(\frac{2\pi l_i x}{L} + \phi_i\right),$$

with domain length $L = 2\pi$, amplitudes $A_i \sim \mathcal{U}([-0.5, 0.5])$, phases $\phi_i \sim \mathcal{U}([0, 2\pi])$, and frequencies $l_i \sim \mathcal{U}(\{1, 2, 3\})$. Each trajectory is simulated for 100 time steps over a spatial grid of 256 points. To train our models, we subsample each trajectory to retain only 20 time steps over a spatial grid of 128 points.

**Out-Domain** For out-of-distribution evaluation, we consider a more complex scenario by adding a fourth-order spatial derivative. The resulting PDE is:

$$\frac{\partial u(x,t)}{\partial t} - \frac{\partial}{\partial x}\left(\alpha u(x,t)^2 - \beta\frac{\partial u(x,t)}{\partial x} + \gamma\frac{\partial^2 u(x,t)}{\partial x^2} + \delta\frac{\partial^3 u(x,t)}{\partial x^3}\right) = 0,$$

with parameters sampled as $\alpha \sim \mathcal{U}([0.3, 0.5])$, $\beta \sim \mathcal{U}([0.0005, 0.5])$, $\gamma \sim \mathcal{U}([0.01, 1])$, and $\delta \sim \mathcal{U}([0.5, 1])$. Aside from this modification, simulations follow the same configuration as the in-distribution setup. Visualizations of the combined equation for in and out-domain trajectories are showed in Figure 4 and 5 respectively.

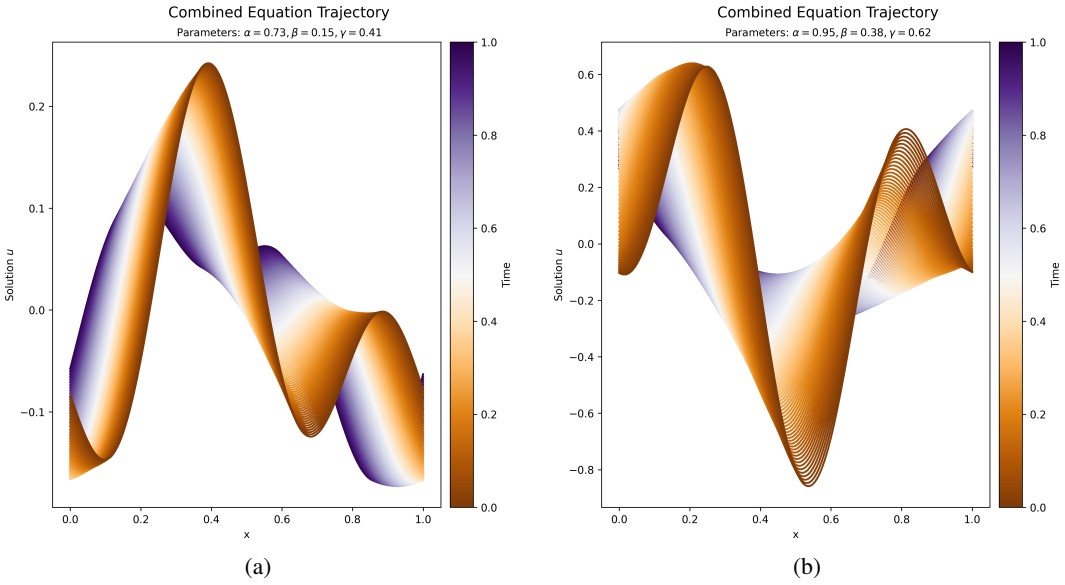

Figure 4: Samples from the Combined Dataset.

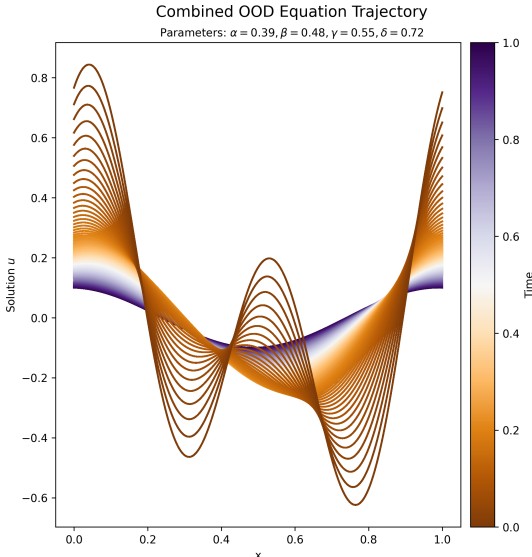

Figure 5: OOD sample of the Combined equation.

## C.2 Advection Equation

The advection equation models the transport of a quantity at constant speed. For out-domain sets, the advection speed is sampled from

For our experiments, it is solved over the domain $(\Omega \times [0, T]) = ([0, 1] \times [0, 1])$ and is defined as:

$$\frac{\partial u(x,t)}{\partial t} + \alpha \frac{\partial u(x,t)}{\partial x} = 0 \qquad (x,t) \in \Omega \times (0, T],$$
$$u(x, 0) = u^0(x) \qquad x \in \Omega,$$

where the advection speed $\alpha$ is sampled from $\mathcal{U}([-5, 5])$. For *out-of-distribution* (Out-D) evaluation, we increase the difficulty by sampling the advection speed $\alpha$ from a uniform distribution $\mathcal{U}([-7, -5] \cup [5, 7])$, explicitly excluding the training range $[-5, 5]$. For both in-domain and out-domain sets, we generate trajectories defined by three types of initial conditions:

- **Sine sum:**

$$u^0(x) = \sum_{i=1}^{N} A_i \sin\left(\frac{2\pi l_i x}{L} + \phi_i\right)$$

- **Cosine sum:**

$$u^0(x) = \sum_{i=1}^{N} A_i \cos\left(\frac{2\pi l_i x}{L} + \phi_i\right)$$

- **Sum of sine and cosine:**

$$u^0(x) = \sum_{i=1}^{N} A_i \left[\sin\left(\frac{2\pi l_i x}{L} + \phi_i\right) + \cos\left(\frac{2\pi l_i x}{L} + \phi_i\right)\right]$$

where $L = 1$, $A_i \sim \mathcal{U}([-0.5, 0.5])$, $\phi_i \sim \mathcal{U}([0, 2\pi])$, and $l_i \sim \mathcal{U}(\{1, 2, 3\})$. Each trajectory is simulated for 100 time steps over a spatial grid of 1024 points. To train our models, we subsample each trajectory to retain only 20 time steps over a spatial grid of 128 points. Visualizations of the advection equation for in and out-domain trajectories are shown in Figures 6 and 7.

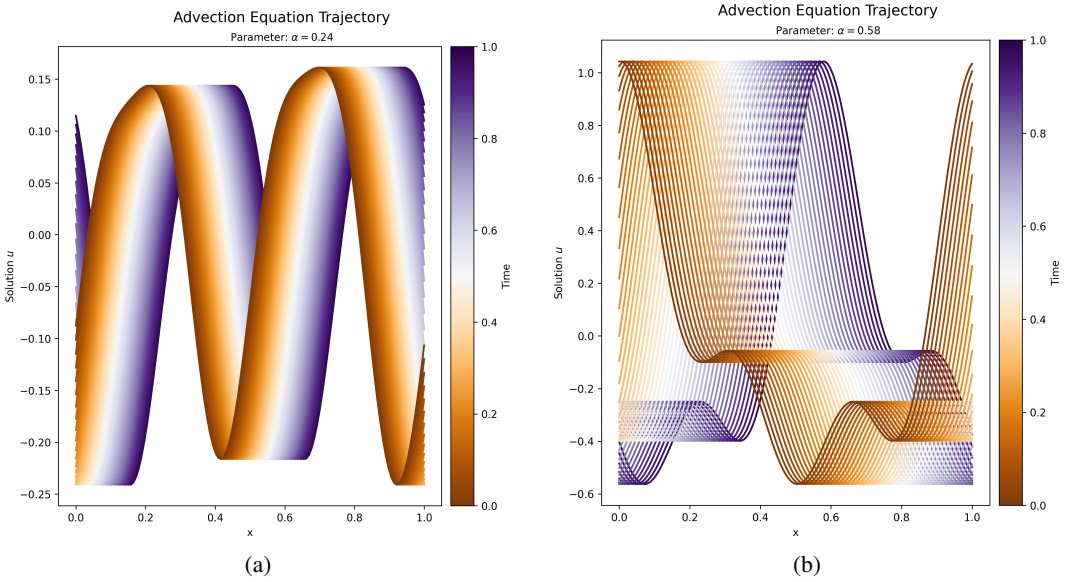

Figure 6: In-domain samples from the Advection Dataset.

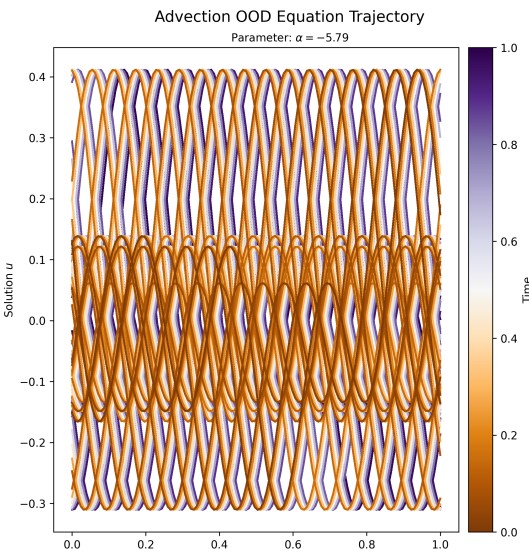

Figure 7: Out-domain sample from the Advection equation.

## C.3 Wave Equation

We consider a 2D damped wave equation defined over the domain $\Omega \times [0, T] = [0, 1]^2 \times [0, 0.005]$, given by:

$$\frac{\partial^2 \omega(x, t)}{\partial t^2} - c^2 \Delta \omega(x, t) + k \frac{\partial \omega(x, t)}{\partial t} = 0, \qquad (x, t) \in \Omega \times (0, T],$$

$$\omega(x, 0) = \omega^0(x), \qquad x \in \Omega,$$

where $c \sim \mathcal{U}([100, 500])$ is the wave speed and $k \sim \mathcal{U}([0, 50])$ the damping coefficient for the *In-D* datasets. For *Out-D*, we sample $c \sim \mathcal{U}([500, 550])$ and $k \sim \mathcal{U}[50, 60]$. We focus on learning the scalar field $\omega$. The initial condition is defined as a sum of Gaussians:

$$\omega^0(x, y) = \sum_{i=1}^{N} \exp\left(-\frac{(x - x_i L)^2 + (y - y_i L)^2}{2\sigma_i^2}\right),$$

with $x_i, y_i \sim \mathcal{U}([0, 1])$, $\sigma_i \sim \mathcal{U}([0.025, 0.1])$, $L = 1$, and $N \sim \mathcal{U}(\{2, 3, 4\})$. Simulations are performed on a $64 \times 64$ spatial grid with 30 time steps. Two in-domain trajectories can be visualized in Figure 8 and in Figure 9 for out-domain trajectories.

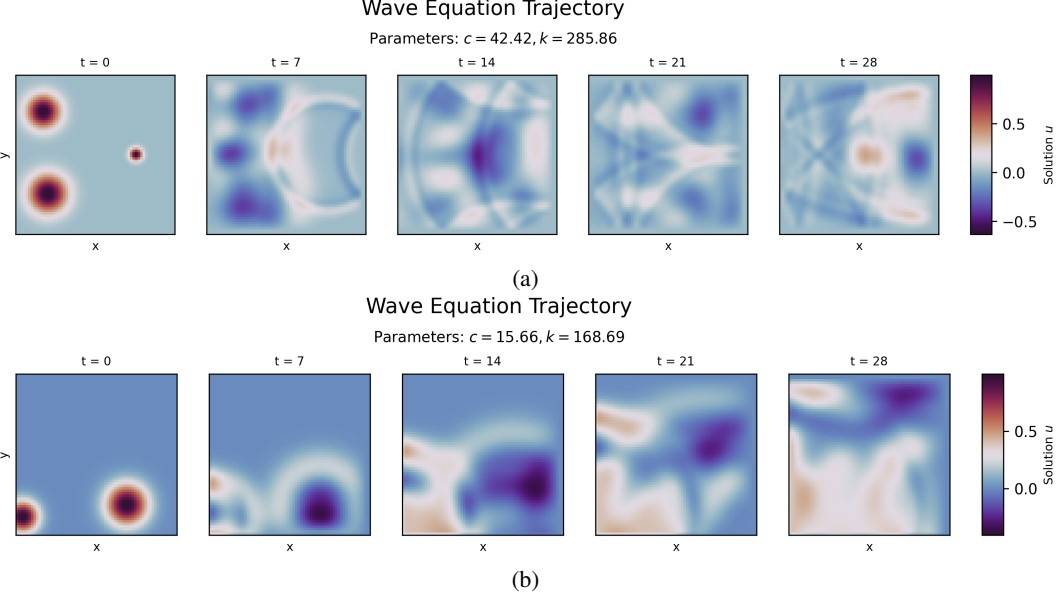

Figure 8: Samples from the Wave Dataset.

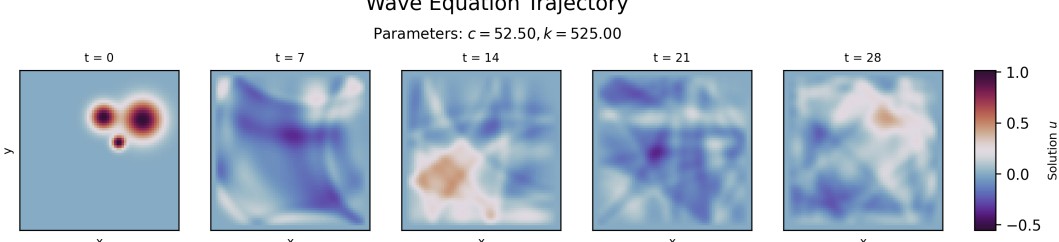

Figure 9: OOD sample of the Wave equation.

## C.4 Gray-Scott Equation

The PDE describes a reaction-diffusion system generating complex spatiotemporal patterns, governed by the following 2D equations:

$$\begin{cases} \dfrac{du}{dt} = D_u \Delta u - uv^2 + F(1-u), \\ \dfrac{dv}{dt} = D_v \Delta v - uv^2 - (F+k)v, \end{cases}$$

where $u, v$ represent the concentrations of two chemical species over a 2D spatial domain $S$, with periodic boundary conditions. The diffusion coefficients are fixed across all trajectories: $D_u = 0.102$ and $D_v = 0.204$. Each PDE instance $\gamma$ is defined by variations in the reaction parameters. For *In-D* datasets, we sample $F \sim \mathcal{U}([0.023, 0.045])$ and $k \sim \mathcal{U}([0.0590, 0.0640])$. For *Out-D* evaluation, we sample $F \sim \mathcal{U}([0.045, 0.0467])$ and $k \sim \mathcal{U}([0.0570, 0.00590])$. The spatial domain is discretized as a $32 \times 32$ grid with spatial resolution $\Delta s = 2$. Trajectories are presented in Appendix C.4 for in-domain and in Figure 11 for out-domain.

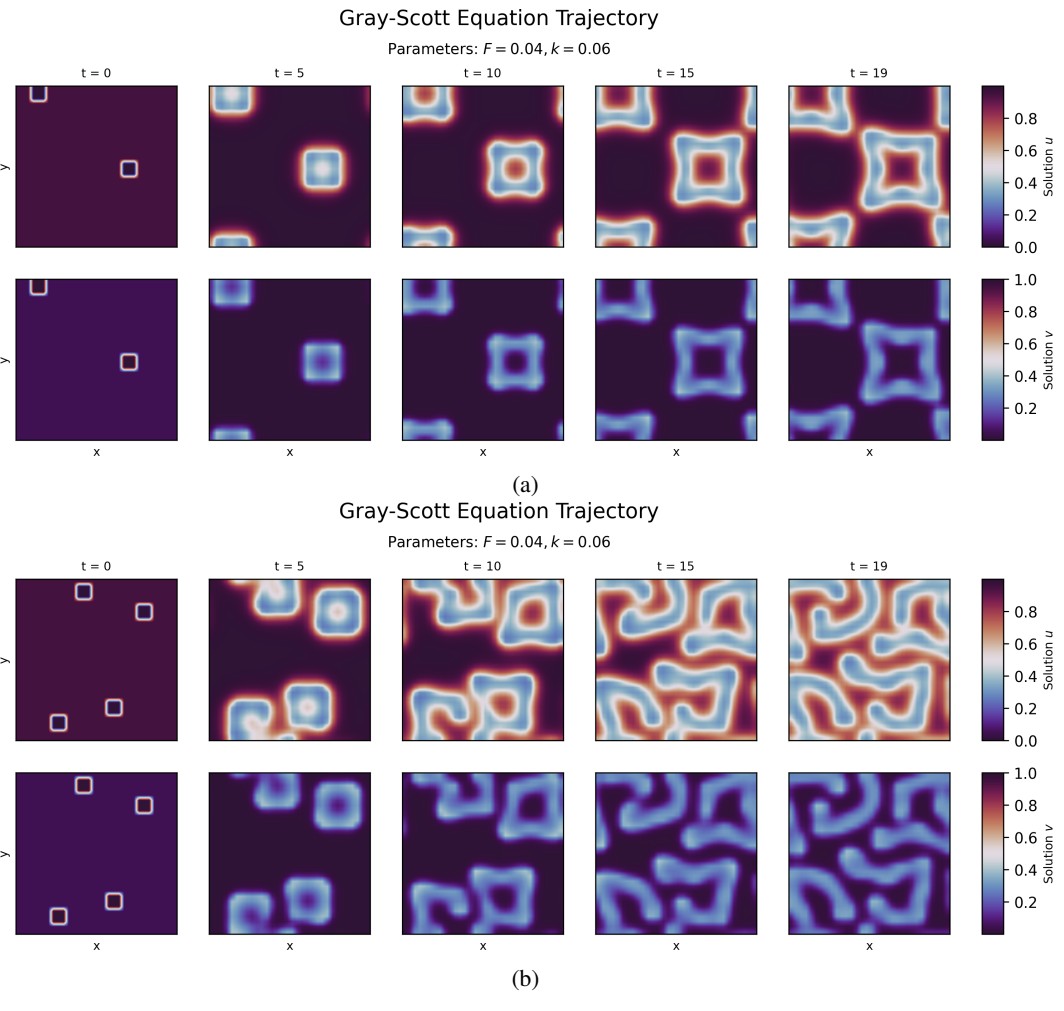

Figure 10: Samples from the Gray-Scott Dataset.

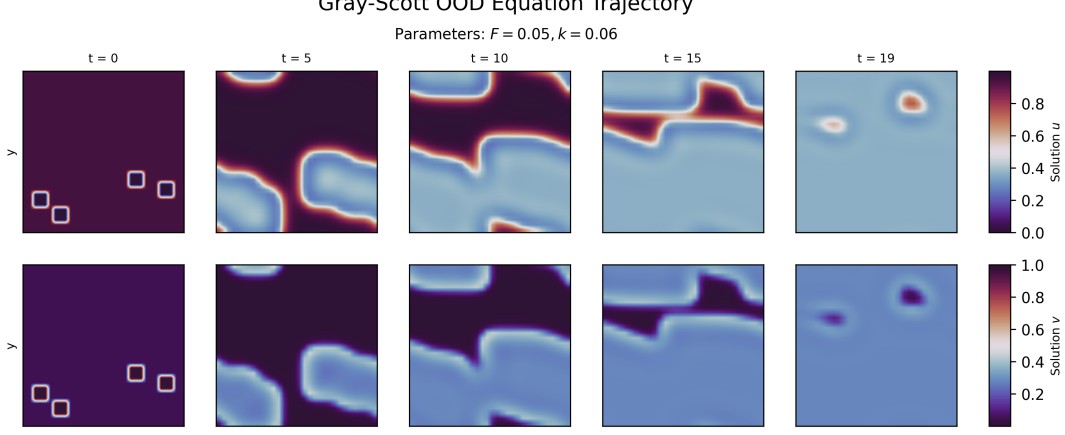

Figure 11: OOD sample of the Gray-Scott equation.

### C.5 Vorticity Equation

We consider a 2D turbulence model and focus on the evolution of the vorticity field $\omega$, which captures the local rotation of the fluid and is defined as $\omega = \nabla \times \mathbf{u}$, where $\mathbf{u}$ is the velocity field. The governing equation is:

$$\frac{\partial \omega}{\partial t} + (\mathbf{u} \cdot \nabla)\omega - \nu \nabla^2 \omega = 0, \tag{9}$$

where $\nu$ denotes the kinematic viscosity, defined as $\nu = 1/\text{Re}$. For *In-D* datasets, we sample $\nu \sim \mathcal{U}([10^{-3}, 10^{-2}])$, while for *Out-D*, we consider a more challenging turbulent regime with $\nu \sim \mathcal{U}([10^{-5}, 10^{-4}])$. The initial conditions are generated from the energy spectrum:

$$E(k) = \frac{4}{3}\sqrt{\pi} \left(\frac{k}{k_0}\right)^4 \frac{1}{k_0} \exp\left(-\left(\frac{k}{k_0}\right)^2\right), \tag{10}$$

where $k_0$ denotes the characteristic wavenumber. Vorticity is linked to energy by the following equation :

$$\omega(k) = \sqrt{\frac{E(k)}{\pi k}} \tag{11}$$

In-distribution and out-distribution trajectories can be visualized in Figure 12 and fig. 13 respectively.

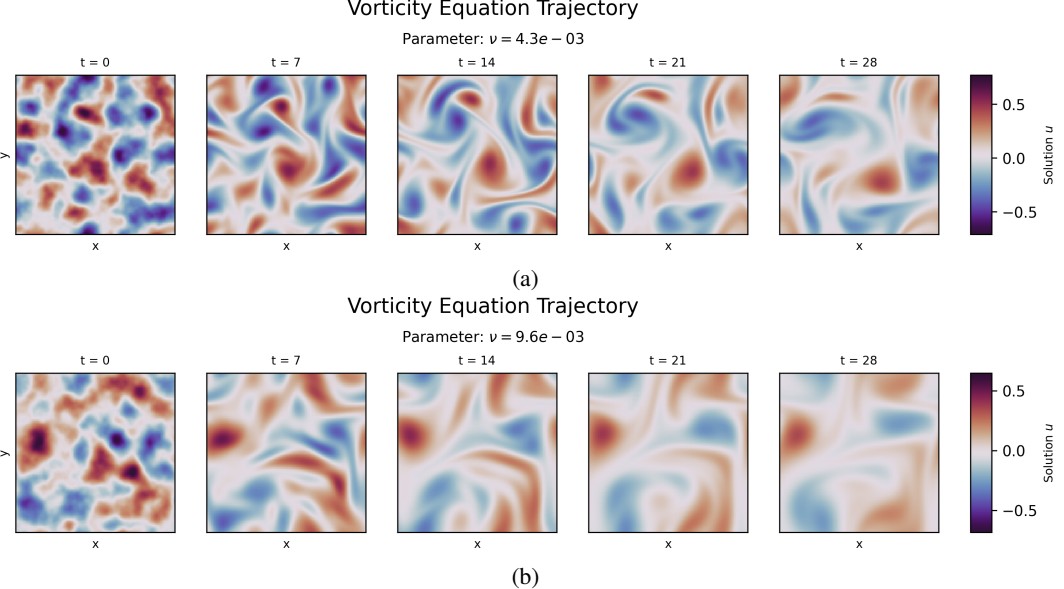

(a)

(b)

Figure 12: Samples from the Vorticity Dataset.

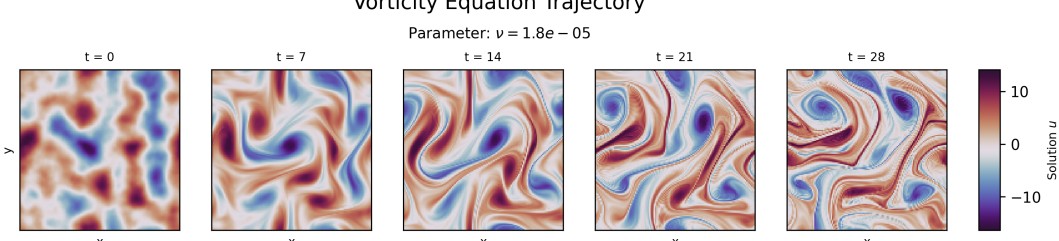

Figure 13: OOD sample of the Vorticity equation.

# D Architecture details

## D.1 ENMA encoder-decoder

We describe the architecture of the encoder–decoder module in detail below. The overall encoding and decoding pipeline is illustrated in Figure 14. Starting from an input physical field $u^{0:L-1}$ defined on a (possibly irregular) spatial grid $\mathcal{X}_{in}$, the encoder first applies an attention-based interpolation module (appendix D.1.1) to project the field onto a regular intermediate grid $\Xi$, producing $u^{0:L-1}(\Xi)$. This interpolation operates purely in space and is applied independently at each timestep.

Next, $u^{0:L-1}(\Xi)$ is passed through a causal convolutional network that encodes the data in both space and time (appendix D.1.2), yielding latent tokens $Z^{0:L_e-1}$ of length $L_e < L$ when temporal compression is used. Temporal compression is optional, for notation simplicity, we will assume we do not compress time dimension. These latent tokens form the input to the autoregressive dynamics model.

The decoder mirrors the encoder's structure. It first upsamples the latent tokens back to the intermediate grid $\Xi$ (appendix D.1.2), then uses a cross-attention module to reconstruct the physical field on any desired output grid $\mathcal{X}_{out}$. This yields the predicted trajectory $\hat{u}^{0:L-1}(\mathcal{X}_{out})$. The final stage employs the same cross-attention mechanism as the encoder's initial interpolation module (appendix D.1.1).

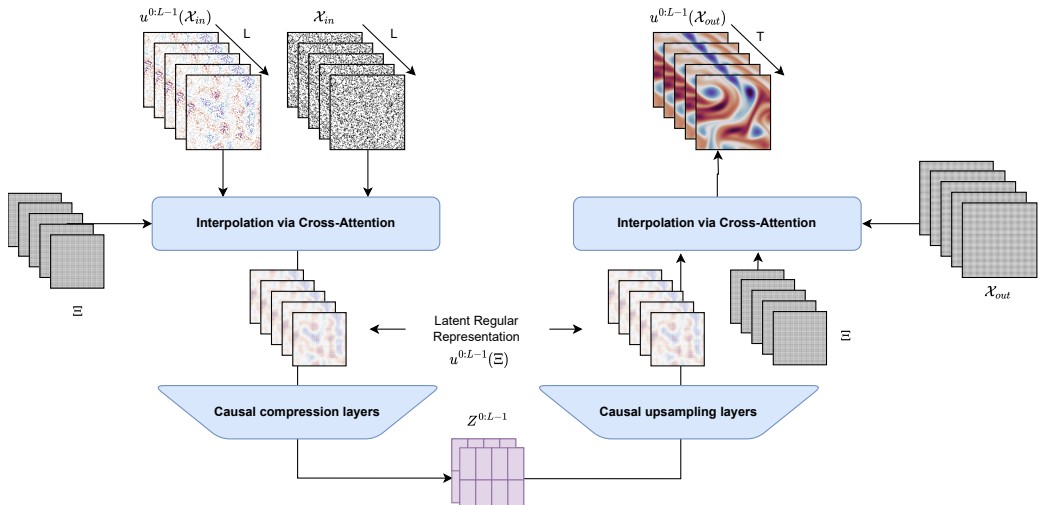

Figure 14: General architecture of ENMA's encoder/decoder.

### D.1.1 Interpolation via cross-attention

Figure 15 shows the detailed architecture of the interpolation module used in the encoder. The decoder mirrors this structure by directly reversing its operations.

The interpolation module takes as input a spatio-temporal field $u^{0:L-1} \in \mathbb{R}^{|\mathcal{X}_{in}| \times L \times c}$ defined on a potentially irregular spatial grid $\mathcal{X}_{in}$. Positional encodings are first applied to the input coordinates as in Mildenhall et al. (2021), and the physical field is projected into a higher-dimensional representation using a linear layer, yielding embeddings $p_e$ and $h$, which serve as the keys and values for the cross-attention module. Following Serrano et al. (2024a), the queries are learned latent embeddings that act as an intermediate representation.

To promote structured and localized interactions, we also initialize a learned coordinate grid $\Xi$ for these latent queries. These coordinates are used to compute attention biases, encouraging stronger attention between spatially adjacent tokens. The cross-attention module outputs an intermediate field $\in \mathbb{R}^{|\Xi| \times h}$, which is further refined using Physics Attention (Wu et al., 2024a) to reduce computational overhead.

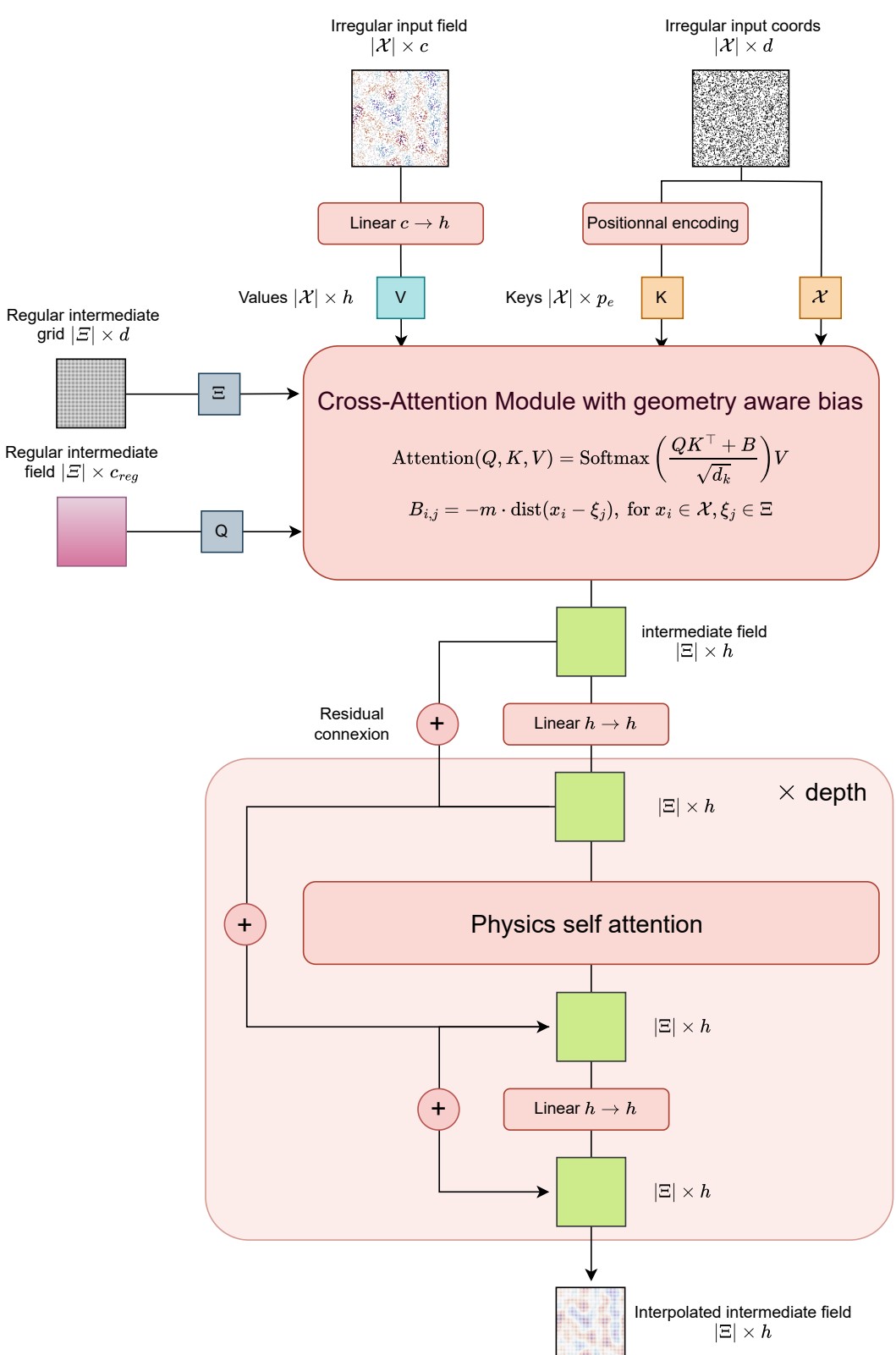

Figure 15: Detailed architecture of the Interpolation module.

**Geometry-aware encoding**    As presented in section 3.2, we introduce a geometry-aware attention bias that promotes attention locality. Appendix D.1.1 illustrates how this penalization is applied. Formally, we define the cross-attention from queries located at $\mathcal{X}$ to keys and values located at $\Xi$ as:

$$\text{Attention}(Q, K, V) = \text{Softmax}\left(\frac{QK^\top + B}{\sqrt{d_k}}\right) V, \quad B_{i,j} = -m \cdot \text{dist}(x_i - \xi_j), \text{ for } x_i \in \mathcal{X}, \xi_j \in \Xi$$

where $Q = q(\mathcal{X})$, $K = k(\Xi)$, and $V = v(\Xi)$ are the query, key, and value embeddings defined over $\mathcal{X}$ and $\Xi$, respectively; $d_k$ is the key/query dimensionality; and $m$ is a scaling factor, either fixed or learned as in ALiBi (Press et al., 2022). Intuitively, the bias $B$ imposes stronger penalties for distant pairs $(x_i, \xi_j)$, reducing their attention weights and encouraging the model to focus on local interactions during interpolation.

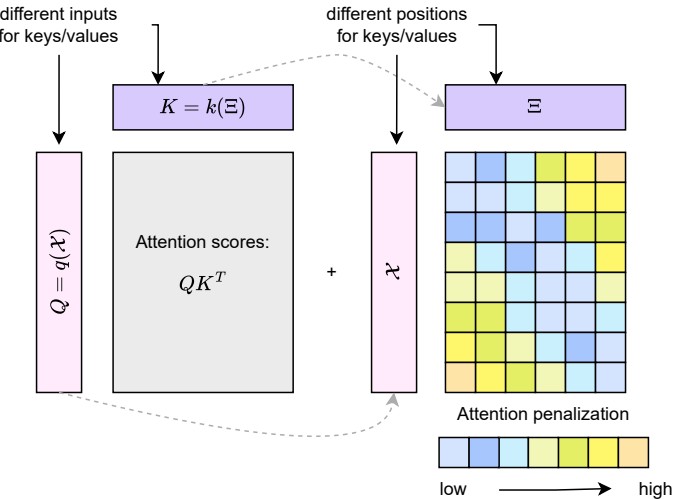

Figure 16: Geometry-aware Alibi bias.

### D.1.2    Compression via causal convolutions

For the architecture of the CNN, we adopt a design similar to that of Yu et al. (2023). In our case, it takes as input the output of the interpolation module, $u^{0:L-1}(\Xi)$, and lifts it to a higher-dimensional representation of size $h_{\text{comp}}$ using a causal convolution layer (see fig. 3). The lifted tensor is then compressed through a stack of three building blocks: `residual`, `compress_space`, and `compress_time`. Finally, a concluding residual block projects the representation back to the target latent dimension. The up-sampling module strictly mirrors this compression pipeline. This type of architecture has been shown to be effective for spatio-temporal compression (Serrano et al., 2025; Yu et al., 2023). The three types of layers used in the compression module are detailed below, and the full architecture is shown in fig. 17.

`residual` **blocks:**    The `residual` block processes the input while preserving its original shape. It consists of a causal convolution with kernel size $k$, followed by a linear layer and a Global Context layer adapted from Cao et al. (2020). If the output dimensionality differs from the input's, an additional convolution is used to project the spatial channels to the desired size.

`compress_space` **blocks:**    The `compress_space` block reduces spatial resolution by a factor of $2^d$, i.e., each spatial dimension is downsampled by a factor of 2 using a convolutional layer with stride $s = 2$. The kernel size $k$ and padding $p$ are set accordingly, with $p = k//2$. To ensure that only spatial dimensions are compressed, inputs are reshaped so that this operation does not affect the temporal axis.

`compress_time` **blocks:**    The `compress_time` block performs temporal compression similarly to the spatial case, but operates along the time dimension. To preserve causality, padding is applied

only to the past, with size $p = k - 1$, so that a frame at time $t$ attends only to frames at times $< t$. A convolution with stride $s = 2$ is used to reduce the temporal resolution by a factor of 2.

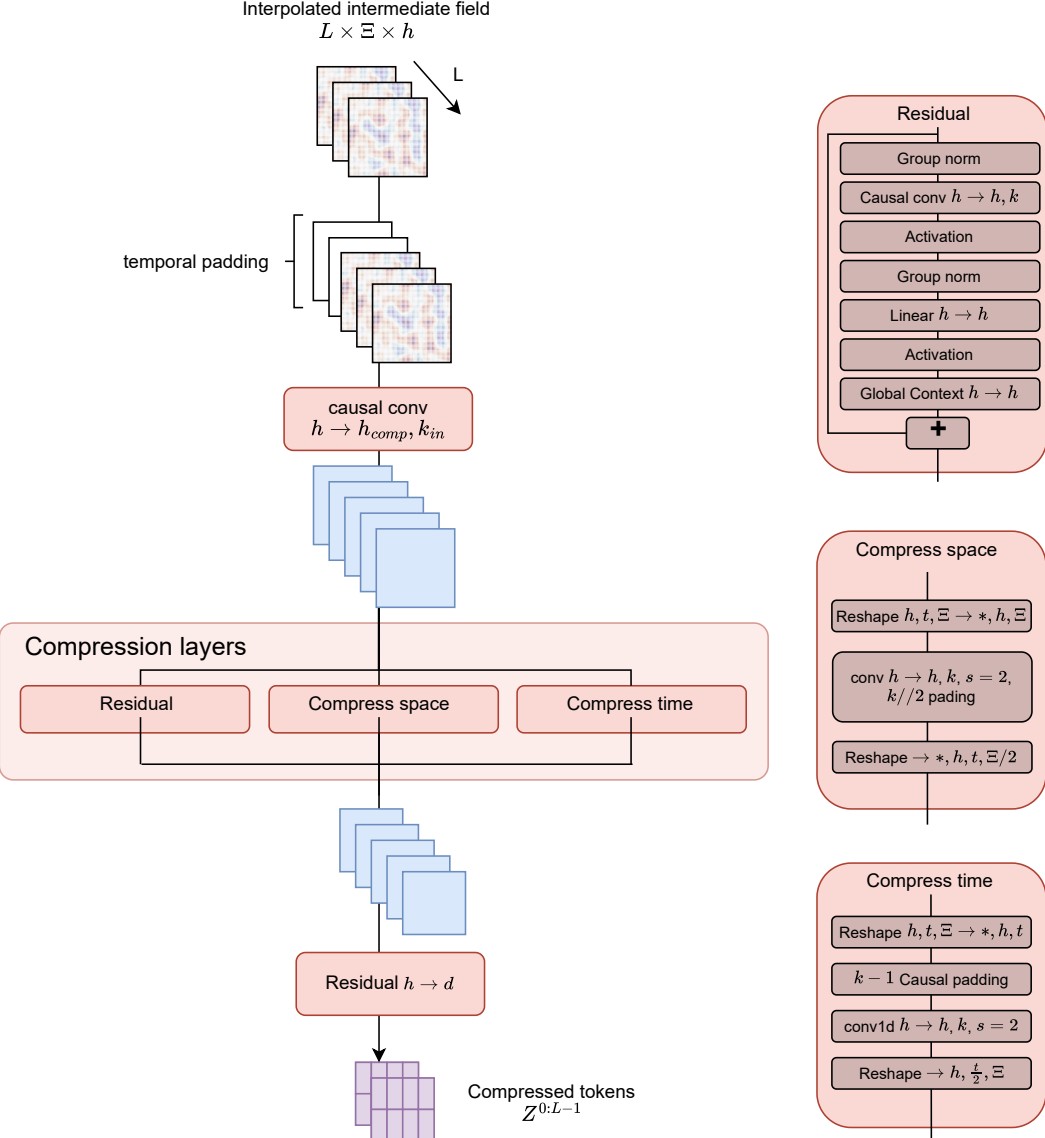

Figure 17: Detailed architecture of the Compression module.

## D.2 Tokenwise Autoregressive Generation

We provide additional details about our generative process for tokenwise autoregression of physical fields.

### D.2.1 Patchification of Latent Codes

Given a sequence of physical states $\boldsymbol{u}_{\mathcal{X}}^{0:L-1}$, the VAE encoder produces a sequence of latent representations $\boldsymbol{Z}^{0:L-1} \in \mathbb{R}^{M \times L \times d}$, where $M$ is the number of spatial tokens. For 2D physical systems, the latent representation has spatial dimensions $\boldsymbol{Z}_{M_1 \times M_2}$, with $M = M_1 \times M_2$ and $M_1 = M_2$ in our setting. The number of tokens per frame plays a critical role in capturing fine-grained spatial details. However, it also directly impacts the computational cost of tokenwise autoregressive generation, which scales with the number of tokens in each latent state.

To mitigate this cost while preserving spatial structure, we adopt a *patchify* strategy, commonly used in vision transformers. Specifically, we apply spatial down-sampling to the latent representation by dividing it into non-overlapping patches, reducing the token count per frame while retaining local coherence (see Figure 18). This significantly improves inference efficiency without sacrificing modeling fidelity.

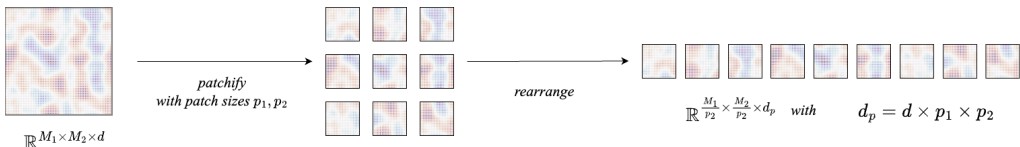

Figure 18: Illustration of latent patchification. The spatial latent map $\boldsymbol{Z}_{M_1 \times M_2}$ is partitioned into coarser patches to reduce the number of autoregressed tokens.

### D.2.2 Causal Transformer

The causal Transformer introduced in section 3.1.1 captures the dynamics necessary to predict the next latent state $\hat{\boldsymbol{Z}}^L$ from the past sequence $\boldsymbol{Z}^{0:L-1}$. We implement a next-state prediction strategy using a block-wise causal attention mask, where tokens within each frame can attend to one another, and tokens in frame $i$ can attend to all tokens from earlier frames $j < i$. The attention mask used is defined as:

$$\mathcal{M}_{\texttt{attn}} = \underbrace{\begin{bmatrix} \mathbf{1}_{M \times M} & \mathbf{0}_{M \times M} & \cdots & \mathbf{0}_{M \times M} \\ \mathbf{1}_{M \times M} & \mathbf{1}_{M \times M} & \cdots & \mathbf{0}_{M \times M} \\ \vdots & \vdots & \ddots & \vdots \\ \mathbf{1}_{M \times M} & \mathbf{1}_{M \times M} & \cdots & \mathbf{1}_{M \times M} \end{bmatrix}}_{T \text{ columns}} \left.\rule{0pt}{40pt}\right\} T \text{ rows}$$

This structure ensures that temporal causality is respected while allowing rich intra-frame interactions. It predicts a latent code $\boldsymbol{Z}_{\texttt{DYN}}^L = \text{CausalTransformer}(\boldsymbol{Z}_{\texttt{BOS}}, \boldsymbol{Z}^{0:L-1})$ that captures the dynamics given the input sequence history.

For positional information, we apply standard sine-cosine embeddings along spatial dimensions. In the temporal direction, we omit explicit positional encodings, relying instead on the causal structure to encode temporal ordering implicitly (Kazemnejad et al., 2023).

## D.3 Spatial Transformer

At inference time, given a target number of steps $S$, the spatial Transformer generates the final latent frame $\hat{\boldsymbol{Z}}^L$ in an autoregressive manner over $S$ steps. Each step selectively predicts a subset of masked tokens, conditioned on the dynamic context $\boldsymbol{Z}_{\texttt{DYN}}^L = \text{CausalTransformer}(\boldsymbol{Z}_{\texttt{BOS}}, \boldsymbol{Z}^{0:L-1})$. The token prediction schedule follows a cosine scheduler, progressively revealing tokens in a smooth, annealed fashion. Figure 19 illustrates how tokens are generated at inference for $S = 4$ steps.

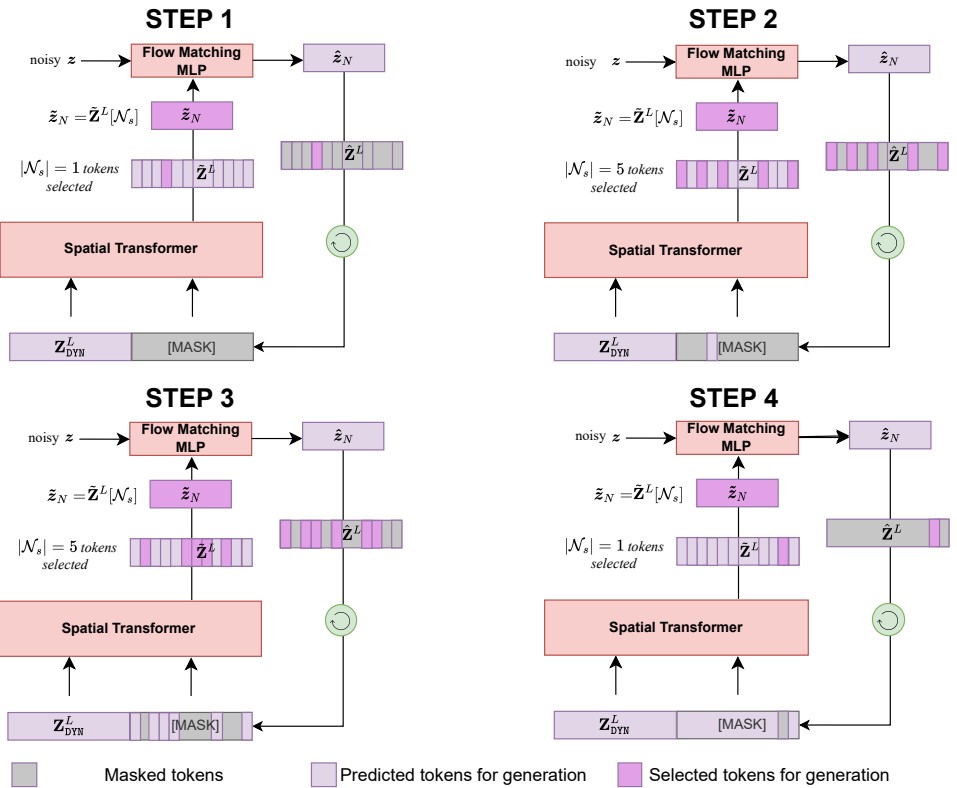

Figure 19: Illustration of tokenwise autoregressive generation at inference with a spatial Transformer over $S = 4$ steps. At each step, a subset of tokens is selected for generation based on a cosine schedule.

# E    Implementation details

The code has been written in Pytorch (Paszke et al., 2019). All experiments were conducted on a A100. We estimate the total compute budget—including development and evaluation—to be approximately 1000 GPU-days.

## E.1    Dynamics forecasting: implementation Details

This section outlines the implementation details related to the autoregressive time-stepping and generative experiments with ENMA, along with the corresponding baseline configurations.

### E.1.1    Architecture Configuration

To enable a fair comparison with existing baselines, we disable the interpolation component of ENMA's encoder-decoder when conducting autoregression on a fixed grid, consistent with the setup used in Alkin et al. (2024a). The hyperparameter settings for ENMA's generation architecture across all datasets are summarized in table 7.

Table 7: Model hyper-parameters configuration for all datasets for the ENMA's generation process.

| Hyperparameters | Combined | Advection | GS | Wave | Vorticity |
|---|---|---|---|---|---|
| Vae embedding dimension | 4 | 4 | 4 | 8 | 8 |
| Number of tokens | 16 | 16 | 64 | 256 | 256 |
| Patch size | 1 | 1 | 1 | 4 | 4 |
| Spatial Transformer depth | 6 | 6 | 6 | 6 | 6 |
| Causal Transformer depth | 6 | 6 | 6 | 6 | 6 |
| hidden size | 512 | 512 | 512 | 512 | 512 |
| mlp ratio | 2 | 2 | 2 | 2 | 2 |
| num heads | 8 | 8 | 8 | 8 | 8 |
| dropout | 0 | 0 | 0 | 0 | 0 |
| QK normalization | True | True | True | True | True |
| Normalization Type | RMS | RMS | RMS | RMS | RMS |
| activation | Swiglu | Swiglu | Swiglu | Swiglu | Swiglu |
| Layer Norm Rank | 24 | 24 | 24 | 24 | 24 |
| Positional embedding | Sinus | Sinus | Sinus | Sinus | Sinus |
| MLP depth | 3 | 3 | 3 | 3 | 3 |
| MLP width | 512 | 512 | 512 | 512 | 512 |
| number of steps $S$ | 6 | 6 | 16 | 16 | 16 |
| FM steps | 10 | 10 | 10 | 10 | 10 |

### E.1.2    Baseline details

We detail here the architecture of the baselines used to evaluate ENMA's dynamics forecasting experiments. We compareed our method to FNO (Li et al., 2020a), BCAT (Liu et al., 2025), AVIT (McCabe et al., 2023), AR-DiT (Kohl et al., 2024), Zebra (Serrano et al., 2025), an In-Context ViT and a [CLS] ViT as done in (Serrano et al., 2025).

**FNO**    For FNO, we followed the authors guidelines and concatenanted the temporal historic directly with the channels. We considered 10 modes for both 1D and 2D datasets and used a width of 128. We stacked 4 Fourier layers.

**BCAT**    BCAT is a deterministic block-wise causal transformer approach for learning spatio-temporal dynamics. It has been proposed for tackling multi-physics problems, and we adapted it for parametric PDEs. BCAT performs autoregression in the physical space. As vision transformers (Dosovitskiy

et al., 2021), it relies on patchs to reduce the number of tokens. We report the model hyper-parameters details for the Transformer in table 8.

Table 8: Model hyper-parameters configuration for all datasets for the BCAT process.

| Hyperparameters | Combined | Advection | GS | Wave | Vorticity |
|---|---|---|---|---|---|
| Patch size | 8 | 8 | 8 | 8 | 8 |
| Transformer depth | 6 | 6 | 6 | 6 | 6 |
| hidden size | 512 | 512 | 512 | 512 | 512 |
| mlp ratio | 2 | 2 | 2 | 2 | 2 |
| num heads | 8 | 8 | 8 | 8 | 8 |
| dropout | 0 | 0 | 0 | 0 | 0 |
| QK normalization | True | True | True | True | True |
| Normalization Type | RMS | RMS | RMS | RMS | RMS |
| activation | Swiglu | Swiglu | Swiglu | Swiglu | Swiglu |
| Positional embedding | Sinus | Sinus | Sinus | Sinus | Sinus |

**AVIT**    We evaluate against the Axial Vision Transformer (AViT) introduced in (Müller et al., 2022), which performs attention separately along spatial and temporal dimensions. Their proposed MPP model extends AViT to multi-physics settings by incorporating strategies for handling multi-channel inputs and system-specific normalization. In our parametric setting, such mechanisms are unnecessary. We adopt the same hyperparameter configuration as in table 8, which we found to yield the best results.

**AR-DiT**    Our autoregressive diffusion transformer follows the setup proposed in Kohl et al. (2024), where known frames are concatenated with noise and denoised progressively to generate the next physical state. As in (Rombach et al., 2022), we employ AdaLayerNorm to condition the model during the diffusion process. The corresponding hyperparameters are provided in table 9.

Table 9: Model hyper-parameters configuration for all datasets for the AR-DIT process.

| Hyperparameters | Combined | Advection | GS | Wave | Vorticity |
|---|---|---|---|---|---|
| Patch size | 8 | 8 | 8 | 8 | 8 |
| Transformer depth | 6 | 6 | 6 | 6 | 6 |
| hidden size | 512 | 512 | 512 | 512 | 512 |
| mlp ratio | 4 | 4 | 4 | 4 | 4 |
| num heads | 8 | 8 | 8 | 8 | 8 |
| dropout | 0 | 0 | 0 | 0 | 0 |
| Normalization Type | AdaLayer | AdaLayer | AdaLayer | AdaLayer | AdaLayer |
| Positional embedding | Sinus | Sinus | Sinus | Sinus | Sinus |
| Diffusion steps | 100 | 100 | 100 | 100 | 100 |

**Zebra**    Zebra is a token-based autoregressive model that combines a VQ-VAE for discretizing spatial fields with a causal transformer for modeling temporal dynamics. At each time step, it autoregressively predicts discrete latent tokens corresponding to the next frame, allowing for uncertainty quantification through stochastic sampling in latent space. The full set of hyperparameters for both the VQ-VAE and the transformer components is provided in table 10.

**In-Context ViT**    We implement an in-context Vision Transformer (Dosovitskiy et al., 2020) to address the initial value problem using a context trajectory. The model is trained to forecast the next frame in a target sequence, conditioned on both the observed context and preceding frames. For architectural configuration, we adopt the same hyperparameters as reported in table 8.

Table 10: Hyperparameters for Zebra's VQVAE and Transformer components.

| Hyperparameters | Advection | Combined | GS | Wave | Vorticity |
|---|---|---|---|---|---|
| **VQ-VAE** | | | | | |
| start_hidden_size | 64 | 64 | 128 | 128 | 128 |
| max_hidden_size | 256 | 256 | 1024 | 1024 | 1024 |
| num_down_blocks | 4 | 4 | 2 | 3 | 2 |
| codebook_size | 256 | 256 | 2048 | 2048 | 2048 |
| code_dim | 64 | 64 | 16 | 16 | 16 |
| num_codebooks | 2 | 2 | 1 | 2 | 1 |
| shared_codebook | True | True | True | True | True |
| tokens_per_frame | 32 | 32 | 256 | 128 | 256 |
| start learning_rate | 3e-4 | 3e-4 | 3e-4 | 3e-4 | 3e-4 |
| weight_decay | 1e-4 | 1e-4 | 1e-4 | 1e-4 | 1e-4 |
| scheduler | Cosine | Cosine | Cosine | Cosine | Cosine |
| num_epochs | 1000 | 1000 | 300 | 300 | 300 |
| **Transformer** | | | | | |
| max_context_size | 2048 | 2048 | 8192 | 8192 | 8192 |
| batch_size | 4 | 4 | 2 | 2 | 2 |
| num_gradient_accumulations | 1 | 1 | 4 | 4 | 4 |
| hidden_size | 256 | 256 | 384 | 512 | 384 |
| mlp_ratio | 4.0 | 4.0 | 4.0 | 4.0 | 4.0 |
| depth | 8 | 8 | 8 | 8 | 8 |
| num_heads | 8 | 8 | 8 | 8 | 8 |
| vocabulary_size | 264 | 264 | 2056 | 2056 | 2056 |
| start learning_rate | 1e-4 | 1e-4 | 1e-4 | 1e-4 | 1e-4 |
| weight_decay | 1e-4 | 1e-4 | 1e-4 | 1e-4 | 1e-4 |
| scheduler | Cosine | Cosine | Cosine | Cosine | Cosine |
| num_epochs | 100 | 100 | 30 | 30 | 30 |

**[CLS] ViT**    This variant of the Vision Transformer is adapted to a meta-learning setting, where a dedicated [CLS] token captures the environment-specific variations across different PDE settings. During inference, the [CLS] token is updated via 100 gradient steps to adapt to new environments. We use the same model configuration as in table 8.

### E.1.3    Training details

Models for 1D datasets were trained for approximately 8 hours, while training on 2D datasets took around 20 hours. Unless otherwise specified, all datasets followed the same training objective.

**Optimizer and Learning Rate Schedule**    We use the AdamW optimizer with $\beta_1 = 0.9$ and $\beta_2 = 0.95$ for all experiments. The learning rate follows a cosine decay schedule, starting from an initial value of $10^{-3}$ and annealing to $10^{-5}$ over the course of training. To stabilize the early training phase, we apply a linear warmup over the first 500 optimization steps.

**Baselines**    All baselines were trained following the same protocol as ENMA. Each model was trained from scratch for 1000 epochs, and the best-performing checkpoint was selected based on training performance. A cosine learning rate scheduler was used across all methods, which consistently improved prediction quality. All models were trained with a learning rate of $1e^{-3}$, except for AR-DiT, which required a lower rate of $1e^{-4}$ due to unstable training dynamics.

Table 11: Training hyper-parameters configuration for all datasets for the ENMA's generation process.

| Hyperparameters | Combined | Advection | GS | Wave | Vorticity |
|---|---|---|---|---|---|
| Epochs | 1000 | 1000 | 1000 | 1000 | 1000 |
| batch size | 128 | 128 | 128 | 64 | 64 |
| learning rate | 1e-3 | 1e-3 | 1e-3 | 1e-3 | 1e-3 |
| weight Decay | 1e-4 | 1e-4 | 1e-4 | 1e-4 | 1e-4 |
| grad clip norm | 1 | 1 | 1 | 1 | 1 |
| betas | [0.9, 0.95] | [0.9, 0.95] | [0.9, 0.95] | [0.9, 0.95] | [0.9, 0.95] |

## E.2 Encoder-Decoder implementation protocol

### E.2.1 Architecture details

We present in table 12 the hyper parameters for the architecture of ENMA's auto encoder.

Table 12: Architecture details of the auto encoder as presented in table 1. We refer the reader to appendices D.1.1 and D.1.2 for the notation.

| Module | Block | Parameter | Advection | Vorticity |
|---|---|---|---|---|
| | | token dim | 4 | 8 |
| Interpolation module | intermediate grid size | $\lvert \Xi \rvert$ | 128 | $16 \times 16$ |
| | Positional encoding | positional encoding | Nerf | Nerf |
| | | positional encoding num freq | 12 | 12 |
| | | positional encoding max freq | 4 | 4 |
| | Cross attention block | intermediate grid dim $c_{reg}$ | 64 | 16 |
| | | hidden dim $h$ | 16 | 16 |
| | | n cross-attention heads | 4 | 4 |
| | | cross-attention dim heads | 4 | 4 |
| | Geometry-aware bias | alibi scaling $m$ | 1, 2, 3, 4 | 1, 2, 3, 4 |
| | | n alibi heads | 4 | 4 |
| | Physics self attention | depth | 2 | 2 |
| | | n attention heads | 4 | 4 |
| | | attention dim heads | 4 | 4 |
| | | physics-attention n slice | 16 | 64 |
| Compression module | Compression layers | Compression layers | residual compress_space residual compress_space residual compress_space residual compress_time residual | residual compress_space residual compress_time residual - - - - |
| | | $h_{comp}$ | 16 | 16 |
| | | compression kernel size $k$ | 7 | 7 |
| | causal input layer | kernel size $k_{in}$ | 7 | 7 |
| | causal output layer | kernel size $k_{out}$ | 7 | 7 |

### E.2.2 Baseline details

We detail here the architecture of the baselines used to evaluate ENMA's auto encoding. Hyper-parameters are chosen from the original papers and/or closest available implementation. We recall that Oformer acts pointwise in the physical space, and thus does not make any spatial nor temporal compression. CORAL, AROMA and GINO act at a frame level *i.e.* they compress space but not time. Finally, our ENMA's auto encoder architecture performs both a spatial and a temporal compression. We provide a comparison in table 13.

Table 13: Token sizes used to evaluate auto encoders as presented in table 1. We show sizes using the following formatting: temporal size $\times$ spatial size $\times$ token dimension.

| Model | Advection | Vorticity |
|---|---|---|
| *Trajectory sizes* | $50 \times 128 \times 1$ | $30 \times (64 \times 64) \times 1$ |
| Oformer | $50 \times 128 \times 4$ | $30 \times (64 \times 64) \times 8$ |
| GINO | $50 \times 16 \times 4$ | $30 \times (16 \times 16) \times 8$ |
| AROMA | $50 \times 16 \times 4$ | $30 \times 64 \times 8$ |
| CORAL | $50 \times 64$ | $30 \times 512$ |
| ENMA | $26 \times 16 \times 4$ | $16 \times (8 \times 8) \times 8$ |

**Oformer**    Oformer models are trained point wise *i.e.* no spatial neither temporal compression are processed. We used 4-depth layers with Galerkin attention type and 128 latent channels.

**GINO**    GINO architecture is a combination of 2 well-known neural operator architecture (GNO (Li et al., 2020b) and FNO (Li et al., 2020a)). The GNO uses the following architecture parameters: a GNO radius of 0.033 with linear transform. The latent grid size is 16 for 1d datasets and $16 \times 16$ for 2d. As a consequence, we adapt the FNO modes to 8 for 1d datasets and $8 \times 8$ for 2d's.

**AROMA**    AROMA makes use of a latent token of size 16 in 1d and 64 in 2d. Other architecture parameters are kept similar *i.e.* 4 cross and latent heads with dimension 32. The hidden dimension of AROMA's architecture is set to 128. The INR has 3 layers of width 64.

**CORAL**    Finally, CORAL is a meta-learning baseline that represents solution using an INR (SIREN (Sitzmann et al., 2020)), conditioned from a hyper-network. The hyper-network takes as input a latent code, optimized with 3 gradient descent steps. The inner learning rate is set to 0.01. The hyper network has 1 layer with 128 neurons. SIREN models have 6 layers with 256 neurons.

As a comparison between baselines, we show on table 13 the latent token sizes that are created with the hyper parameters detailed above.

### E.2.3    Training details

**Reconstruction performances**    For training the auto-encoder (as well as the baseline models, unless stated otherwise), we followed a unified training procedure. The loss function combines a relative mean squared error term with a KL divergence term: $\mathcal{L} = \mathcal{L}\text{recon} + \beta \cdot \mathcal{L}\text{KL}$, where $\beta = 0.0001$.

To enhance robustness to varying levels of input sparsity, we randomly subsample the spatial input grid during training. The number of input points ranges from $20\%$ to $100\%$ of the full grid $\mathcal{X}$. The output grid used for loss computation remains fixed to the full grid. Grid subsampling is performed independently for each sample in a batch and is refreshed at every iteration.

Models are optimized using the AdamW optimizer with an initial learning rate of 0.001, scheduled via a cosine decay down to $1e^{-7}$ over the course of training. Training batch sizes for each encoder–decoder architecture are listed in table 14.

- **ENMA:** Trained with a learning rate of 0.001 and batch size of 64 in 1D. The autoencoder is trained for $1,000$ epochs in 1D and 150 epochs in 2D.

- **AROMA:** Training procedure follows ENMA. 2D datasets are trained for $1,000$ epochs.

- **CORAL:** To stabilize training, the KL loss term is removed. The model is trained as a standard autoencoder minimizing relative MSE, with an initial learning rate of $1e-6$.

- **Oformer:** Follows the same training procedure as ENMA. 2D datasets are trained for $1,000$ epochs.

- **GINO:** Training setup is similar to ENMA. 2D datasets are trained for 100 epochs due to the inner loop over batches in the forward pass.

Table 14: Batch size used for training autoencoders as presented in table 1.

| Model | Advection | Vorticity |
|---|---|---|
| Oformer | 64 | 4 |
| GINO | 64 | 32 |
| AROMA | 128 | 8 |
| CORAL | 128 | 8 |
| ENMA | 64 | 8 |

**Time-stepping**   The time-stepping task consists in training a small FNO to unroll the dynamics in the latent space created by the different models. This allows us to assess the quality of the extracted features for dynamic modeling. Training proceeds as follows: Starting from a trajectory, we encode the entire trajectory. We then take the 2 first tokens in the temporal dimension as input for the FNO. By concatenating this to the PDE parameters, we unroll the dynamic in the latent space, to build the full sequence of tokens. We compute the loss (Relative MSE) **in the latent space** directly and at each step in the auto regressive process. All models are trained using an initial learning rate of 0.0001 with and AdamW optimizer. The learning is also scheduled using a cosine scheduler. 250 epochs are performed for all baselines in 1d and in 2 GINO and CORAL performs 50 training epochs, while Oformer and CORAL are trained for 150 epochs. This trainings can differs depending on the computational cost of the encoder.

The FNO used for processing is a simple 1d/2d FNO, with 3 layers of width 64. FNO modes are set to 8 and the activation function is a GELU fonction.

### E.3   Dynamics forecasting on complex systems: implementation details

This section outlines the implementation details related to the auto-regressive time-stepping experiments with ENMA on the two complex physical systems: Rayleigh-Bénard and Active Matter (Ohana et al., 2024).

On these two datasets, we disabled the interpolation component of ENMA's encoder-decoder and used a standard auto-encoder to boost reconstruction performance, instead of using a variational formulation. Autoregression is then performed on a fixed grid with a sequence length of size 3. 12 and 17 time-steps are predicted respectively for Rayleigh-Bénard and Active Matter. We report the hyper-parameteres used for the experiments in table 15.

Table 15: Model hyperparameter configuration for ENMA on the Active Matter and Rayleigh–Bénard.

| Hyperparameters | Active Matter | Rayleigh–Bénard |
|---|---|---|
| VAE embedding dimension | 32 | 32 |
| Number of tokens | 256 | 256 |
| Patch size | 2 | 2 |
| Spatial Transformer depth | 6 | 6 |
| Causal Transformer depth | 6 | 6 |
| Hidden size | 512 | 512 |
| MLP ratio | 4 | 4 |
| Num heads | 8 | 8 |
| QK normalization | True | True |
| Normalization type | RMS | RMS |
| Positional embedding | Sinus | Sinus |
| MLP depth | 3 | 3 |
| MLP width | 512 | 512 |
| Number of steps $S$ | 16 | 16 |
| FM steps | 10 | 10 |

# F Additional experiments

We conduct a wide range of experiments to evaluate the capabilities of our model **ENMA**:

- Generative capabilities (appendix F.1.1): evaluation on uncertainty quantification, sample diversity, solver fidelity, and latent distribution alignment.
- Compression study (appendix F.1.2): Dynamics forecasting evaluation with reduced latent compression on 2D datasets.
- Inference speed comparison (appendix F.1.3): benchmarking ENMA against generative baselines.
- Ablation studies (appendix F.1.4): investigating the impact of history length, autoregressive steps, flow matching iterations and the dynamics modeling objective.
- OOD reconstruction (appendix F.2.2): testing under high sparsity with only 10% of the spatial input grid.
- Super-resolution (appendix F.2.3): evaluation on finer output resolutions beyond the training grid.
- Encoder-decoder variants (appendix F.2.4): ablations on positional bias and time-stepping architectures.
- Geometry-Aware Attention Analysis (appendix F.2.5): qualitative inspection of the geometry-aware attention module
- Token space visualization (appendix F.2.6): qualitative visualization of the encoded tokens

## F.1 ENMA process task

### F.1.1 Generative ability of ENMA

We illustrate here additional benefits from the generative capabilities of **ENMA** through two example tasks: *uncertainty quantification* and *new trajectory generation*. For all uncertainty experiments, we focus on the Combined equation, as generating multiple samples can be costly for 2D data.

**Uncertainty quantification** ENMA naturally enables uncertainty quantification by generating multiple stochastic samples from the learned conditional distribution. This is achieved by sampling multiple candidates in the latent space via different noise realizations during the spatial decoding process. Specifically, for each autoregressive step, ENMA injects Gaussian noise into the flow matching sampler, producing diverse plausible predictions for the spatial tokens. Repeating this process yields an ensemble of trajectories. In contrast, discrete AR models rely on sampling from categorical distributions (e.g., top-$k$ or nucleus sampling), which often leads to overconfident outputs, limited diversity, and poorly calibrated uncertainty estimates.

An illustration is shown in fig. 20, where the red curve denotes the ground truth, the blue curve is the predicted mean at the final time step, and the shaded region indicates the empirical confidence interval defined by $\pm 3$ standard deviations.

To evaluate uncertainty quality, we report two standard metrics:

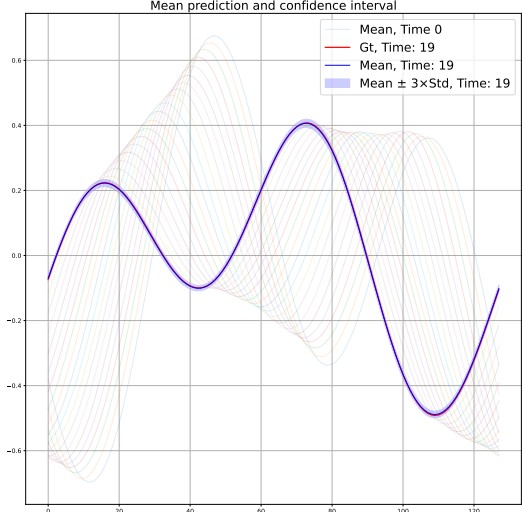

Figure 20: Uncertainty quantification using ENMA. Multiple trajectories are sampled, and the final time step is used to compute the pointwise mean (blue), standard deviation (shaded), and ground truth (red).

- **CRPS** (Continuous Ranked Probability Score) measures the accuracy and sharpness of probabilistic forecasts by comparing the predicted distribution to the true outcome. Lower CRPS indicates better probabilistic calibration.

- **RMSCE** (Root Mean Squared Calibration Error) quantifies calibration by comparing predicted confidence intervals with empirical coverage. A low RMSCE suggests well-calibrated uncertainty estimates.

| Method | Metric | Combined |
|---|---|---|
| AR-DiT | RMSCE | 2.68e-1 |
| | CRPS | 1.27e-2 |
| Zebra | RMSCE | 2.19e-1 |
| | CRPS | 9.00e-3 |
| ENMA (ours) | RMSCE | **8.68e-2** |
| | CRPS | **1.70e-3** |

Table 16: Comparison of uncertainty metrics (↓ is better) for Combined.

**CRPS and RMSCE Results** ENMA significantly outperforms both **AR-DiT** and **Zebra** in terms of uncertainty calibration and probabilistic accuracy, as shown in table 16. These results indicate that ENMA provides both sharper and more reliable uncertainty estimates. The stochastic sampling in latent space enabled by flow matching allows ENMA to model diverse plausible trajectories while maintaining strong calibration. In contrast, discrete token-based methods like Zebra and AR-DiT tend to produce overconfident or underdispersed forecasts. This demonstrates the advantage of per continuous latent-space modeling for uncertainty-aware forecasting.

**Temporal Uncertainty Evaluation.** To further assess the quality of uncertainty calibration over time, we report the evolution of RMSCE and CRPS across autoregressive prediction steps in fig. 21. Ideally, both metrics should remain low and stable over time, indicating that the model maintains well-calibrated uncertainty estimates and accurate probabilistic forecasts throughout the trajectory. For CRPS, all models exhibit increasing scores over time due to the compounding errors typical of neural PDE solvers. However, ENMA consistently achieves substantially lower CRPS values than both Zebra and AR-DiT, indicating sharper and more accurate trajectory forecasts. In the case of RMSCE, ENMA displays a distinct trend: its calibration error decreases over time. This behavior reflects the model's evolving confidence—initial predictions, informed by ground truth history, tend to be overly confident, resulting in narrower (and sometimes miscalibrated) confidence intervals. As the model progresses through the trajectory and relies more on its own predictions, it becomes more conservative, yielding better-calibrated uncertainty estimates. ENMA maintains the lowest RMSCE at every step, underscoring its robustness in providing reliable confidence intervals throughout the rollout. Together, these results highlight ENMA's superiority in delivering both accurate and well-calibrated probabilistic forecasts over time.

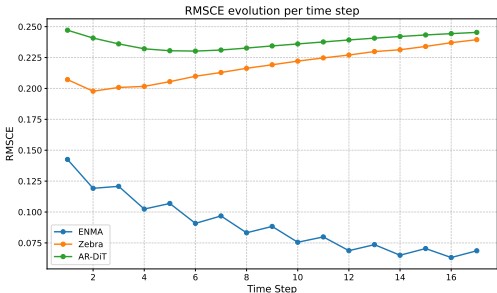 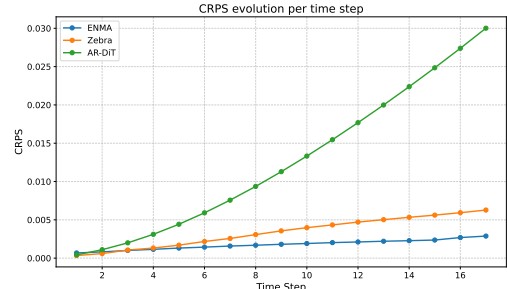

(a) RMSCE evolution over time. Lower values indicate better uncertainty calibration.

(b) CRPS evolution over time. Lower values reflect sharper and more accurate forecasts.

Figure 21: Evolution of uncertainty metrics over time for the Combined dataset. (a) RMSCE and (b) CRPS demonstrate that ENMA outperforms baselines in both calibration and predictive sharpness.

**Data generation** As a second demonstration of ENMA's generative capabilities, we evaluate its ability to produce plausible and diverse trajectories *without conditioning on the initial state* $\mathbf{u}^0$.

Specifically, we consider the setting of *generation conditioned on a context example*. Unlike classical neural approaches that typically require an initial condition and generate future states given $\gamma$, this setup assumes no access to either $\boldsymbol{u}^0$ or $\gamma$. Instead, the model is provided with a context trajectory from which it must infer the underlying parameter $\gamma$ and generate a coherent trajectory, including a plausible initial condition. This setting highlights ENMA's capacity to function as a generative solver for parametric PDEs.

To benchmark generative performance, we compare ENMA against baseline models using standard metrics from the generative modeling literature. We introduce the *Fréchet Physics Distance* (FPD), a physics-adapted variant of Fréchet Inception Distance (FID) Heusel et al. (2017), along with Precision and Recall Kynkäänniemi et al. (2019). These metrics are computed in a compressed feature space extracted by a lightweight CNN encoder trained to regress the underlying PDE parameters $\gamma$, treated as instance classes.

The CNN processes the full trajectory and encodes it into a 64-dimensional feature vector, enabling semantically meaningful comparisons while mitigating the curse of dimensionality. In this space, FPD estimates the 2-Wasserstein distance between real and generated distributions, while Precision and Recall respectively measure sample fidelity and diversity. Following standard protocol, we use the training set as the reference distribution. Results for the Combined dataset are reported in table 17.

Table 17: Comparison of generative (FPD, Precision and Recall) metrics on the Combined dataset.

| Model | FPD $\downarrow$ | Precision $\uparrow$ | Recall $\uparrow$ |
|---|---|---|---|
| Zebra | 1.03e-1 | 7.70e-1 | **8.61e-1** |
| ENMA | **9.50e-3** | **7.92e-1** | 7.80e-1 |

**Results**  ENMA achieves the lowest FPD, significantly outperforming Zebra— indicating that ENMA's generated trajectories are statistically closer to the real data distribution in the learned feature space. This reflects ENMA's ability to generate high-fidelity samples that are well-aligned with physical ground truth. In terms of Precision, ENMA also outperforms Zebra, demonstrating superior sample quality. However, Zebra obtains a higher Recall, suggesting that it covers a broader range of modes from the data distribution. This highlights a trade-off: ENMA prioritizes fidelity and realism, whereas Zebra exhibits slightly more diversity, potentially at the cost of precision. Overall, these results illustrate ENMA's strength in producing high-quality, physically plausible trajectories, while maintaining reasonable diversity in its generative predictions.

**Latent distribution alignment**  To further analyze the generative behavior of ENMA and Zebra, we visualize PCA projections of feature representations extracted by a CNN trained on ground truth data. As shown in fig. 22, ENMA's samples (orange) closely align with the real data (blue), indicating that the generated trajectories remain well-covered within the training distribution. In contrast, Zebra exhibits a large number of outliers that fall outside the support of the real data distribution. This mismatch suggests poorer calibration and lower fidelity to the training dynamics, which aligns with the lower FPD and precision scores reported in table 17. These findings confirm ENMA's ability to generate physically plausible and distributionally consistent trajectories.

**Fidelity with respect to the numerical solver**  To evaluate the fidelity of ENMA's generations, we take advantage of the known parametric structure of the PDEs. For each sample, we condition ENMA on a context trajectory and let it generate a full trajectory, including the initial state. Since the true PDE parameters $\gamma$ used to generate the context are known, we can pair ENMA's generated initial condition with the ground-truth $\gamma$ and run the numerical solver used during dataset creation. This produces a reference trajectory governed by the true physical dynamics. By comparing ENMA's generated trajectory with the solver-based rollout, we can assess how well ENMA captures the underlying PDE. A close match indicates that ENMA has learned to infer physically consistent initial conditions and dynamics from context.

**Results**  As shown in fig. 23, the trajectories produced by ENMA closely match those from the solver, both qualitatively and quantitatively. We also report a relative L2 error of **0.081** between the

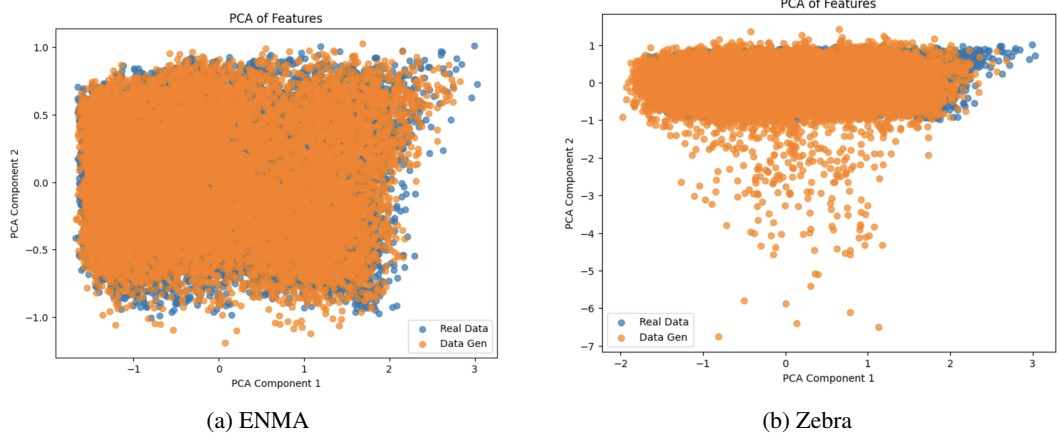

(a) ENMA                                    (b) Zebra

Figure 22: PCA projections of CNN features from generated (orange) and real (blue) trajectories at the final timestep.

trajectories generated by ENMA and by the solver, demonstrating that ENMA's generated states are not only coherent but also physically meaningful.

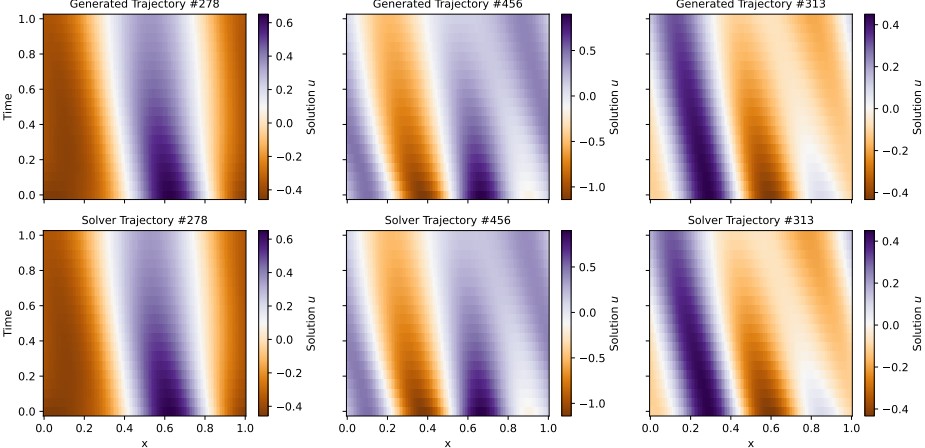

Figure 23: Comparison between trajectories generated by ENMA (top row) and the corresponding rollouts obtained from the PDE solver using the generated initial condition (bottom row). ENMA's predictions yield physically consistent rollouts.

### F.1.2 Dynamics forecasting with less compression

As discussed in section 4.3, ENMA exhibits lower performance on the Vorticity dataset compared to baseline models. We hypothesize that this is due to the aggressive latent compression, which limits the VAE's ability to capture fine-scale, high-frequency structures—ultimately constraining generation quality at decoding time. To investigate this, we evaluate ENMA under reduced compression settings. While the main experiments used a spatial compression factor of 4 for Vorticity, we report additional results on 2D datasets in the temporal conditioning setting, comparing against AVIT and Zebra with a relaxed compression factor.

**Results**    Table 18 shows results on Gray-Scott, Wave, and Vorticity under relaxed compression to match baseline settings. This setup addresses ENMA's lower performance on Vorticity observed in section 4.3, attributed to overly aggressive compression. ENMA outperforms AVIT and Zebra on Gray-Scott and Wave across both In-D and Out-D settings. On Vorticity, ENMA remains competitive,

Table 18: Comparison of model performance on Gray-Scott, Wave, and Vorticity under the temporal conditioning setting. Metrics in Relative MSE ($\downarrow$).

| Model | Gray-Scott | | Wave | | Vorticity | |
|---|---|---|---|---|---|---|
| | In-D | Out-D | In-D | Out-D | In-D | Out-D |
| AVIT | 4.26e-2 | 1.68e-1 | 1.57e-1 | 5.88e-1 | 1.76e-1 | 3.77e-1 |
| Zebra | 4.21e-2 | 1.82e-1 | 1.40e-1 | 3.15e-1 | **4.43e-2** | **2.23e-1** |
| ENMA | **2.23e-2** | **1.52e-1** | **7.00e-2** | **2.83e-1** | 5.21e-2 | 2.82e-1 |

closely matching Zebra. These results highlight ENMA's strong predictive performance under comparable constraints.

### F.1.3 Inference speed against generative approaches

One of the primary limitations of generative models compared to deterministic surrogates lies in their inference speed. Diffusion-based methods, for instance, require multiple iterative passes through a large model—typically a Transformer—to generate a single sample, resulting in high computational cost. Similarly, autoregressive (AR) approaches like (Serrano et al., 2025) generate tokens sequentially, which becomes especially expensive when modeling high-dimensional spatio-temporal data.

ENMA addresses these inefficiencies by combining the strengths of both paradigms. It adopts an AR framework that supports multi-token generation and leverages key-value caching for faster inference, while also benefiting from the expressiveness of generative methods through its use of Flow Matching. Crucially, rather than relying on a large model at each step, ENMA performs flow matching through a lightweight MLP, substantially reducing the computational overhead compared to traditional diffusion models.

**Results** We compare the inference time per sample for AR-DiT, Zebra, and ENMA in table 19. ENMA achieves the fastest inference across both the Combined and Vorticity datasets, with an average of 2s and 4.5s per sample, respectively. This represents a 2× speedup over AR-DiT and a 3× speedup over Zebra on Combined. Zebra particularly exhibits slower inference on the Vorticity dataset where it reaches 31s per sample. These results highlight the efficiency of ENMA's continuous latent autoregressive generation, which scales favorably compared to existing generative baselines.

Table 19: Inference speed comparison on the Combined and Vorticity datasets. Lower is better.

| Model | Combined | Vorticity |
|---|---|---|
| AR-DiT | 4s | 6.7s |
| Zebra | 6s | 31s |
| ENMA | **2s** | **4.5s** |

### F.1.4 Ablation Studies

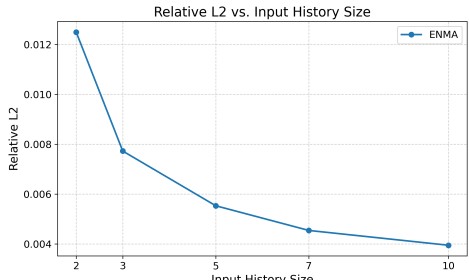

Figure 24: Relative L2 error vs. input history size. ENMA benefits from longer histories.

To better understand the behavior and flexibility of our generative framework, we conduct ablations examining ENMA's performance across three key dimensions: (i) sensitivity to the length of the input history, (ii) the number of autoregressive spatial steps $S$, and (iii) the number of Flow Matching (FM) steps during generation.

**Impact of Input History Length** In many real-world scenarios, the number of available historical observations varies, making it important for a model to adapt seamlessly to different history lengths. We evaluate ENMA's predictive performance as a function of the number of past frames provided as context and evaluate the predictions quality on the last 10 steps of the trajectory. As shown in fig. 24, the relative L2 error consistently decreases as the input

history length increases. This trend confirms that ENMA effectively leverages additional temporal context to refine its predictions. Notably, the model already achieves competitive performance with as few as 2–3 input frames and continues to improve as more information is made available. This highlights the model's capacity for conditional generalization across varying input regimes.

**Impact of the Number of Autoregressive Steps** The number of autoregressive steps $S$ directly impacts inference efficiency: fewer steps accelerate generation, but may reduce accuracy. In the case of the Combined equation, each spatial state of 128 points is compressed into 16 latent tokens. We evaluate ENMA's performance as we vary $S$ from 1 to 16 in fig. 25, where $S = 1$ corresponds to generating all tokens at once, and $S = 16$ to generating one token per step.

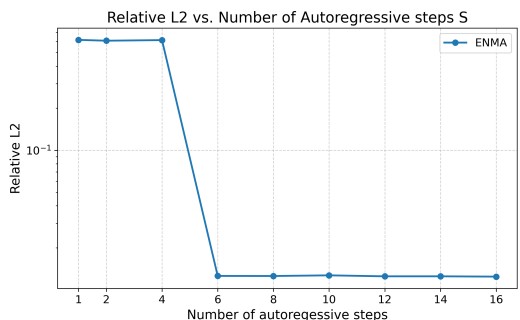

We observe that performance is poor when $S \leq 4$, but significantly improves at $S = 6$, after which it quickly plateaus. This behavior reflects the masking ratio used during training: the model is trained to reconstruct frames with

Figure 25: Relative L2 error as a function of the number of autoregressive steps $S$. Performance improves markedly around $S = 6$ and plateaus thereafter.

75%–100% of tokens masked. Hence, inference scenarios where a majority of tokens are already visible (i.e., large $S$ values) fall outside the model's training regime, which may limit gains from further increasing $S$. These results suggest that a modest number of autoregressive steps—around 6—balances speed and accuracy. Expanding the training schedule to include lower masking ratios could further improve performance for large $S$, but would increase training time due to the larger space of masking patterns.

**Impact of the Number of Flow Matching Steps** The number of flow matching (FM) steps governs the granularity of sampling from the learned conditional distribution at each autoregressive step. While more FM steps can, in principle, yield smoother and more accurate trajectories, they also increase inference cost—even if the MLP used for token decoding remains lightweight. In fig. 26, we evaluate ENMA's performance as a function of FM steps. We observe a sharp drop in relative L2 error between 2 and 5 steps, after which the performance plateaus. Beyond 10 steps, improvements are marginal or even slightly inconsistent. This indicates that ENMA captures most of the necessary detail with very few sampling steps.

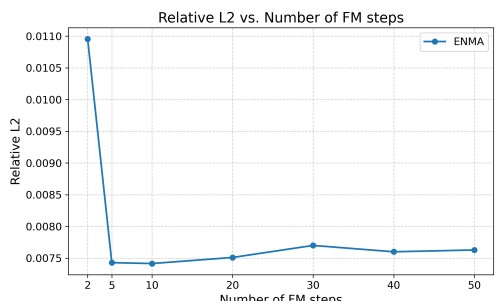

Figure 26: Relative L2 error versus number of flow matching steps per token. Performance quickly stabilizes after 5 steps.

From a practical standpoint, this suggests that using as few as 5 FM steps can offer an optimal trade-off between speed and accuracy, making ENMA efficient at inference without compromising quality.

**Latent Dynamics Modeling** We assess the impact of the training objective on the quality of latent trajectory generation by comparing **Flow Matching (ENMA)** against a **Diffusion**-based latent model and a fully **Deterministic** variant. As shown in table 20, flow matching yields the lowest relative MSE, outperforming both alternatives. While diffusion-based training theoretically offers strong generative capacity, we observe convergence instability and degraded accuracy in practice. The deterministic approach performs better than diffusion but remains inferior to flow matching, likely due to its inability to capture stochastic variability in the latent dynamics. Overall, these results highlight the robustness and efficiency of the flow-matching paradigm for continuous stochastic trajectory modeling in ENMA.

Table 20: Ablation of the training objective on the **Vorticity** dataset. Metrics are reported in Relative MSE on the test set.

| Training Objective | Relative MSE |
|---|---|
| Flow Matching (ENMA) | **0.0644** |
| Diffusion | 0.7579 |
| Deterministic | 0.0879 |

## F.2 Experiments on the encoder-decoder

### F.2.1 Additional Experiments on new datasets

We provide additional experiment on the encode-decoder of ENMA on a new dataset: *Cylinder Flow* (Pfaff et al., 2021). This dataset is inherently irregular and contains multiple output channels. The results are shown in table 21 and illustrate the superior performance of ENMA on this additional dataset.

**Setting** In this example, we used tokens with a spatial size of $8 \times 8$ and a feature dimension of $8$ for all baselines except OFormer, which does not perform spatial compression. ENMA uses a latent grid of size $32 \times 32$ with a feature dimension of $16$, and then applies two 'compress_space' and one 'compress_time' layers to reach the desired latent size. All other configurations follow those described in appendix E. All models are trained as autoencoders by minimizing the RMSE loss. The FNO architecture follows the same configuration as described in appendix E.2.3. Models were trained for 1,000 epochs on the reconstruction task and 250 epochs for the time-stepping experiment.

Table 21: **Reconstruction error** – Test results and compression rates. Metrics in Relative MSE. The compression rate reflects how much the latent representation is reduced compared to the input data. A compression rate of $\times 2$ indicates that the latent space is half the size of the input in terms of raw elements.

| $\downarrow \mathcal{X}_{\text{te}}$ | Dataset $\rightarrow$ | Cylinder Flow | | |
|---|---|---|---|---|
| | Model $\downarrow$ | *Reconstruction* | *Time-stepping* | *Compression rate* |
| | OFormer | 1.71e-1 | 7.96e-1 | $\times 0.38$ |
| | GINO | 8.04e-1 | 1.56 | $\times 10$ |
| 100% | AROMA | 9.38e-2 | 6.50e-1 | $\times 10$ |
| | CORAL | 3.39e-1 | 5.35e-1 | $\times 10$ |
| | ENMA | **8.78e-2** | **1.71e-1** | $\times 20$ |
| | OFormer | 1.75e-1 | 7.96e-1 | - |
| | GINO | 8.04e-1 | 1.55 | - |
| 50% | AROMA | 1.02e-1 | 6.63e-1 | - |
| | CORAL | 3.40e-1 | 5.40e-1 | - |
| | ENMA | **9.06e-2** | **1.75e-1** | - |
| | OFormer | 1.92e-1 | 8.08e-1 | - |
| | GINO | 8.03e-1 | 1.54 | - |
| 20% | AROMA | 1.26e-1 | 7.22e-1 | - |
| | CORAL | 3.43e-1 | 5.50e-1 | - |
| | ENMA | **9.98e-2** | **1.82e-1** | - |

**Results** These additional results on a widely known dataset further validate the conclusions drawn in section 4.2. ENMA outperforms the baselines on the reconstruction task while achieving twice the compression. Moreover, its tokens capture informative features that enable strong performance on the time-stepping task.

### F.2.2 OOD encoding on 10% of the input grid

We evaluate the performance of the auto-encoder in an out-of-distribution (OOD) reconstruction setting by reducing the input grid size to $\pi = 10\%$ of the original spatial grid. This requires models to reconstruct the full field from very sparse observations—only 12 points on the Advection dataset and 409 points on the Vorticity dataset. Notably, such extreme sparsity was not encountered during training, where input sampling ratios ranged from $\pi = 20\%$ to $\pi = 100\%$.

Table 22: **Reconstruction error** using $\pi = 10\%$ of the initial grid Test results. Metrics in Relative MSE.

| $\downarrow \mathcal{X}_{\text{te}}$ | Dataset → 
 Model ↓ | Advection 
 *Reconstruction* | Vorticity 
 *Reconstruction* | Cylinder Flow 
 *Reconstruction* |
|---|---|---|---|---|
| | OFormer | 5.82e-1 | 1.00e+0 | 2.19e-1 |
| | GINO | 2.63e-1 | 6.46e-1 | 8.05e-1 |
| $\pi = 10\%$ | AROMA | 4.69e-1 | 3.28e-1 | 1.51e-1 |
| | CORAL | 1.13e+0 | 1.38e+0 | 3.58e-1 |
| | ENMA | **2.45e-1** | **2.78e-1** | **1.15e-1** |

**Results** This reconstruction experiment demonstrates that ENMA maintains superior reconstruction quality even when provided with extremely sparse input fields. Among the baselines, AROMA and GINO also exhibit robust performance under such challenging conditions. In contrast, OFormer and CORAL fail to reconstruct the physical field, highlighting their limited generalization in low-data regimes.

### F.2.3 Super-resolution

In this section, we assess the models' ability to generalize to finer spatial grids. For the Advection dataset, we evaluate reconstructions starting from input subsamplings of $\pi = 20\%, 50\%, 100\%$, and query the model on denser grids with $\pi_{\text{sr}} = 200\%, 400\%, 800\%$ of the original resolution. On the Vorticity dataset, we use the same input subsamplings but perform super-resolution only at $\pi_{\text{sr}} = 200\%$ due to data availability constraints.

For comparison, we also report reconstruction results at $\pi_{\text{sr}} = 100\%$, as in table 1, and present the full results in table 23. The table is structured as follows: rows correspond to the input grid ratio $\pi$, matching the setup of table 1, while columns indicate the relative query grid size $\pi_{\text{sr}}$. For instance, on the Advection dataset with an original grid size of 128, the second row ($\pi = 50\%$) corresponds to 64 input points, and the third column ($\pi_{\text{sr}} = 400\%$) corresponds to querying 512 points. Other rows and columns follow the same interpretation.

**Results** As shown in table 23, all models exhibit performance degradation under the super-resolution setting. Despite this, ENMA consistently outperforms other encoder–decoder architectures across all resolutions. CORAL demonstrates stable performance as output resolution increases, while OFormer struggles significantly when queried on unseen grids for both datasets. ENMA and AROMA show similar trends as the super-resolution difficulty increases; however, AROMA's performance degrades more rapidly with lower input grid densities. Overall, ENMA's auto-encoder proves to be the most robust, both to variations in input sparsity and to changes in query resolution.

Table 23: **Reconstruction error** on super resolution task. Test results. Metrics in Relative MSE.

| ↓ $\mathcal{X}_{\text{te}}$ | Dataset → | Advection | | | | Vorticity | |
| | $\pi_{sr}$ → 
 **Model ↓** | *100%* | *200%* | *400%* | *800%* | *100%* | *200%* |
|---|---|---|---|---|---|---|---|
| $\pi = 100\%$ | OFormer | 1.70e-1 | 5.20e+0 | 5.19e+0 | 5.19e+0 | 9.99e-1 | 1.00e+0 |
| | GINO | 5.74e-2 | 7.77e-2 | 8.83e-2 | 9.43e-2 | 5.63e-1 | 5.76e-1 |
| | AROMA | 5.41e-3 | 3.78e-2 | 5.62e-2 | 6.54e-2 | 1.45e-1 | 1.71e-1 |
| | CORAL | 1.34e-2 | 4.00e-2 | 5.76e-2 | 6.66e-2 | 4.50e-1 | 4.55e-1 |
| | ENMA | **1.83e-3** | **3.71e-2** | **5.56e-2** | **6.49e-2** | **9.20e-2** | **1.36e-1** |
| $\pi = 50\%$ | OFormer | 1.79e-1 | 5.20e+0 | 5.19e+0 | 5.18e+0 | 9.99e-1 | 1.00e+0 |
| | GINO | 6.64e-2 | 8.38e-2 | 9.37e-2 | 9.95e-2 | 5.69e-1 | 5.82e-1 |
| | AROMA | 2.34e-2 | 4.44e-2 | 6.09e-2 | 6.94e-2 | 1.64e-1 | 1.89e-1 |
| | CORAL | 7.57e-2 | 8.80e-2 | 9.96e-2 | 1.06e-1 | 4.93e-1 | 4.98e-1 |
| | ENMA | **4.60e-3** | **3.74e-2** | **5.59e-2** | **6.51e-2** | **9.90e-2** | **1.41e-1** |
| $\pi = 20\%$ | OFormer | 2.50e-1 | 5.18e+0 | 5.17e+0 | 5.16e+0 | 9.99e-1 | 1.00e+0 |
| | GINO | 9.13e-2 | 1.04e-1 | 1.12e-1 | 1.17e-1 | 5.90e-1 | 6.01e-1 |
| | AROMA | 1.67e-1 | 1.72e-1 | 1.78e-1 | 1.81e-1 | 2.29e-1 | 2.45e-1 |
| | CORAL | 4.77e-1 | 4.79e-1 | 4.82e-1 | 4.84e-1 | 7.59e-1 | 7.62e-1 |
| | ENMA | **3.05e-2** | **4.94e-2** | **6.49e-2** | **7.31e-2** | **1.37e-1** | **1.69e-1** |

### F.2.4 Ablation studies

We conduct two additional ablation studies to evaluate specific architectural components of the encoder–decoder framework: (i) the effect of using a geometry-aware attention bias (table 24), (ii) the impact of the additional temporal causal compression, and (iii) the impact of the time-stepping process used in the decoder (table 26).

**Geometry-aware attention bias.** Table 24 reports the reconstruction error when ENMA is trained with and without the geometry-aware attention bias described in appendix D.1. This positional bias encourages spatial locality in attention by penalizing interactions between distant query–key pairs. The results show consistent improvements across all input sparsity levels, with relative gains ranging from 25% to 40%. We provide additional analysis and some visualizations in appendix F.2.5.

Table 24: **Reconstruction error** with and without geometry-aware attention bias. Metrics are Relative MSE. Relative improvement is reported in parentheses.

| $\pi$ | **Input Ratio** | **Model Variant** | **Reconstruction ↓** |
|---|---|---|---|
| 100% | Full | w/o bias 
 w/ bias | 3.09e-3 
 **1.83e-3** (-40%) |
| 50% | Half | w/o bias 
 w/ bias | 6.84e-3 
 **4.60e-3** (-33%) |
| 20% | Sparse | w/o bias 
 w/ bias | 4.13e-2 
 **3.05e-2** (-26%) |

**Causal vs Non-causal encoder** We compare a causal convolution module in the encoder with a non causal encoder, to assess its impact on temporal feature representation. Table 25 presents an ablation of the causal component in ENMA. Compared to the non causal autoencoder, the causal version adds temporal compression, reducing the number of tokens by half. Despite this, it matches or outperforms the non causal version—especially in low-data settings— indicating that the causal layer helps in providing more informative tokens.

Table 25: **Reconstruction error**: Ablation of the Causal component on the Advection dataset. Metrics in Relative MSE on the test set (please refer to table 1 for the sampling operation).

| $\pi$ | Input Ratio | Model Variant | Reconstruction $\downarrow$ |
|---|---|---|---|
| 100% | Full | Causal | 5.16e-3 |
| | | Non causal | **4.55e-3** |
| 50% | Half | Causal | 6.98e-3 |
| | | Non causal | **6.02e-3**aa |
| 20% | Sparse | Causal | **1.66e-2** |
| | | Non causal | 2.01e-2 |

**Time-stepping architecture.** We further examine the impact of the time-stepping module by replacing the FNO used in ENMA with a 4-layer U-Net. Both models use identical lifting and projection layers to match the latent token dimensions. The U-Net adopts convolutional layers with kernel size 3, stride 1, and padding 1.

Table 26 summarizes the reconstruction performance on the Advection dataset across three input sparsity levels. We observe that ENMA maintains superior accuracy under both architectures. While the U-Net variant exhibits slightly degraded performance compared to the FNO, it remains competitive and stable. A dash (–) indicates that the model diverged and produced NaNs during training.

These results confirm the robustness of ENMA across different architectural choices for the time-stepping component and further emphasize the benefits of the geometry-aware attention mechanism.

Table 26: **Reconstruction error** for different time-stepping processes (FNO vs U-Net) on the Advection dataset. Metrics are Relative MSE.

| $\pi$ | Model | FNO | U-Net |
|---|---|---|---|
| 100% | OFormer | 1.11e+0 | 1.54e+0 |
| | GINO | 7.89e-1 | 7.52e-1 |
| | AROMA | 2.23e-1 | 6.43e-1 |
| | CORAL | 9.64e-1 | – |
| | ENMA | **1.64e-1** | **3.51e-1** |
| 50% | OFormer | 1.11e+0 | 1.04e+0 |
| | GINO | 7.87e-1 | 7.50e-1 |
| | AROMA | 2.29e-1 | 6.49e-1 |
| | CORAL | 9.74e-1 | – |
| | ENMA | **1.72e-1** | **3.58e-1** |
| 20% | OFormer | 1.13e+0 | 1.03e+0 |
| | GINO | 7.96e-1 | 7.53e-1 |
| | AROMA | 3.21e-1 | 7.32e-1 |
| | CORAL | 1.06e+0 | – |
| | ENMA | **3.13e-1** | **4.15e-1** |

### F.2.5 Geometry-Aware Attention Analysis

We visualize the attention mechanism in ENMA's encoder-decoder using the Advection dataset to better understand the effect of the geometry-aware attention bias.

**Attention Bias** Figure 28 displays the geometry-aware bias $B$ introduced in Section D.1. Each plot corresponds to one attention head, with rows representing different input sizes (128, 64, and 25 points from top to bottom) and columns corresponding to different heads. Bias values near zero (white) indicate minimal distance between query and key coordinates, promoting attention, while larger distances produce strongly negative biases (dark blue), which suppress it. Because the input grid is irregular, attention patterns in the second and third rows are not strictly diagonal. Additionally, since the intermediate coordinates are learned (see fig. 27), the model is not constrained to maintain a perfectly regular grid, which further contributes to these deviations. Notably, the later attention heads appear to penalize distant tokens more strongly—an effect explained by the learned head-specific scaling factors, following the approach of Press et al. (2022).

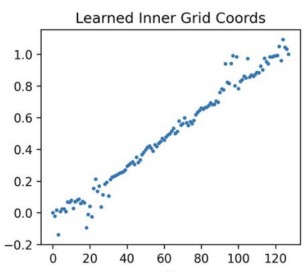

Figure 27: Learned coordinates of the intermediate regular grid.

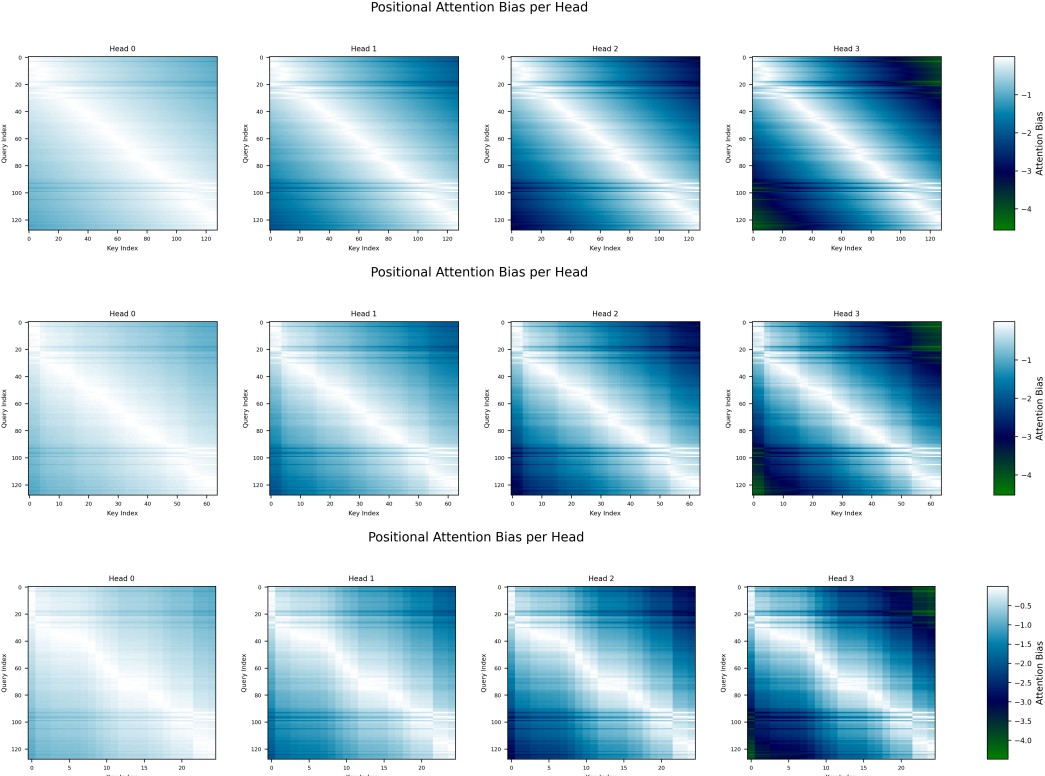

Figure 28: Geometry-aware attention bias $B$ for each attention head, shown for 128 (top), 64 (middle), and 25 (bottom) input points.

**Attention Scores** Figures 29 and 30 present the actual attention scores produced using the geometry-aware bias. These visualizations confirm that attention is concentrated on nearby points, enforcing a spatially local inductive bias during interpolation. We provide visualization for the advection dataset fig. 29 and vorticity fig. 30.

These results demonstrate that the attention mechanism learns to prioritize spatially proximate inputs, even under varying levels of spatial sparsity.

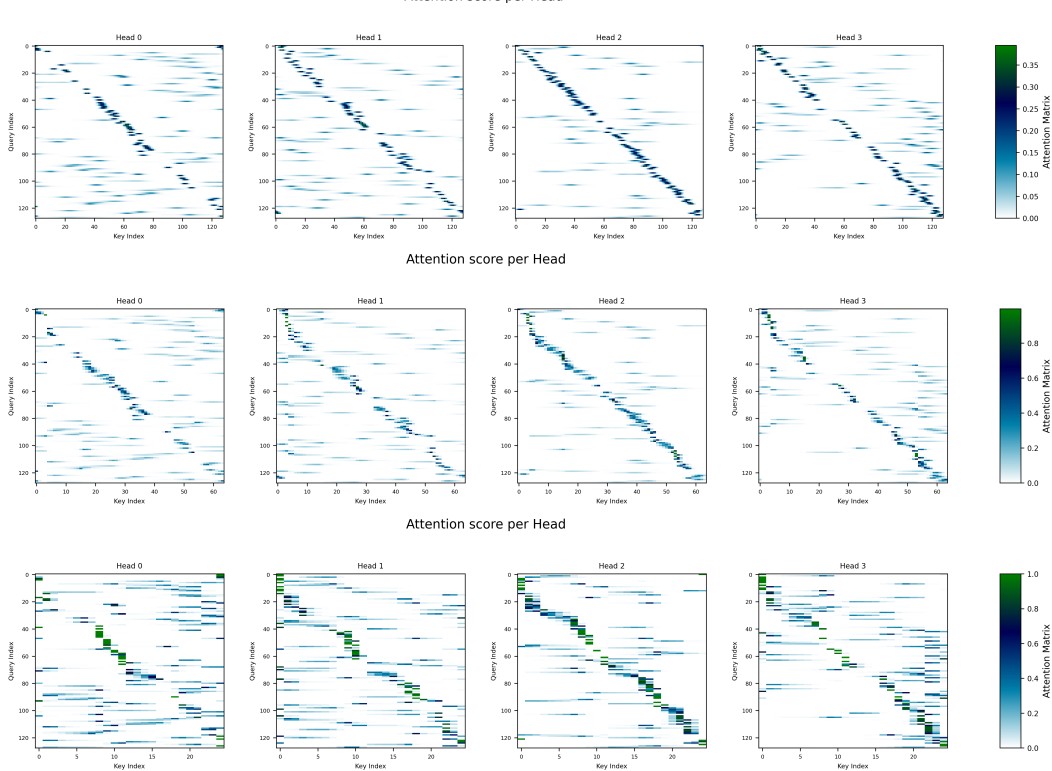

Figure 29: Attention scores per head using the geometry-aware bias for $128$ (top), $64$ (middle), and $25$ (bottom) input points.

**Attention visualization in the physical space** We visualize the final attention weights in physical space in fig. 30. For a selected timestep of a given trajectory, we choose a set of spatial tokens (left panel) and display their corresponding attention scores across the physical domain (middle panel). The resulting patterns clearly demonstrate that the cross-attention mechanism, guided by the geometry-aware bias, preserves locality by assigning high attention weights to nearby regions—effectively focusing on spatially relevant information.

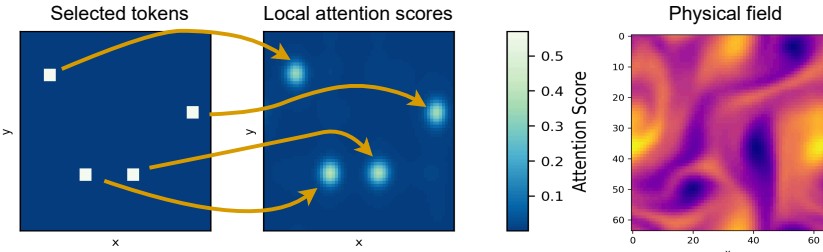

Figure 30: Attention scores on the vorticity dataset, on spatially selected tokens (left) at a given timestep ($t = 15$). The resulting attention score on the output grid keeps the local behavior (middle). Attention score are averaged across heads for visualization. The left figure is the physical field considered.

### F.2.6 Visualization in the Token Space

Figure 31 shows the latent token representations on the Advection dataset for varying input sparsity (128, 64, and 25 points from top to bottom). The final column displays the corresponding ground-truth physical trajectory.

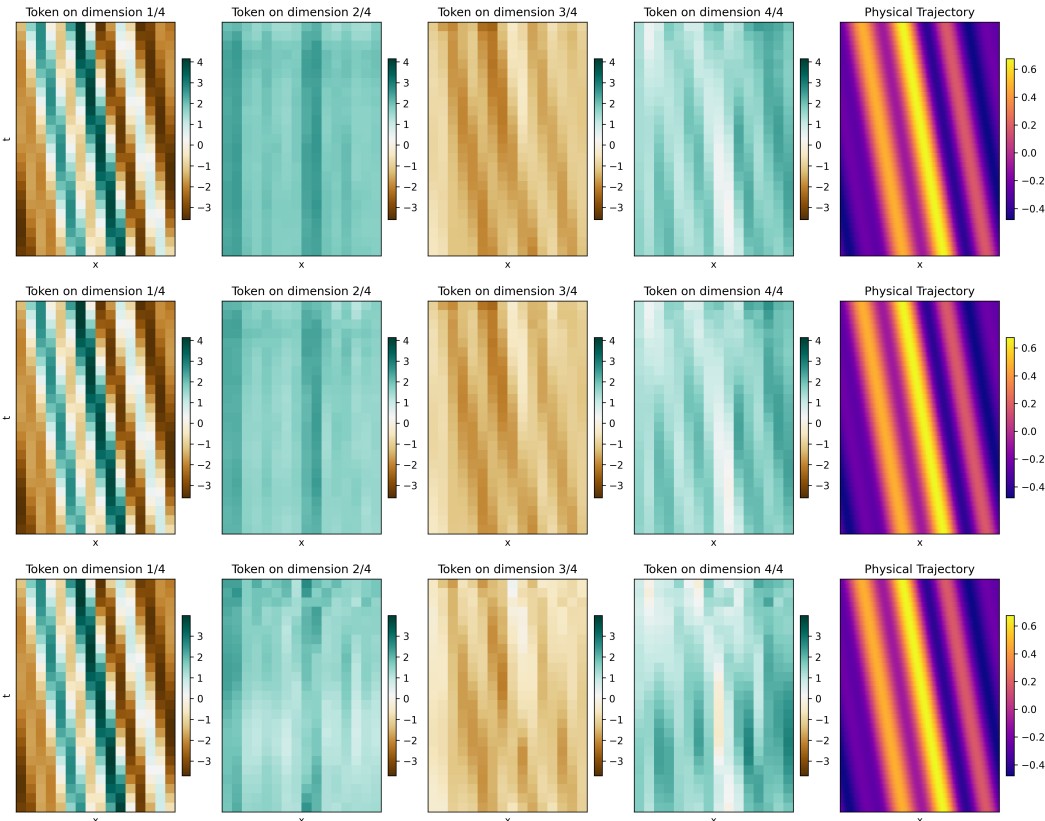

Figure 31: Token representation when using 128 (top), 64 (middle), and 25 (bottom) input points.

The tokens effectively capture the dynamics in a compressed form, demonstrating the encoder's strong inductive bias. Notably, the first latent dimension remains stable across input resolutions, encoding coarse global structure. In contrast, the remaining dimensions vary with input density, indicating progressive refinement of local details—a sign of partial disentanglement across token dimensions. This behavior supports the model's ability to encode structure hierarchically and adaptively under compression.

**Intermediate Fields in the Encoder–Decoder** Figure 32 visualizes the full trajectory of latent transformation within ENMA's encoder–decoder. Each row corresponds to a processing stage, averaged across channels.

Starting from a sparse, irregular input field (top row, $\pi = 20\%$), the first cross-attention layer projects the observations onto a learned latent grid (row 2), effectively interpolating the missing spatial points. The causal CNN then encodes the trajectory into a compact token space (row 3), capturing the trajectory's temporal evolution in a lower-dimensional representation. These tokens are subsequently upsampled (row 4) and decoded onto the full spatial grid (row 5). The final reconstruction is smooth and well-aligned with the underlying dynamics, highlighting the effectiveness of ENMA's encoder–decoder architecture. Minor artifacts observed in intermediate stages—likely caused by learned positional embeddings (Yang et al., 2024)—are effectively corrected during decoding.

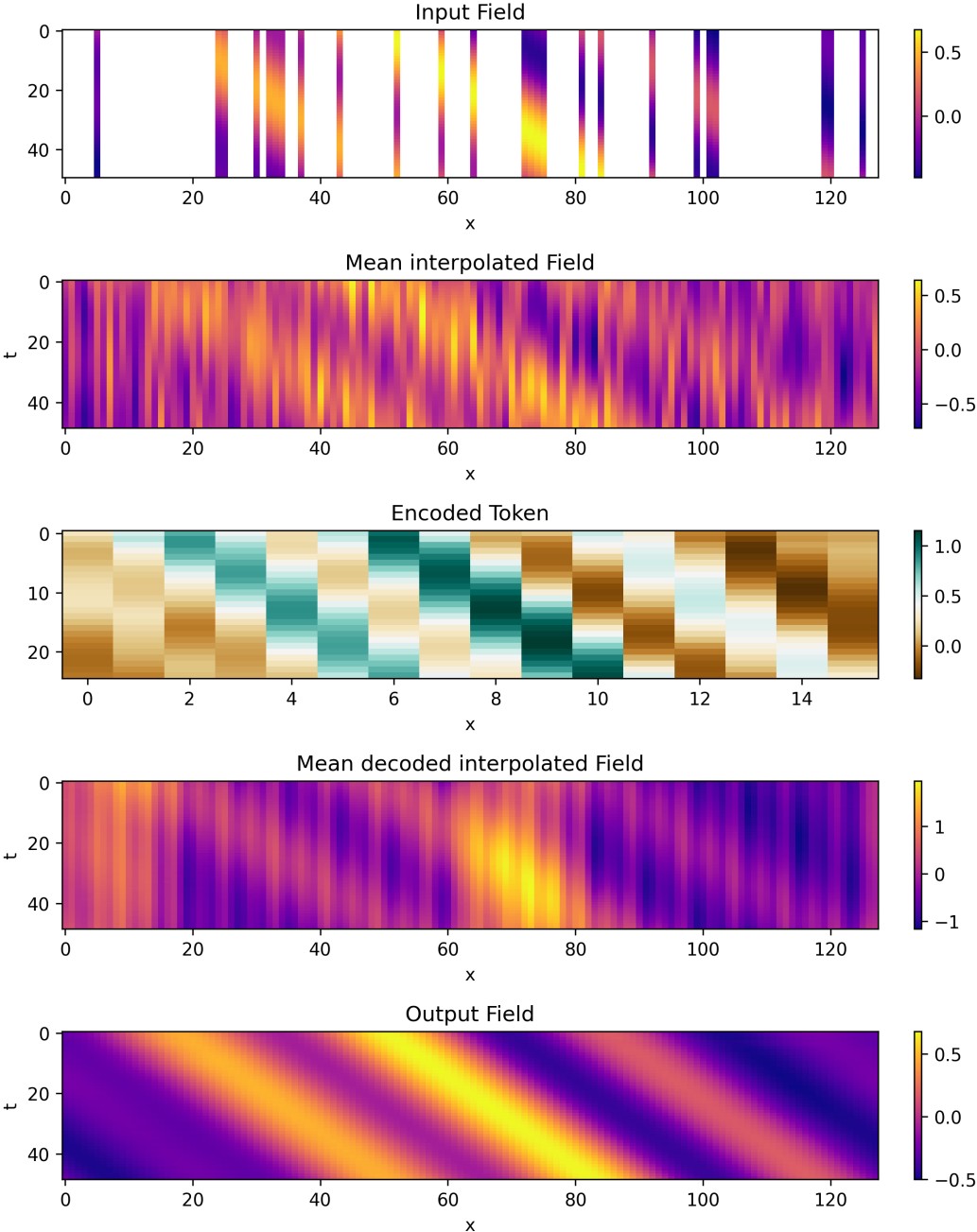

Figure 32: Intermediate fields in the encoder/decoder blocks: interpolation (rows 1–2), compression (rows 2–3), upsampling (rows 3–4), and reconstruction (rows 4–5).

# G Visualization

## G.1 Combined Equation

Figure 33 and Figure 34 provide qualitative results on the Combined equation, showcasing ENMA's accuracy on in-distribution inputs and its generalization capability to out-of-distribution scenarios.

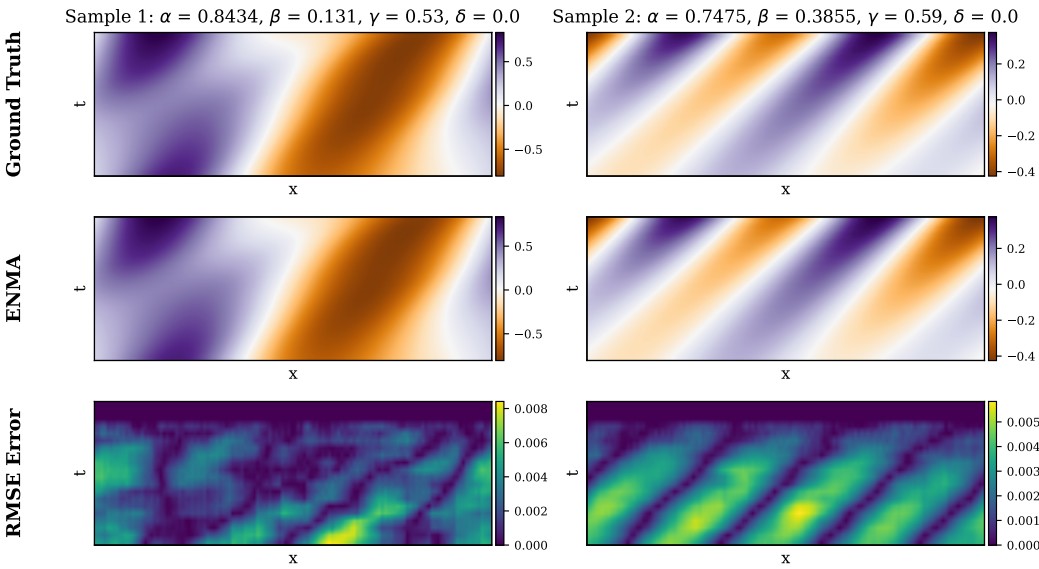

Figure 33: Qualitative comparison between ENMA prediction and ground truth for in-distribution examples from the Combined.

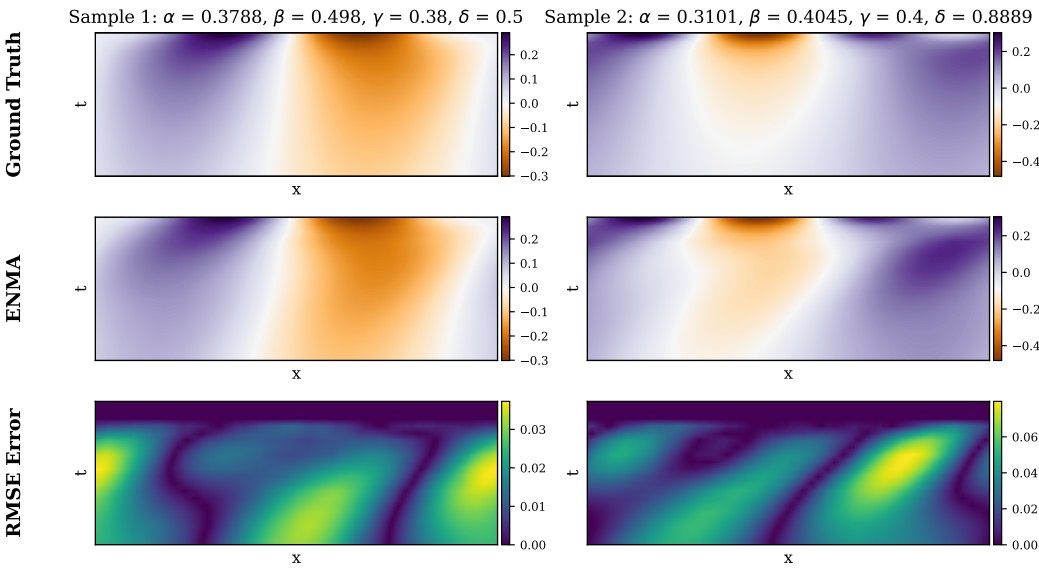

Figure 34: Qualitative comparison between ENMA prediction and ground truth for an OOD example from the Combined.

## G.2 Advection Equation

Figure 35 and Figure 36 provide qualitative results on the Advection equation. These plots highlight ENMA's accurate forecasting on in-distribution samples and its ability to generalize to out-of-distribution scenarios.

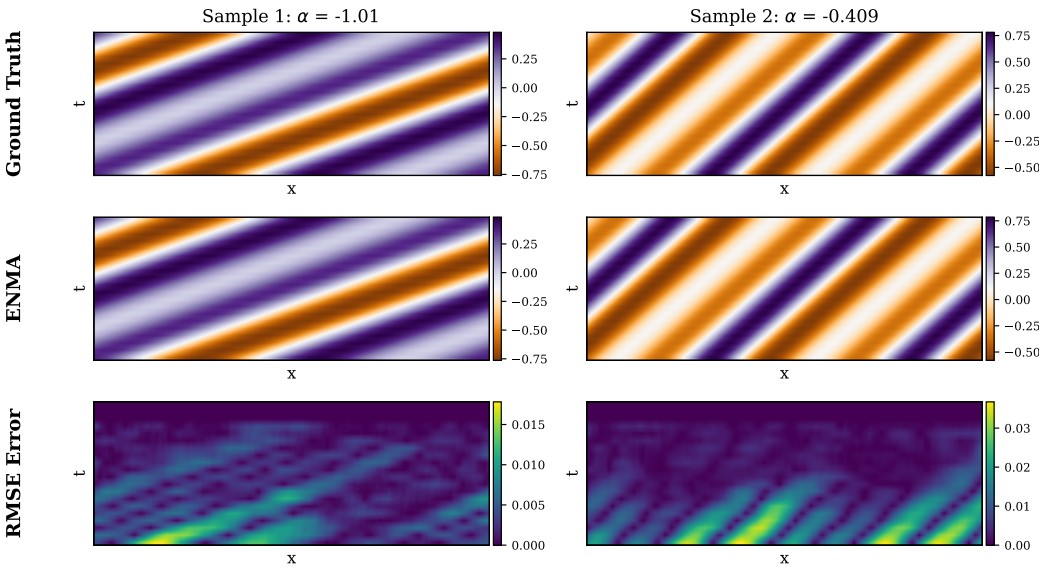

Figure 35: Qualitative comparison between ENMA prediction and ground truth for in-distribution examples from the Advection dataset.

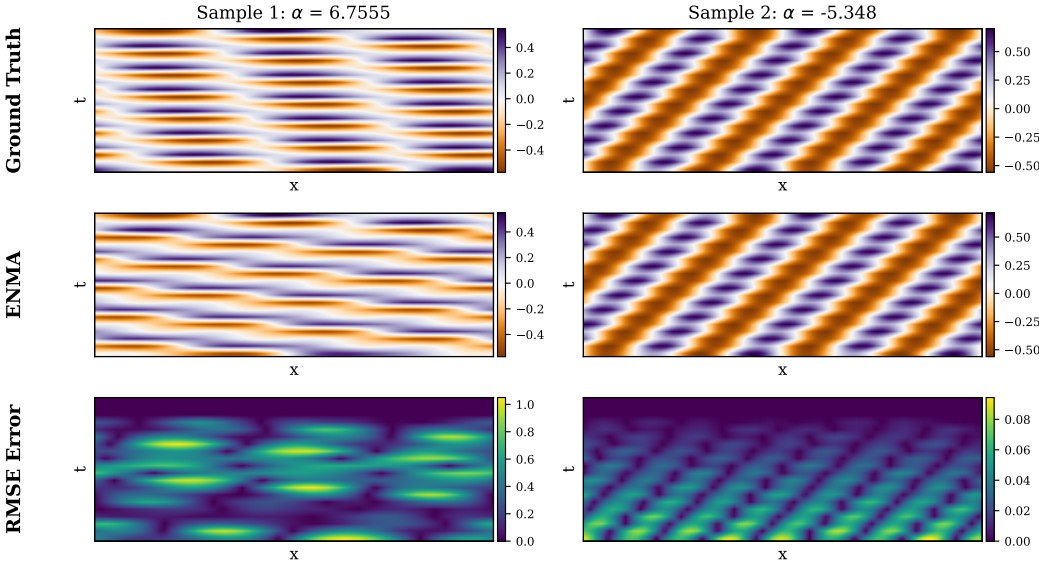

Figure 36: Qualitative comparison between ENMA prediction and ground truth for an OOD example from the Advection dataset.

### G.3  Gray-Scott Equation

Figure 37, Figure 38, and Figure 39 present qualitative comparisons between ENMA and ground truth on the Gray-Scott equation. ENMA accurately reconstructs complex spatiotemporal patterns on in-distribution samples and demonstrates good qualitative performance on out-of-distribution parameters.

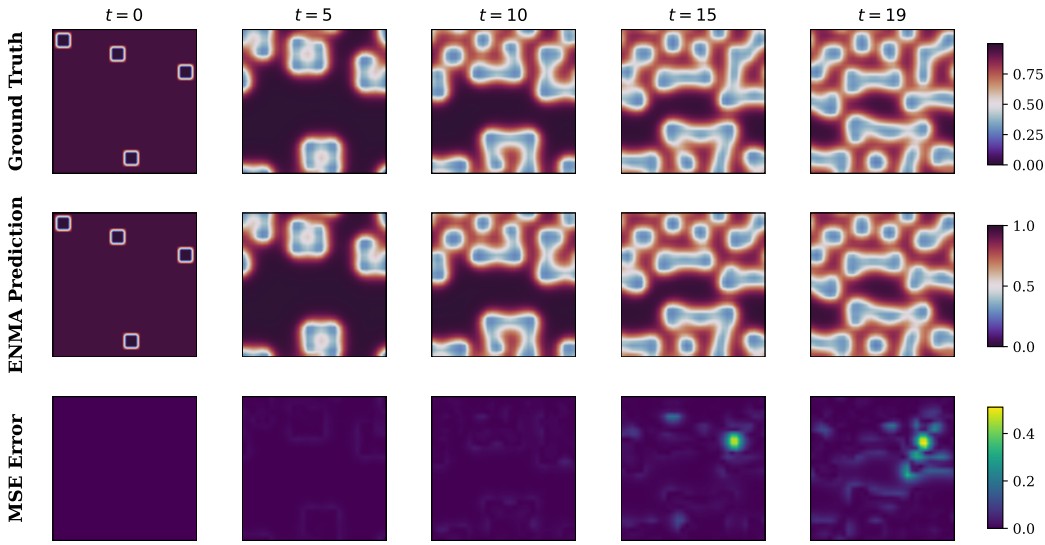

Figure 37: Qualitative comparison between ENMA prediction and ground truth for an in-distribution sample from the Gray-Scott dataset ($F = 0.0323$, $k = 0.0606$).

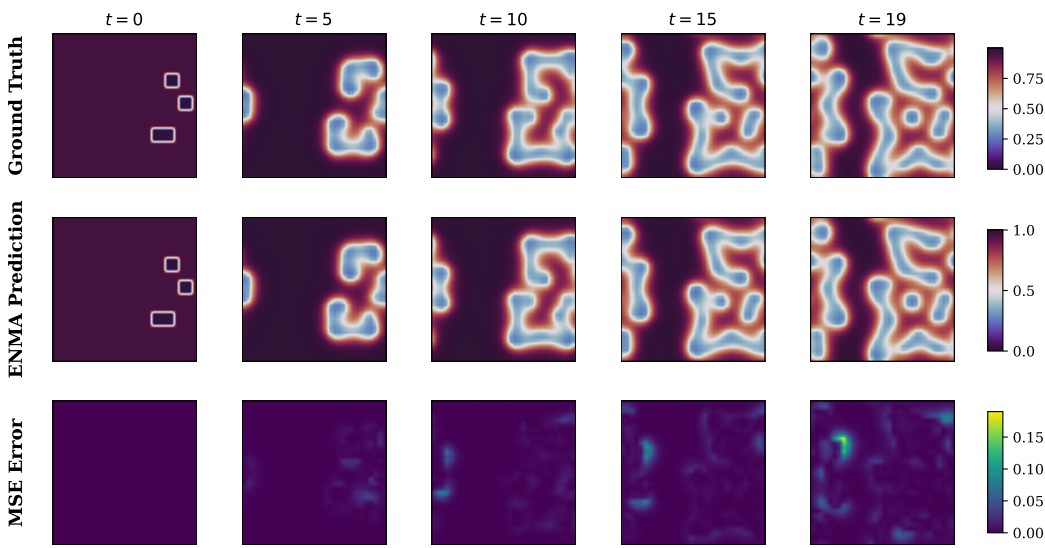

Figure 38: Another in-distribution Gray-Scott example showing the agreement between ENMA prediction and ground truth ($F = 0.0316$, $k = 0.0597$).

ENMA vs Ground Truth Parameters: $F = 0.0467, k = 0.058$

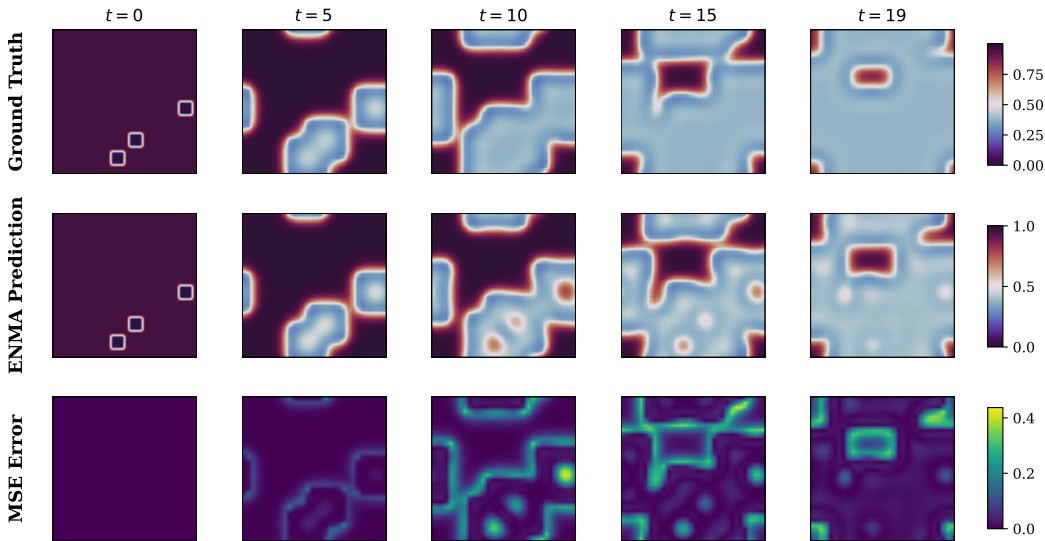

Figure 39: Out-of-distribution (OOD) generalization for the Gray-Scott equation. ENMA prediction remains consistent despite being evaluated at unseen parameters ($F = 0.0467$, $k = 0.058$).

## G.4 Wave Equation

Figure 40, Figure 41, and Figure 42 present qualitative comparisons for the Wave equation. ENMA accurately captures wavefront propagation in in-distribution scenarios and generalizes well to out-of-distribution conditions.

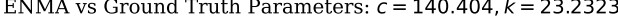

ENMA vs Ground Truth Parameters: $c = 140.404, k = 23.2323$

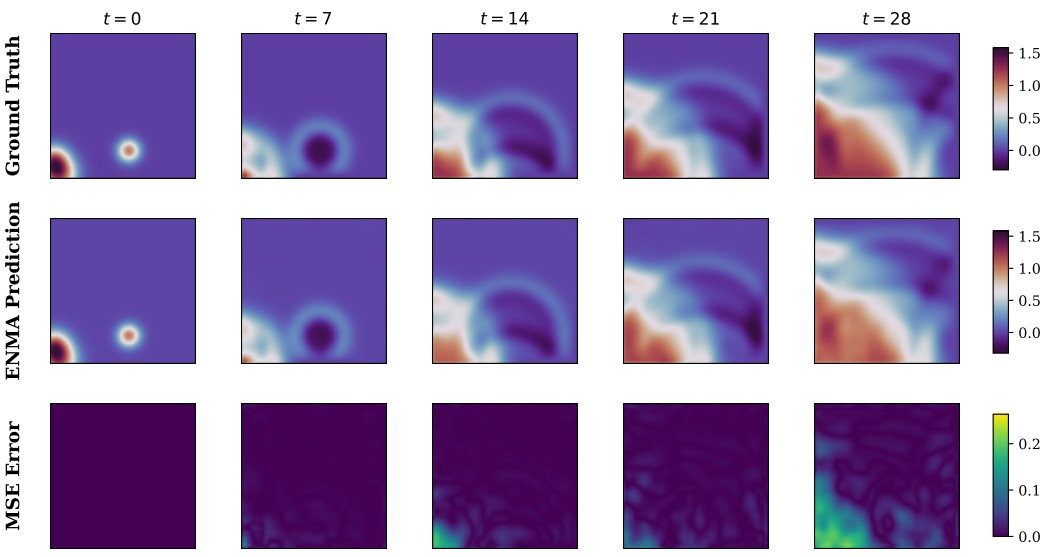

Figure 40: Qualitative comparison between ENMA prediction and ground truth for an in-distribution sample from the Wave dataset ($c = 140.404$, $k = 23.2323$).

ENMA vs Ground Truth Parameters: $c = 144.4444, k = 21.2121$

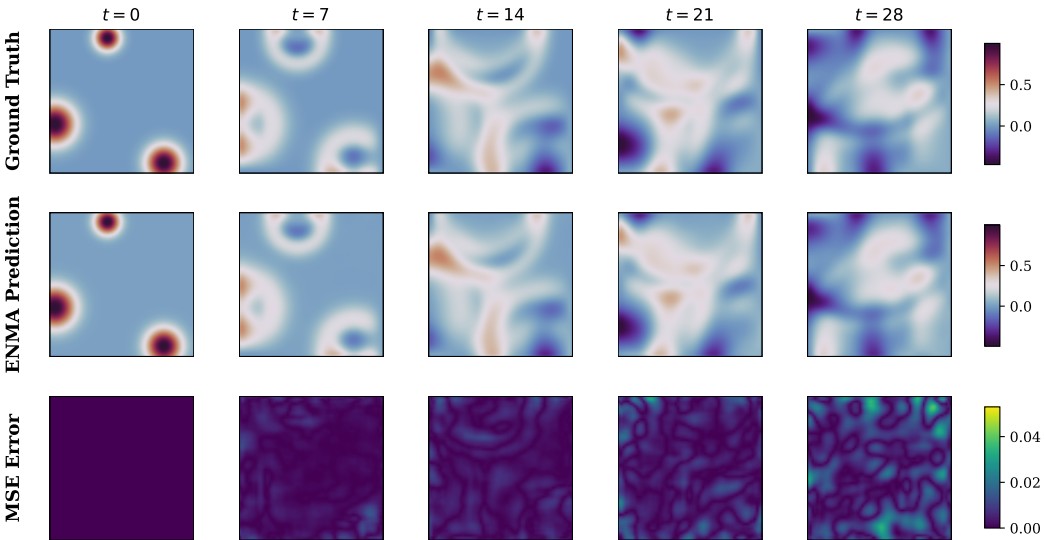

Figure 41: Second in-distribution sample from the Wave dataset ($c = 144.4444$, $k = 21.2121$), showing ENMA's prediction, ground truth, and the corresponding MSE error.

ENMA vs Ground Truth Parameters: $c = 500.0, k = 52.5$

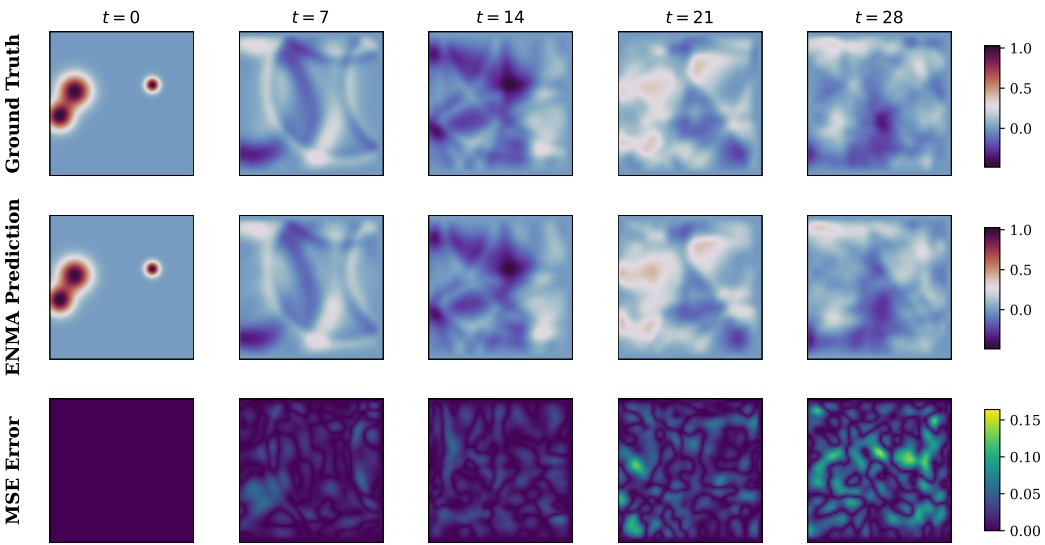

Figure 42: Out-of-distribution example from the Wave dataset ($c = 500.0$, $k = 52.5$), demonstrating ENMA's generalization capabilities beyond the training regime.

## G.5 Vorticity Equation

Figure 43, Figure 44, and Figure 45 show qualitative comparisons for the Vorticity equation. ENMA reliably captures fluid dynamics behavior on in-distribution samples and maintains accuracy in out-of-distribution regimes.

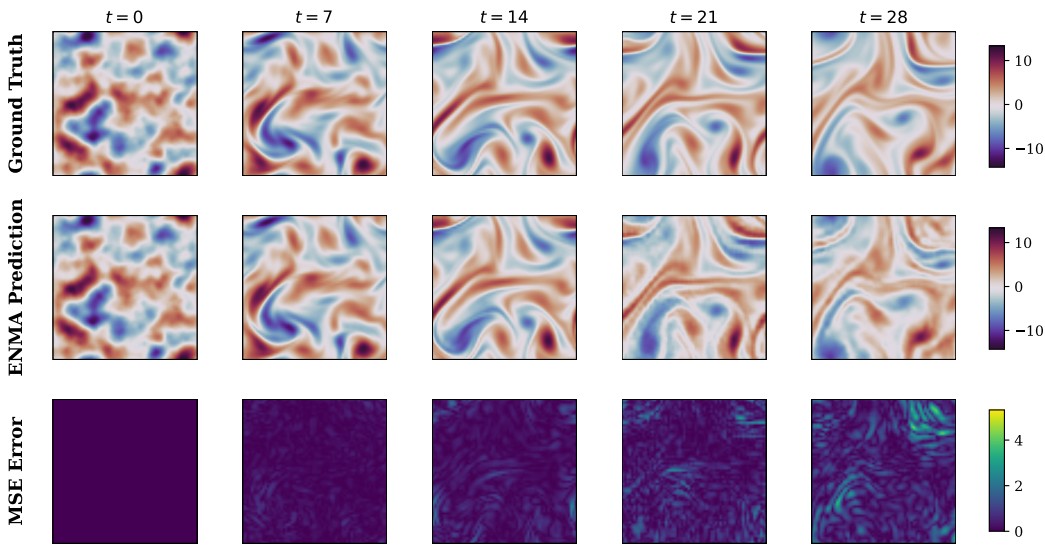

Figure 43: Qualitative comparison between ENMA prediction and ground truth for an in-distribution sample from the Vorticity dataset ($\nu = 0.0019$).

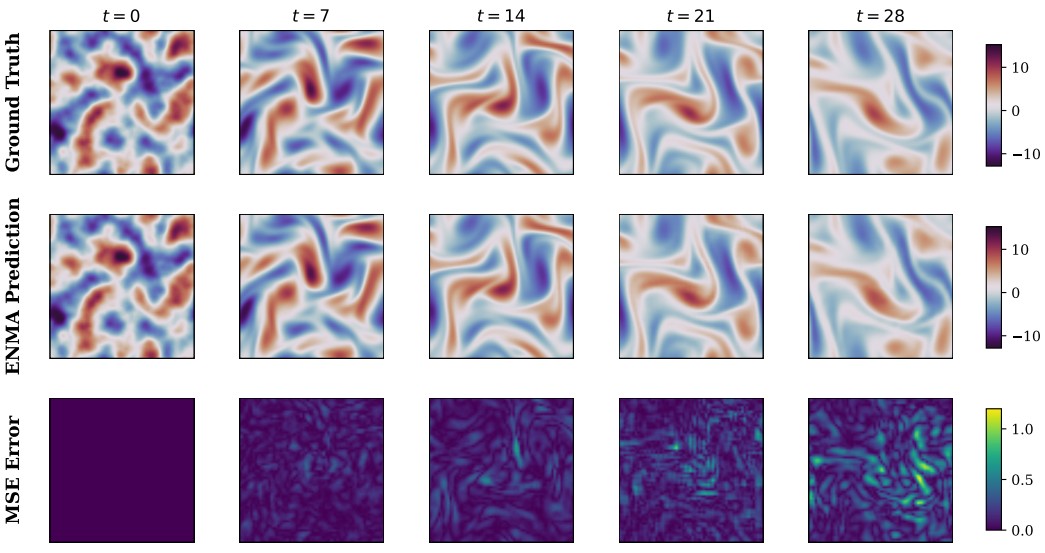

Figure 44: Second in-distribution sample from the Vorticity dataset ($\nu = 0.0048$), illustrating ENMA's ability to reproduce complex vortex structures.

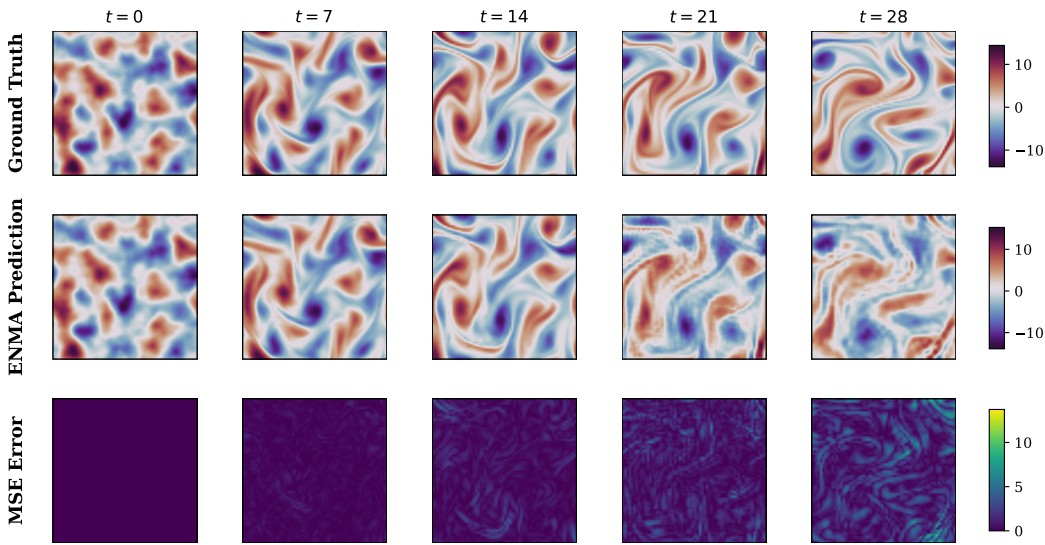

Figure 45: Out-of-distribution example from the Vorticity dataset ($\nu = 0.0007$), highlighting ENMA's robustness in extrapolating vortex dynamics.

