# OpenReview forum: "ENMA: Tokenwise Autoregression for Continuous Neural PDE Operators"
_NeurIPS.cc/2025/Conference — NeurIPS 2025 spotlight_

### Official Review · Reviewer_aeqi · 2025-06-23

**Clarity:** 2
**Significance:** 3
**Originality:** 3
**Rating:** 5
**Confidence:** 3

**Summary:**

This paper presents ENMA, a generative neural operator for time-dependent parametric PDEs that performs continuous autoregressive generation in latent space. Unlike existing approaches that rely on discrete tokenization or full-frame diffusion, ENMA uses a two-stage generation process: a causal transformer predicts future latent states, followed by masked spatial transformers that generate tokens using flow matching. The method can perform in-context learning by conditioning on historical states or auxiliary trajectories, allowing adaptation to new PDE regimes without retraining.

**Questions:**

1. The encoder uses cross-attention to interpolate irregularly sampled inputs onto a structured grid, but it’s unclear why this is necessary. Why wouldn’t simple masking of missing spatial points suffice? From my understanding, masking does not inherently result in information loss. The cross-attention mechanism adds architectural complexity—could the authors clarify what is gained from this choice?

2. Equation (4) suggests that the model outputs predictions on the locations as input. Yet the paper claims the model can evaluate on arbitrary spatial grids. Please revise or clarify.

3. The sentence “A compression rate of ×2 indicates that the latent space is half the size of the input in terms of raw elements” is somewhat vague. Does this refer to the number of spatial tokens, total elements across space and time, or some other metric? A more precise definition would help interpret the reported results.

4. The paper emphasizes that ENMA operates with continuous latent tokens, but it’s not clear what this means in contrast to standard dense embeddings. How is this different from using a ViT encoder, where the inputs are mapped to latent embeddings without any vector quantization? If such embeddings also qualify as continuous tokens, what is the specific advantage or novelty of ENMA’s formulation in terms of continuous latent space/tokens?

5. Appendix F contains valuable architectural analysis and ablation results, yet the main paper only refers to it in passing (e.g., “Additional analysis... in Appendix F”). I recommend the authors briefly summarize the key insights and conclusions from Appendix F in the main text to give readers a better sense of what is covered and why it matters.

**Ethical Concerns:**

["NO or VERY MINOR ethics concerns only"]

**Final Justification:**

Through the rebuttal process, my main concerns have been addressed, and I will increase my score.

**Limitations:**

Yes

**Quality:**

2

**Strengths And Weaknesses:**

**Strength:**

- The proposed approach offers a promising alternative for PDE operator learning, particularly in settings requiring uncertainty quantification and in-context adaptability.

- The experimental validation is relative comprehensive, including evaluations on both standard and challenging scenarios, such as out-of-distribution generalization and sparse input regimes.

**Weakness:**

- The paper feels rushed in several places. For example, in Figure 1, the text "Continuous VAE tokenizer" is not centered in its box, and section titles are inconsistently formatted (e.g., Section 3.1 vs. Section 4.2).

-  The method section lacks clear structure, with architectural design, training procedure, and inference details all mixed together. This makes it difficult to understand the overall pipeline. I strongly suggest reorganizing this section into distinct parts to improve clarity.

- While the proposed framework is sensible, it largely reads as a combination of existing techniques—causal transformers, masked spatial decoding, and flow matching. There’s limited discussion or ablation to isolate what’s truly novel or impactful in this particular integration. As a result, the contribution feels incremental rather than innovative.

- One major concern is the choice of benchmarks. The model is not evaluated on standard public datasets such as PDEBench or PDEArena, which include multiscale and stiff systems and present challenges to neural PDE surrogates. Without comparison on these benchmarks, it's hard to assess ENMA's performance relative to strong baselines including DPOT and MPP.   I believe evaluating on at least one of these would substantially strengthen the empirical results.

---

> ### Author Rebuttal · Authors · 2025-07-29
>
> We thank the reviewer for the detailed and constructive feedback. We address the raised concerns point-by-point below.
> ## **Typographical and Formatting Issues (W1)**
>  We appreciate the reviewer highlighting these issues. All typographical and formatting inconsistencies (e.g., Figure 1 alignment, inconsistent section headers) will be corrected in the revised version.
> ## **Clarity of the Method Section (W2)**
> Thanks for the suggestion. In the final version, we will restructure this section to clearly distinguish between (i) model architecture, (ii) training procedure, and (iii) inference pipeline. We believe this will improve the readability and understanding of ENMA’s generative framework.
> ## **Contribution and novelty (W3)**
> While we acknowledge that the components of ENMA—causal transformers, masked decoding, and flow matching—are based on existing ideas, their integration within ENMA offers an entirely new paradigm for PDE generation. Specifically:
> - ENMA performs token-wise autoregressive generation entirely in a continuous latent space. In contrast to prior work relying on discrete tokenization (e.g., Zebra) or full-frame diffusion (e.g., AR-DiT), ENMA combines the fine-grained generation of LLM-style models with the smooth nature of continuous representations.
> - We demonstrate that in-context generation is possible in this setting—an advance over previous generative operator learning frameworks, which typically rely on vector quantization.
> - On the encoding side, we propose a geometry-aware cross-attention encoder to project irregular spatial inputs onto a structured latent grid. This enables us to apply causal 3D CNNs and spatial patchification efficiently (Appendix D.2.1). This strategy is novel and we believe is an interesting direction; a concurrent work [1] proposed the same idea to patchify physical fields, which we will cite in the revised version.
> - We introduce a novel metric, the Fréchet Physics Distance (FPD), inspired by vision benchmarks, to evaluate generative modeling performance in physical systems (Appendix F.1.1).
>
> As for the ablations, we detail below existing (Appendix F) and some additional experiments, following the reviewer's suggestion. We agree that this should be better referenced in the manuscript, and we will update the main text accordingly.
> ### **Existing Ablations**
> - **Compression ratio** (Appendix F.1.2): We show that reducing the compression improves performance on Vorticity, making ENMA competitive while maintaining strong results on Wave and Gray-Scott.
> - **Geometry-aware encoding** (Appendix F.2.3): We compare reconstruction performance with and without the geometry-aware component. Across various input sparsity settings (full, half, sparse), the geometry-aware encoder consistently improves reconstruction accuracy.
> ### **New Ablations**:
> We have evaluated the importance of: (i)  causal encoder, (ii) flow-matching vs deterministic component, (iii) scheduler.
>
> - **Causal CNN**: We ablate the use of the causal convolution module in the encoder to assess its impact on temporal feature representation. Table R1 presents an ablation of the causal component in ENMA. Compared to the non causal autoencoder, the causal version uses causal layers and adds temporal compression, reducing the number of tokens by half. Despite this, it matches or outperforms the non causal version—especially in low-data settings— indicating that the causal layer helps in providing more informative tokens.
>
>     Table R1: Ablation of the Causal component (Advection). Metrics in Relative MSE
>     |Sampling|Relative MSE||
>     |-|:-:|:-:|
>     ||Causal|Non causal|
>     |100\% | 5.16e-3 | 4.55e-3 |
>     ||||
>     |50\%  | 6.98e-3 | 6.02e-3 |
>     |||
>     |20\%  | 1.66e-2 | 2.01e-2 |
> - **Latent dynamics modeling**: We compare the flow-matching-based generation to both a diffusion-based latent model and a fully deterministic alternative. In table R2, we observe that a Flow matching framework works better than a deterministic approach, while offering the advantages of generative modeling. Diffusion didn't converge correctly in our experiments leading to lower performance.
>
>     Table R2: Ablation of the generation mode (Vorticity).
>     |Training Objective |Relative MSE|
>     |-|:-:|
>     |Flow Matching (ENMA) |0.0644|
>     |Diffusion|0.7579|
>     |Deterministic |0.0879|
> - **Scheduler**: We evaluate the importance of the scheduler strategy for choosing the number of tokens to generate. In table R3, we observe that the choice of the scheduler is not of particular importance; all strategies lead to very similar performance.
>
>     Table R3: Ablation of the Scheduler (Vorticity).
>     |Scheduler|Relative MSE|
>     |-|:-:|
>     |Linear |0.0619|
>     |Cosine  |0.0612|
>     |Exponential  |0.0615|
>     |Reverse exponential |0.0618|
>
> These new experiments will be included in the final version to further validate ENMA’s design choices.
> ## **Choice of Benchmarks (W4)**
> Our primary goal was to assess ENMA’s capacity to generalize across parametric PDEs—where PDE parameters vary  from one trajectory to another while most existing public benchmarks exhibit only limited parametric variability: for each physical system, we consider 1200 distinct PDE instances, each with a different set of parameters, when available benchmarks  only include a few (about 10) parametric instances. Thus current public benchmark, do not allow to fully evaluate our target parametric setting.  Nevertheless, following your suggestion, we  include new results on two public benchmarks:
> - **CylinderFlow** [2]: A well-known time-dependent PDE on irregular spatial grids, used to evaluate ENMA’s encoder-decoder stage against baselines (as Table 1 in the main text).
>     Table R4 shows the results of the auto encoding and time-stepping tasks. This experiment clearly highlights the capabilities of ENMA to manage irregular grids, both for reconstruction and time-stepping. ENMA also is more efficient in terms of compression due to the temporal compression proposed ($\times 22$ compression rate for ENMA, whereas baselines have $\times 11$ compression rate).
>
>     Table R4: Results of the reconstruction task on the cylinder flow dataset.
>     |Sampling|Baseline|Reconstruction|Time-stepping|Compression
>     |-|-|:-:|:-:|:-:|
>     || GINO| 6.85e-1 | 1.59e0|$\times 11$
>     |100\%| AROMA| 9.59e-1 | 6.59e-1 | $\times 11$
>     || CORAL | 5.06e-2 | 7.84e-1 | $\times 11$
>     || ENMA | **7.78e-3** | **1.44e-1** | **$\times 22$**
>     |||||
>     || GINO| 6.85e-1 | 1.59e0 |
>     |50\%| AROMA| 9.86e-3 | 6.78e-1 |
>     || CORAL | 4.96e-2 | 7.83e-1|
>     || ENMA | **7.80e-3** | **1.44e-1**|
>     |||||
>     || GINO| 6.85e-1 | 1.59e0 |
>     |20\%| AROMA| 1.06e-2 | 6.82e-1 |
>     || CORAL | 5.14e-2 | 7.83e-1 |
>     || ENMA | **7.87e-3** | **1.44e-1** |
>
> - **Active Matter** (The Well [3]): A complex system on a $256 \times 256$ regular grid with 11 input channels. We compare ENMA’s generative capacity under temporal conditioning to autoregressive baselines (Table 2 of the manuscript). In Table R5, we observe that ENMA is close to AVIT while operating in a very compressed space. Relaxing this compression ratio would yield to better performance, as we observed in Appendix F.1.2.
>
>     Table R5: Results of the temporal generation task on the Active Matter dataset.
>     |Baseline|Temporal conditioning|Compression ratio|
>     |-|:-:|:-:|
>     |AVIT|0.383| $\times 1$|
>     |ENMA|0.412|$\times 10$|
>
> Regarding baselines: Our AViT baseline uses the same architecture as MPP, minus the
> modules designed for multi-field modeling, which are not required in our setting.
>
> ## Questions
> 1. **Cross-attention vs masking**: ENMA has been designed to handle inputs defined on point clouds without requiring any prior mesh structure. Cross-attention with geometry-aware bias is a mechanism that allows interpolation from arbitrary, unordered spatial coordinates onto a latent regular grid. This is important in many settings where observations are irregular and may change along time. This cannot be performed through masking which assumes inputs lie on a predefined grid with entries are potentially missing.
> 2. **Equation (4)**: Thank you for pointing this out. We used this notation for simplification, but we agree that this is misleading and we will revise the equation and text to clearly reflect ENMA’s ability to evaluate on arbitrary spatial grids.
> 3. **Compression ratio**: Compression rate refers to the number of points in the token space compared to the original spatio-temporal resolution. We will clarify this in the revision.
> 4. **Continuous tokens**: By "continuous latent tokens", we refer to standard dense embeddings—used commonly in neural operator models. The novelty here lies in the use of dense embeddings in a tokenwise generative autoregressive model. All the tokenwise generative models so far leverage token quantization and make use of a classifier as for the generative component like in LLMs. Here we extend this setting to the generation of continuous distributions. ViTs also use continuous tokens but cannot be used in a generative mode, contrarily to our model.
> 5. **Appendix F summarization**: We agree that Appendix F contains valuable content and will summarize its key findings in the main paper before the conclusion, and improve referencing throughout the text.
>
> We hope this clarifies the motivation, novelty, and empirical contributions of our work. We thank the reviewer again for the constructive feedback and would be happy to engage in further discussion.
>
> ## References
>
> [1] Wen et al., Geometry-Aware Operator Transformer as an Efficient and Accurate Neural Surrogate for PDEs on Arbitrary Domains, arXiv:2505.18781 (2025).
>
> [2] Fortunato et al. Learning mesh-based simulation with graph networks, International Conference on Learning Representations, 2021.
>
> [3] Ohana et al. The Well: a Large-Scale Collection of Diverse Physics Simulations for Machine Learning, NeurIPS 2024 Track Datasets and Benchmarks Poster, 2024.

---

> > ### Comment · Reviewer_aeqi · 2025-08-05
> >
> > I thank the authors for the detailed response. My main concerns have been addressed, and I will increase my score accordingly.

---

### Official Review · Reviewer_8eQS · 2025-07-02

**Clarity:** 2
**Significance:** 3
**Originality:** 2
**Rating:** 5
**Confidence:** 4

**Summary:**

1) Introduces ENMA, a continuous‐token autoregressive neural operator that models time‐dependent parametric PDEs in a compact latent space via masked spatial and causal temporal Transformers.

2) Encode: irregularly sampled spatio‐temporal inputs are interpolated onto a regular grid and compressed with causal 3D convolutions into latent tokens.

3) Generate: a causal Transformer predicts future latent frames, then a masked spatial Transformer + flow‐matching MLP decodes per‐token next‐state distributions without discrete quantization.

4) Decode: A continuous VAE decoder reconstructs full physical fields from generated latent sequences, supporting in‐context learning and probabilistic forecasting.

**Questions:**

1) Can the authors provide wall‑clock benchmarks for end‑to‑end inference and memory usage as a function of token count and mask schedule?

2) How does performance change without flow matching (e.g., using a direct MLP regression) or with different masking schedules in the spatial decoder?

3) Have you tested ENMA on irregular meshes or 3D PDEs to validate its claimed domain‐agnostic adaptability?

4) How well do the flow‐matching–derived uncertainties correlate with actual forecasting errors, especially under out‐of‐distribution parameter shifts?

5) Can authors compare the proposed method with DiffusionPDE as a baseline?

**Ethical Concerns:**

["NO or VERY MINOR ethics concerns only"]

**Final Justification:**

My main concerns have been addressed, and I hope the authors incorporate all the new results and open-source the codebase as suggested. I am increasing my score from 4 to 5.

**Limitations:**

Yes

**Paper Formatting Concerns:**

No formatting issue

**Quality:**

2

**Strengths And Weaknesses:**

**Strengths**

1) Continuous latent autoregression avoids quantization artifacts and enables fine‐grained uncertainty estimation via flow‐matching.

2) Irregular‐grid compatibility: attention‐based encoder handles unordered inputs and varying sampling rates gracefully.

3) In‐context adaptation: conditions on either past target states or auxiliary trajectories to generalize across parameter regimes without retraining.

4) Scalable compression: combines spatial and temporal token compression while maintaining high reconstruction and forecasting fidelity.

**Weaknesses**

1) Compression trade‑off: High latent compression can degrade accuracy on complex dynamics (e.g., vorticity), highlighting limited expressiveness.

2) Computational overhead: Two-stage Transformers plus flow‑matching sampling incur nontrivial inference cost and memory use.

3) Benchmark scope: evaluation limited to five PDE families; lacks tests on stiff, multiphysics, or 3D systems from the PDEBench dataset.

4) Ablation gap: missing analysis isolating the impacts of masked decoding schedule, flow‐matching vs. simpler autoregression, and causal CNN design.

---

> ### Author Rebuttal · Authors · 2025-07-29
>
> We thank the reviewer for the detailed and thoughtful comments, which have helped us improve the quality and clarity of the paper. Below, we address each point raised.
> # Compression Trade-off (W1)
> We acknowledge that latent compression often leads to reduced performance, especially for complex PDEs with high-frequency components.  However, for high dimensional datasets, operating directly in the physical space is usually prohibitive with transformers, hence compressing is often required. We believe that our work highlight a direction for improving this trade-off for solving PDEs.
> In order to evaluate the influence of compression, in Appendix F.1.2, we ablate the compression ratio of ENMA across 2D datasets. We observe that reducing compression indeed significantly boosts ENMA’s performance, while still reducing the computational complexity compared to operating in the physical space, making it competitive on Vorticity with the best baselines and outperforming them on Wave and Gray-Scott.
> # Computational Cost (W2 & Q1)
> Regarding computational overhead, we agree that ENMA’s generative process incurs higher inference cost than deterministic models. This is inherent to generative methods leveraging iterative denoising or sampling. However, as shown in Appendix F.1.3, this overhead is limited and ENMA achieves strong performance with only a few autoregressive and denoising steps. We report in appendix section F.1.3 wall-clock inference times against other generative baselines (Zebra and AR-DiT), showing that ENMA is faster due to its lightweight design and flow-matching objective. This will be better referenced in the final version. As requested, we report below the memory usage as a function of token count for one sample. The experiment demonstrates a limited increase in memory requirements.
>
> Table R1: Memory consumption (Go) and inference time (s) wrt token count (on Vorticity).
> | Token Count | Memory (MB) | Time (s) |
> |-|:-:|:-:|
> | 64 | 3896 | 0.5 |
> | 512 | 3896 | 1.5 |
> | 1024 | 3897 | 5.7 |
> | 2048 | 3897 | 10.6 |
> | 4096 | 3898 | 20.9 |
> | 8192 | 3900 | 40.9 |
> | 16384 | 3904 | 83.7 |
>
> # New experiments on public benchmarks (W3 & Q3):
>
>
> We fully agree that evaluating on public datasets enhances the reproducibility and comparability of our results. While our primary goal was to assess ENMA’s capacity to generalize across parametric time-dependent PDEs—where PDE parameters may vary significantly from one trajectory to another—most existing public benchmarks exhibit only limited parametric variability. Thus, they do not fully reflect the challenges addressed in our setting. For each physical system, we generated 1200 distinct PDE instances, each with a different set of parameters, and10 trajectories for each instance with a distinct initial condition, leading to 12000 samples. In comparison, datasets from public benchmarks offer a different setting: each PDE system in these datasets only includes a few number of parametric instances (about 10 vs 1200 for our setting), offering a limited diversity for each PDE. Thus current public benchmark, focus on solving a variety of PDEs, but do not allow to fully evaluate the parametric setting which is our target in the paper.
>
> Nonetheless, ENMA is designed to be general and flexible, and it can readily be applied to a wide range of time-dependent PDE problems. To this end, we have included results on two public datasets that capture complementary aspects of ENMA’s capabilities:
> - **CylinderFlow**: A time-dependent PDE on an irregular grid. We evaluate ENMA’s encoder-decoder performance and compare to neural operator baselines designed for irregular spatial inputs. Table R2 shows the results of the auto encoding and time-stepping tasks as presented in table 1 in the manuscript. This experiment clearly highlights the capabilities of ENMA to manage irregular grids.  The time-stepping task is much better with ENMA's tokens, likely due to the temporal compression proposed ($\times 22$ compression rate for ENMA, whereas baselines have $\times 11$ compression rate).
>
>     Table R2: Results of the reconstruction task on the cylinder flow dataset. Metrics in Relative MSE on the test set. We highlight in bold the best performing model.
>     |Sampling|Baseline|Reconstruction|Time-stepping|Compression
>     |-|-|:-:|:-:|:-:|
>     || GINO| 6.85e-1 | 1.59e0|$\times 11$
>     |100\%| AROMA| 9.59e-1 | 6.59e-1 | $\times 11$
>     || CORAL | 5.06e-2 | 7.84e-1 | $\times 11$
>     || ENMA | **7.78e-3** | **1.44e-1** | **$\times 22$**
>     |||||
>     || GINO| 6.85e-1 | 1.59e0 |
>     |50\%| AROMA| 9.86e-3 | 6.78e-1 |
>     || CORAL | 4.96e-2 | 7.83e-1|
>     || ENMA | **7.80e-3** | **1.44e-1**|
>     |||||
>     || GINO| 6.85e-1 | 1.59e0 |
>     |20\%| AROMA| 1.06e-2 | 6.82e-1 |
>     || CORAL | 5.14e-2 | 7.83e-1 |
>     || ENMA | **7.87e-3** | **1.44e-1** |
>
> - **Active Matter**: A complex PDE on a 256×256 regular grid with 11 input fields. We test ENMA’s generative capabilities under temporal conditioning and compare to autoregressive transformer baselines (Table 2). In Table R3, we observe that ENMA is close to AVIT (also known as MPP) while operating in a very compressed space. Relaxing this compression ratio would yields to better performance, as we observed for other baselines, but we believe that operating in such latent space for high dimensional data such as Active Matter is essential.
>
>     Table R3: Results of the temporal generation task on the Active Matter dataset. Metrics in Relative MSE on the test set.
>     |Baseline|Temporal conditioning|Compression ratio|
>     |-|:-:|:-:|
>     |AVIT|0.383| $\times 1$|
>     |ENMA|0.412|$\times 10$|
>
> # Ablation studies on ENMA's components (W4, Q2)
> We appreciate the suggestions for further ablations. Note that Appendix F already includes several ablations—e.g., geometry-aware attention (F.2.3), compression ratio (F.1.2), sampling schedule and history length (F.1.4)—we include below new ablations, following the reviewer's suggestion:
>
> - **Masking schedules**: We evaluate decoding under different masking strategies (e.g., linear, exponential). In R4, we observe all strategies lead to similar performance.
>
>     Table R4: Ablation on the masking strategy on the test set (on Vorticity). Metrics in relative MSE.
>     |Scheduler|Relative MSE|
>     |-|:-:|
>     |Linear |0.0619|
>     |Cosine  |**0.0612**|
>     |Exponential  |0.0615|
>     |Reverse exponential |0.0618|
>
> - **Generative modeling choice**: We compare flow-matching with a denoising-based objective and a fully deterministic approach (table R5).  Flow matching works better than a deterministic approach, while offering the advantages of generative modeling (sampling etc).  Our experiments with diffusion training didn't converge correctly, leading to higher loss. Overall flow-matching is easier to train.
>
>     Table R5: Ablation of the generation method (on Vorticity). Metrics in Relative MSE on the test set.
>     |Training Objective |Relative MSE|
>     |-|:-:|
>     |Flow Matching (ENMA) |0.0644|
>     |Diffusion|0.7579|
>     |Deterministic |0.0879|
>
> - **Causal CNN**: Causal vs non causal convolutions in the encoder.
>  Table R6 presents an ablation of the causal component in ENMA. Compared to the non causal autoencoder, the causal version uses causal layers and adds temporal compression, reducing the number of tokens by half. Despite this, it matches or outperforms the non causal version—especially in low-data settings— indicating that the causal layer helps in providing more informative tokens.
>
>     Table R6: Ablation of the Causal component (on Advection). Metrics in Relative MSE on the test set.
>     |Sampling|Relative MSE||
>     |-|:-:|:-:|
>     ||Causal|Non causal|
>     |100\% | 5.16e-3 | 4.55e-3 |
>     ||||
>     |50\%  | 6.98e-3 | 6.02e-3 |
>     ||||
>     |20\%  | 1.66e-2 | 2.01e-2 |
>
> # Uncertainty Calibration under Distribution Shifts (Q4)
>
> In Appendix F.1.1 (Figures 21–22), we evaluate UQ in the in-distribution setting by plotting CRPS and RMSCE across forecast steps. Following your suggestion, we have added the relative L2 error at each timestep in Table R7, highlighting its correlation with the UQ metrics. As expected in autoregressive models, L2 increases over time due to error accumulation. CRPS follows a similar trend, indicating that probabilistic accuracy degrades with the prediction horizon, while RMSCE decreases, suggesting the model is initially overconfident and becomes better calibrated as it increasingly relies on its own predictions—despite the rising forecast error.
>
> Table R7: Uncertainty calibration wrt time steps (on Combined).
> | Metrics | t=0 | t=2 | t=4 | t=6 | t=8 | t=10 | t=14 | t=16 | t=18 |
> |-|:-:|:-:|:-:|:-:|:-:|:-:|:-:|:-:|:-:|
> | CRPS |0.0006       |0.0008       |0.001       |0.0013       |0.0015       |0.0018       |0.0020       | 0.0022       |0.0023       |
> | RMSCE |0.141       |0.117       |0.106       |0.096       |0.087       |0.076      |0.0697       |0.064       |0.060       |
> | Relative L2 |0.016       |0.022       |0.025       |0.040       |0.052       |0.050       |0.058       |0.071       |0.072       |
>
> We will also investigate the behavior in OoD settings in additional experiments. We expect ENMA to yield larger confidence intervals under distribution shift.
>
> # DiffusionPDE baseline (Q5)
> Thanks for mentioning DiffusionPDE—we will quote this work in the paper. DiffusionPDE however belongs to a different family of generative methods: it generates trajectories in one shot and not through an auto-regressive process which is the focus of the paper. It also leverages a PDE loss requiring the knowledge of the PDE equations, when we consider pure data-driven approaches. It is not designed for solving time-dependent parametric PDEs but one PDE instance at a time. Hence direct comparison is not possible.
>
>  We thank the reviewer again for their valuable feedback and are happy to engage further if needed.

---

> > ### Comment · Reviewer_8eQS · 2025-08-03
> > **Response to Author's Rebuttal**
> >
> > Thank you for the authors' responses. My main concerns have been addressed, and I hope the authors incorporate all the new results and open-source the codebase as suggested. Taking into account the revisions and the concerns raised by other reviewers, I am increasing my score from 4 to 5.

---

> > > ### Author Response · Authors · 2025-08-05
> > > **Response to Reviewer 8eQS**
> > >
> > > We thank Reviewer 8eQS for their careful consideration of our responses. We will make sure to include all their suggestions in the updated manuscript.

---

### Official Review · Reviewer_ydbq · 2025-07-02

**Clarity:** 3
**Significance:** 3
**Originality:** 3
**Rating:** 5
**Confidence:** 4

**Summary:**

The authors propose a new generative neural operator that is able to both operate on a token level and in a continuous space. The framework is made of the following components: 1) an encoder which projects irregularly sampled points into a fixed size continuous embedding using geometry aware attention, 2) a generative model which uses a causal transformer to predict future latent states temporally, then a spatial transformer with flow matching to performtokenwise spatial generation, and 3)a decoder to map the generated tokens back to the data space. ENMA shows clear empirical improvements over existing baselines.

**Questions:**

- **justification needed**: can the authors provide additional justification for choosing flow-matching over other generative frameworks? an analysis of the pros and cons of a few alternatives would suffice.
- **Sharing code**: please share the source code to replicate your results anonymously.
- **Missing some ablations**: an interesting ablation would be to vary the compression rate and see if less compression would indeed help complex system like vorticity.
- **More complex setups**: it would be interesting to see how ENMA fares in modelling higher dimensional systems
- **Small changes**:
    - can you explain in a few words what relative mse is in the experimental section and why you use it?
    - can you elaborate more on the FNO architecture used in the main paper?
- **please clarify**: I understand that the compression rate refers to the size of the latent space over the size of the data space. I don't understand why this is included as a comparison criteria in Table 1. Isn't it a parameter the authors can control in other baselines as well?
-Nitpicks
    - Title of 3.1: tokenwise should be capitalized
    - Table 1 caption: random bolding, was a bit misealding. Either bold all the metrics computed or none.

**Ethical Concerns:**

["NO or VERY MINOR ethics concerns only"]

**Final Justification:**

The authors have addressed all my concerns.

**Limitations:**

Yes.

**Paper Formatting Concerns:**

None.

**Quality:**

3

**Strengths And Weaknesses:**

## Strengths
- The paper is well-written, the figures convey clear ideas and help support the ideas of the text
- The model is tested in various setups (reconstruction for the encoder-decoder, temporal conditioning and ivp with context) with clear improvements in performance.
- Extensive ablations and additional details are provided in the appendix.

## Weaknesses
Overall, I am willing to bump up my score if the authors 1) improve the justification of some components in their model, and 2) provide additional empirical evidence testing their model in more complex PDE setups. More detailed in Questions.

---

> ### Author Rebuttal · Authors · 2025-07-29
>
> We thank the reviewer for the thoughtful and constructive feedback. Below, we address each of the points raised.
> # Ablations: Components of ENMA (W1, Q1 and Q3)
> We provide justifications and ablations concerning ENMA's components.
> We first highlight that several ablations already included in the appendix directly address some of the reviewer’s suggestions (they will be better referenced in the main text):
> ### Existing Ablations
> - **Compression ratio**:(Appendix F.1.2): We provide ablations across multiple 2D datasets (Wave, Gray-Scott, Vorticity). Reducing compression substantially improves ENMA’s performance. On Wave and Gray-Scott, ENMA surpasses all baselines, and on Vorticity, it becomes competitive.
> - **Geometry-aware encoding** (Appendix F.2.3): We compare reconstruction performance with and without the geometry-aware component. Across various input sparsity settings (full, half, sparse), the geometry-aware encoder consistently improves reconstruction accuracy.
> ### New Ablations
> We have evaluated the importance of: (i)  causal encoder, (ii) flow-matching vs deterministic component, (iii) scheduler.
> - **Causal CNN**: We ablate the use of the causal convolution module in the encoder to assess its impact on temporal feature representation.
>     Table R1 presents an ablation of the causal component in ENMA. Compared to the non causal autoencoder, the causal version uses causal layers and adds temporal compression, reducing the number of tokens by half. Despite this, it matches or outperforms the non causal version—especially in low-data settings— indicating that the causal layer helps in providing more informative tokens.
>
>     Table R1: Ablation of the Causal component on the Advection dataset. Metrics in Relative MSE on the test set (please refer to table 1 for the sampling strategy).
>     |Sampling|Relative MSE||
>     |-|:-:|:-:|
>     ||Causal|Non causal|
>     |100\% | 5.16e-3 | 4.55e-3 |
>     ||||
>     |50\%  | 6.98e-3 | 6.02e-3 |
>     |||||
>     |20\%  | 1.66e-2 | 2.01e-2 |
>
> - **Latent dynamics modeling (Q1)**: We chose flow matching due to its well-established advantages: more stable training than diffusion models and fewer sampling steps during inference. Appendix F.1.4 shows that ENMA achieves strong results with as few as 5 steps. To further support this choice, we have added a new experiment comparing ENMA with a diffusion-based model and a fully deterministic model. Our results confirm that flow matching leads to more stable training and better accuracy, while maintaining the uncertainty estimation benefits of generative modeling. Diffusion didn't converge correctly in our experiments leading to lower performance.
>
>     Table R2: Ablation of the generation mode on the Vorticity dataset.
>     |Training Objective |Relative MSE|
>     |-|:-:|
>     |Flow Matching (ENMA) |0.0644|
>     |Diffusion|0.7579|
>     |Deterministic |0.0879|
>
> - **Scheduler**:  We evaluated different scheduling strategies for the number of masks to generate at a given step. In table R3, we observe that all strategies lead to similar performance.
>
>     Table R3: Ablation of the Scheduler on the Vorticity dataset.
>     |Scheduler|Relative MSE|
>     |-|:-:|
>     |Linear |0.0619|
>     |Cosine  |**0.0612**|
>     |Exponential  |0.0615|
>     |Reverse exponential |0.0618|
>
> These new experiments will be included in the final version to further validate ENMA’s design choices.
> # New benchmarks on complex setups (W2 and Q4)
> To evaluate ENMA on more challenging PDE scenarios, we have added results on two publicly available benchmarks:
> - **CylinderFlow** [1]: A time-dependent PDE on a natively irregular spatial grid. We report encoder-decoder performance and compare to neural operator baselines designed for irregular grids. Table R4 shows the results of the auto encoding and time-stepping tasks corresponding to table 1 in the manuscript. This experiment clearly highlights the capabilities of ENMA to manage irregular grids.  The time-stepping task is much better with ENMA's tokens, likely due to our temporal compression ($\times 22$ compression rate for ENMA, when baselines have $\times 11$ compression rate).
>
>     Table R4: Results of the reconstruction task on the cylinder flow dataset. Metrics in Relative MSE on the test set. In bold, the best performing model.
>     |Sampling|Baseline|Reconstruction|Time-stepping|Compression
>     |-|-|:-:|:-:|:-:|
>     || GINO| 6.85e-1 | 1.59e0|$\times 11$
>     |100\%| AROMA| 9.59e-1 | 6.59e-1 | $\times 11$
>     || CORAL | 5.06e-2 | 7.84e-1 | $\times 11$
>     || ENMA | **7.78e-3** | **1.44e-1** | **$\times 22$**
>     |||||
>     || GINO| 6.85e-1 | 1.59e0 |
>     |50\%| AROMA| 9.86e-3 | 6.78e-1 |
>     || CORAL | 4.96e-2 | 7.83e-1|
>     || ENMA | **7.80e-3** | **1.44e-1**|
>     |||||
>     || GINO| 6.85e-1 | 1.59e0 |
>     |20\%| AROMA| 1.06e-2 | 6.82e-1 |
>     || CORAL | 5.14e-2 | 7.83e-1 |
>     || ENMA | **7.87e-3** | **1.44e-1** |
>
> - **Active Matter** (The Well [2]) : A complex PDE defined on a $256 \times 256$ grid with 11 input fields. We evaluate ENMA’s temporal generation ability against autoregressive transformer baselines (Table 2 of the manuscript, temporal conditioning setup). In Table R5, we observe that ENMA is close to AVIT (also known as MPP) while operating in a very compressed space. Relaxing this compression ratio would yield to better performance, as observed in F.1.2, but we believe that operating in such latent space for high dimensional data such as Active Matter is essential.
>
>     Table R5: Results of the temporal generation task on the Active Matter dataset. Metrics in Relative MSE on the test set.
>     |Baseline|Temporal conditioning|Compression ratio|
>     |-|:-:|:-:|
>     |AVIT|0.383| $\times 1$|
>     |ENMA|0.412|$\times 10$|
>
> # Sharing code (Q2)
> According to NeurIPS policy one cannot add links in the rebuttal, hence we cannot provide the code now. However, we will publicly release both the codebase and datasets upon notification of acceptance.
> # Small changes (Q5):
> - **Relative MSE**: This is a standard metric in PDE learning. It normalizes error by the energy of the ground truth, which is particularly important for comparing PDEs with different magnitude scales. It is defined as:
>
> $$\text{Relative MSE}=\frac{1}{n} \sum_{i=1}^n \frac{(y_i - \hat{y}_i)^2}{||y||_2^2}$$
>
> with $ \lVert y \rVert_2^2 = \sum_{j=1}^{n} y_j^2 $
> - **FNO architecture**: In the main text, table 1 evaluates the influence of ENMA’s encoder-decoder choice  in the reconstruction task. We compare several encoder-decoder components baselines using FNO as the time stepper. This allows us to isolate the role of the encoder-decoder model. Note we also performed tests with a U-Net in Appendix F.2.3, with similar conclusions.  All the details on the architectures are provided in Appendix E.2.3.
> In table 2 of the manuscript, we benchmarked FNO against other time steppers, architecture details are in Appendix E.1.2.
>
> # Clarifications (Q6):
> - **Compression rate in table 1**: While most frameworks can control the compression rate, they proceed differently.
>  OFormer operates directly in the physical space (pointwise, no compression). GINO, AROMA, and CORAL compress only in space. In contrast, ENMA compresses both space and time.  We tuned latent channel dimensions for each method to achieve optimal trade-offs between compression and accuracy. This is indicated by the different compression ratios shown in table 1.
> For a better comparison, we provide (Table R6, R7) new experimental results, where the spatial compression is the same for all methods and ENMA performs an additional compression in the time dimension, hence a double compression ratio compared to other methods. This will be discussed in the final version.
>
>     Table R6: Additional results on the Advection dataset with unified spatial compression rate. Metrics in Relative MSE. Models ending with -old indicates results proposed in table 1 of the manuscript, while models ending with -new are the updated results.
>     |Sampling|Baseline|Advection|||
>     |-|-|:-:|:-:|:-:|
>     |Sampling|Baseline|Reconstruction|Time-stepping|Compression|
>     || GINO-new| 3.15e-1 | 8.55e-1 | $\times 2$|
>     |100\%| GINO-old| 5.74e-2 | 7.89e-1 | $\times 0.5$|
>     |||||
>     || GINO-new| 3.21e-1 | 8.64e-1 | |
>     |50\%| GINO-old| 6.64e-2e-2 | 7.87-1 | |
>     |||||
>     || GINO-new| 3.54e-1 | 9.11e-1 | |
>     |20\%| GINO-old| 9.13e-2 | 7.96e-1 | |
>
>     Table R7: Additional results on the Vorticity dataset with unified spatial compression rate. Metrics in Relative MSE. Models ending with -old indicates results proposed in table 1 of the manuscript, while models ending with -new are the updated results.
>     |Sampling|Baseline|Vorticity|||
>     |-|-|:-:|:-:|:-:|
>     |Sampling|Baseline|Reconstruction|Time-stepping|Compression|
>     |100\%| CORAL-new | 4.63e-1 | 9.18e-1 | $\times 8$ |
>     || CORAL-old | 4.50e-1 | 9.85e-1 | $\times 2$|
>     |||||
>     |50\%| CORAL-new | 4.95e-1| 9.22e-1 |
>     || CORAL-old | 4.93e-1 | 9.85e-1 |
>     |||||
>     |20\%| CORAL-new | 6.89e-1 | 9.37e-1 |
>     || CORAL-old | 7.59e-1 | 9.87e-1 |
>
> - **Formatting**:
> We thank the reviewer for catching the issues, such as the capitalization of “Tokenwise” in Section 3.1. We will correct these in the camera-ready version.
> - **Bold formatting in table 1**:
> We bolded the best score and underlined the second-best. We agree this should be clarified in the caption to avoid confusion.
>
> We hope this response resolves the reviewer’s concerns. We appreciate the detailed and constructive
> feedback and would be happy to further engage in discussion
>
> ### References
> [1] Battaglia et al, Learning mesh-based simulation with graph networks, International Conference on Learning Representations, 2021.
>
> [2] Ohana et al., The Well: a Large-Scale Collection of Diverse Physics Simulations for Machine Learning, NeurIPS 2024 Track Datasets and Benchmarks Poster, 2024.

---

> ### Comment · Reviewer_ydbq · 2025-08-06
>
> Thank you, you have addressed all my concerns and I raised my score accordingly.

---

> > ### Author Response · Authors · 2025-08-07
> > **Response to Reviewer ydbq**
> >
> > We thank Reviewer ydbq for their careful consideration of our responses. We will make sure to include all their suggestions in the updated manuscript.

---

### Official Review · Reviewer_HhGJ · 2025-07-06

**Clarity:** 4
**Significance:** 4
**Originality:** 4
**Rating:** 5
**Confidence:** 3

**Summary:**

A method for dynamics propagation in latent space is introduced. It separates the temporal forecast, achieved by a temporal transformer, from the spatial forecast, achieved by a masked autoregressive (MAR) model, by concatenating the two operations and making the MAR input conditional on the output of the temporal transformer. After the MAR, a flow-matching model aligns the predicted spatial latent with the ground truth. The flow-matching model consists in an ODE \dot x = v(x) which uses an MLP as velocity field.

These ingredients are brought together with an encoder that is capable of creating latent vectors on irregularly spaced time grids.

Extensive experiments on simulated data and comparison to many other methods show that the proposed method performs favorably compared to existing literature.

**Questions:**

Would it be possible to have further experiments on public benchmarks?

Would it be possible to see the benefits and necessity of the individual components of the method by ablating them?

**Ethical Concerns:**

["NO or VERY MINOR ethics concerns only"]

**Final Justification:**

This work is solid and I am happy to maintain my score of 5 which I think best reflects the character of this work. If asked, I will advocate for acceptance.

**Limitations:**

yes

**Quality:**

4

**Strengths And Weaknesses:**

The paper is easy to read and the diagrams are self-explanatory. The method is presented clearly and set in context.
The proposed method is a step forward towards accurate dynamics propagation on potentially lower-resolution latents without drifting off the trajectory.

Experiments are rigorous and extensive. However, it would be very useful to have numbers on some public benchmarks. Are there sub-datasets in PDEbench, PDEarena, or The Well, which would lend themselves to benchmark the proposed method and compare to others?

---

> ### Author Rebuttal · Authors · 2025-07-29
>
> We sincerely thank the reviewer for the constructive and encouraging feedback. We appreciate the reviewer’s positive feedback on the clarity of the method and the accompanying diagrams. Below, we address the main points raised.
>
> # New experiment: 2 new Public Benchmarks (W1 & Q1)
>
> We fully agree that evaluating on public datasets enhances the reproducibility and comparability of our results. While our primary goal was to assess ENMA’s capacity to generalize across *parametric time-dependent PDEs*—where PDE parameters may vary significantly from one trajectory to another—most existing public benchmarks exhibit only limited parametric variability. Thus, they do not fully reflect the challenges addressed in our setting. For each physical system, we generated 1200 distinct PDE instances, each with a different set of parameters, and 10 trajectories for each instance with a distinct initial condition, leading to 12000 samples. In comparison, datasets from public benchmarks offer a different setting: each PDE system in these datasets only includes a few number of parametric instances (about 10 vs 1200 for our setting), offering a limited diversity for each PDE. Thus current public benchmark, focus on solving a variety of PDEs, but do not allow to fully evaluate the parametric setting which is our target in the paper.
>
> Nonetheless, ENMA is designed to be general and flexible, and it can readily be applied to a wide range of time-dependent PDE problems. To this end, we have included results on two public datasets that capture complementary aspects of ENMA’s capabilities:
>
> - **CylinderFlow** [1]: A benchmark with natively irregular spatial grids, used to assess ENMA’s encoding/decoding capabilities for irregular meshes in comparison to existing neural operator methods (corresponding to Table 1 in the paper). We report reconstruction performance, time-stepping (used with a FNO for all benchmarks for fair comparison) and the compression ratio used. This experiment clearly highlights the capabilities of ENMA to manage irregular grids.  The time-stepping task is much better with ENMA's tokens, likely due to the temporal compression proposed ($\times 22$ compression rate for ENMA, whereas baselines have $\times 11$ compression rate).
>
>     Table R1: Results of the reconstruction task on the cylinder flow dataset. Metrics in Relative MSE on the test set. We highlight in **bold** the best performing model.
>     |Sampling|Baseline|Reconstruction|Time-stepping|Compression
>     |-|-|:-:|:-:|:-:|
>     || GINO| 6.85e-1 | 1.59e0|$\times 11$
>     |100\%| AROMA| 9.59e-3 | 6.59e-1 | $\times 11$
>     || CORAL | 5.06e-2 | 7.84e-1 | $\times 11$
>     || ENMA | **7.78e-3** | **1.44e-1** | **$\times 22$**
>     |||||
>     || GINO| 6.85e-1 | 1.59e0 |
>     |50\%| AROMA| 9.86e-3 | 6.78e-1 |
>     || CORAL | 4.96e-2 | 7.83e-1|
>     || ENMA | **7.80e-3** | **1.44e-1**|
>     |||||
>     || GINO| 6.85e-1 | 1.59e0 |
>     |20\%| AROMA| 1.06e-2 | 6.82e-1 |
>     || CORAL | 5.14e-2 | 7.83e-1 |
>     || ENMA | **7.87e-3** | **1.44e-1** |
>
> - **Active Matter (The Well)** [2]: A complex physical system involving collective behavior, used to evaluate ENMA’s performance against autoregressive transformer baselines for temporal generation (corresponding to Table 2 of the manuscript). In Table R2, we observe that ENMA is close to AVIT (also known as MPP) while operating in a very compressed space. Relaxing this compression ratio would yield to better performance, as we observed in Appendix F.1.2, but we believe that operating latent space for heavy data such as Active Matter is essential.
>
>    Table R2: Results of the temporal generation task on the Active Matter dataset. Metrics in Relative MSE on the test set.
>     |Baseline|Temporal conditioning|Compression ratio|
>     |-|:-:|:-:|
>     |AVIT|0.383| $\times 1$|
>     |ENMA|0.412|$\times 10$|
>
> # Ablation Studies: Benefits of ENMA's components (Q2)
>
> We appreciate the suggestion to further clarify the contribution of ENMA’s architectural components. Below we summarize both existing and new ablation studies. We will add an ablation section in the main text of the final manuscript with references to the different experiments :
>
> ### Existing ablations
> - **Compression ratio** (Sec. F.1.2): We show that reducing the compression improves performance on Vorticity, making ENMA competitive. On Wave and Gray-Scott, ENMA was on par with existing baselines at a higher compression rate and now outperforms them.
> - **Sampling schedule** (Sec. F.1.4): We evaluate ENMA’s predictive accuracy as a function of the number of autoregressive and denoising steps, showing that accurate forecasts can be obtained with few steps.
> - **History length** (Sec. F.1.4): We observe consistent performance gains with longer input histories, in line with ENMA’s autoregressive formulation.
> - **Geometry-aware encoding** (Sec. F.2.3): We compare reconstruction performance with and without the geometry-aware component. Across various input sparsity settings (full, half, sparse), the geometry-aware encoder consistently improves reconstruction accuracy.
>
> ### New ablations
> We have evaluated the importance of: (i)  causal encoder, (ii) generative vs deterministic component, (iii) scheduler.
>
> - **Causal CNN**: We compare a causal convolution module in the encoder with a non causal encoder, to assess its impact on temporal feature representation.
>     Table R3 presents an ablation of the causal component in ENMA. Compared to the non causal autoencoder, the causal version adds temporal compression, reducing the number of tokens by half. Despite this, it matches or outperforms the non causal version—especially in low-data settings— indicating that the causal layer helps in providing more informative tokens.
>
>     Table R3: Ablation of the Causal component on the Advection dataset. Metrics in Relative MSE on the test set (please refer to table 1 for the sampling operation).
>     |Sampling|Relative MSE||
>     |-|:-:|:-:|
>     ||Causal|Non causal|
>     |100\% | 5.16e-3 | 4.55e-3 |
>     ||||
>     |50\%  | 6.98e-3 | 6.02e-3 |
>     ||||
>     |20\%  | 1.66e-2 | 2.01e-2 |
>
> - **Latent dynamics modeling**: We compare the flow-matching-based generation to both a diffusion-based latent model and a fully deterministic alternative, to quantify the benefit of continuous stochastic trajectory modeling. In table R4, we observe that a Flow matching framework works better than a deterministic approach, while offering the advantages of generative modeling. We also tested a diffusion training objective, but it didn't converge correctly in our experiments leading to lower performance. Overall flow-matching is easier to train.
>
>     Table R4: Ablation of the generation mode on the Vorticity dataset. Metrics in Relative MSE on the test set.
>     |Training Objective |Relative MSE|
>     |-|:-:|
>     |Flow Matching (ENMA) |0.0644|
>     |Diffusion|0.7579|
>     |Deterministic |0.0879|
>
> - **Scheduler**: We evaluated different scheduling strategies to select the number of masks to generate at a given step. In table R5, we observe that the choice of the scheduler is not of particular importance; all strategies lead to very similar performance.
>
>     Table R5: Ablation of the Scheduler on the Vorticity dataset. Metrics in Relative MSE on the test set.
>     |Scheduler|Relative MSE|
>     |-|:-:|
>     |Linear |0.0619|
>     |Cosine  |**0.0612**|
>     |Exponential  |0.0615|
>     |Reverse exponential |0.0618|
>
> These new experiments will be included in the final version to further validate ENMA’s design choices.
>
> We hope this addresses the reviewer’s suggestions and helps clarify the motivation and robustness of the proposed method. We thank again the reviewer for their thoughtful and detailed assessment.
>
> ### References
> [1] Battaglia et al, Learning mesh-based simulation with graph networks, International Conference on Learning Representations, 2021.
>
> [2] Ohana et al., The Well: a Large-Scale Collection of Diverse Physics Simulations for Machine Learning, NeurIPS 2024 Track Datasets and Benchmarks Poster, 2024.

---

> > ### Comment · Reviewer_HhGJ · 2025-08-06
> > **Thank you for the detailed response**
> >
> > Thanks for the detailed response. This is solid work and I think all reviewers' scores reflect this at this moment. I will maintain my score of 5

---

### Decision · Program_Chairs · 2025-09-17

**Decision:**

Accept (spotlight)

**Comment:**

This paper presents ENMA, a novel generative neural operator that leverages tokenwise autoregression to model complex spatio-temporal dynamics arising from time-dependent parametric PDEs, integrating masked spatial and temporal transformers with continuous flow-matching in latent space, enabling both efficient compression and robust out-of-sample generalization.
The reviewers agree on the strong technical quality of the work, highlighting its clear motivation, thorough experimental evaluation (including new results on public benchmarks), and extensive ablation analysis. Some details and comparisons (e.g., to diffusion-based approaches) could be discussed in even greater depth.
All reviewers ultimately support acceptance, citing methodological novelty, evaluation quality, and impactful results.